# Implementation and assessment of a model including mixotrophs and the carbonate cycle (Eco3M_MIX-CarbOx v1.0) in a highly dynamic Mediterranean coastal environment (Bay of Marseille, France) (Part I): Evolution of ecosystem composition under limited light and nutrient conditions

Lucille Barré[1], Frédéric Diaz[1,†], Thibaut Wagener[1], France Van Wambeke[1], Camille Mazoyer[1], Christophe Yohia[2], Christel Pinazo[1]

[1]Aix Marseille Univ., Université de Toulon, CNRS, IRD, MIO, UM 110, 13288, Marseille, France
[2]Aix Marseille Univ., Université de Toulon, CNRS, IRD, OSU Institut Pythéas, 13288, Marseille France
[†]Deceased

*Correspondence to*: Lucille Barré (lucille.barre@mio.osupytheas.fr), Christel Pinazo (christel.pinazo@mio.osupytheas.fr)

**Abstract.** Many current biogeochemical models rely on an autotrophic versus heterotrophic food web representation. However, in recent years, an increasing number of studies have begun to challenge this approach. Several authors have highlighted the importance of protists capable of combining photoautotrophic and heterotrophic nutrition in a single cell. These mixotrophic protists are known to play an important role in the carbon cycle. Here, we present a new biogeochemical model that represents the food web using variable stoichiometry. It contains the classic compartments such as zooplankton, phytoplankton and heterotrophic bacteria, and a newly added compartment to represent two types of mixotrophic protists: non constitutive mixotrophs (NCM) and constitutive mixotrophs (CM). We demonstrate that the model correctly reproduces the characteristics of NCM and CM and proceed to study the impact of light and nutrient limitation on planktonic ecosystem structure in a highly dynamic Mediterranean coastal area: the Bay of Marseille (BoM, France), paying special attention to the dynamics of mixotrophic protists in these limiting conditions. In addition, we investigate the carbon, nitrogen and phosphorus fluxes associated with mixotrophic protists and showed that: (i) the portion of the ecosystem in percentage of carbon biomass occupied by NCM decreases when resources (nutrient and prey concentrations) decrease, although their mixotrophy allows them to maintain a carbon biomass almost as significant as the copepods one (129.8 and 148.7 mmolC m$^{-3}$, respectively), as photosynthesis increase as food source; (ii) the portion of the ecosystem in percentage of carbon biomass occupied by CM increases when nutrient concentrations decrease, due to their capability to ingest prey to supplement their N and P needs. In addition to provide new insights regarding the condition that lead to the emergence of mixotrophs in the BoM, this work provides a new tool to perform long-term studies and prediction of mixotrophs dynamics in coastal environments, under different environmental forcings.

Keywords: Mixotrophy, Bay of Marseille, Modelling, Ecosystem composition, Carbon fluxes, Climate change

# 1 Introduction

Marine protists play a crucial role in biogeochemical cycles and food webs (Sherr et al., 2007) and are typically classified as either photoautotrophs, capable of (strict innate) photosynthesis for nutrition, or phago-heterotrophs which rely on (strict) phagocytose for nutrition. However, several studies have shown that this classification may be overly simplistic as various micro-organisms can be both autotrophic and heterotrophic, either simultaneously or alternately, depending on environmental conditions (Pratt and Cairns, 1985; Dolan, 1992, Stoecker, 1998).

This combination of photo-autotrophy and phago-heterotrophy among protists is one example of mixotrophy, which has been observed in most planktonic functional groups except diatoms (Flynn et al., 2012). Generally, mixotrophic protists are divided into two major subsets depending on the type of photosynthesis, namely into constitutive mixotrophs (CM, innate photosynthesis) and, non-constitutive mixotrophs (NCM, acquired photosynthesis). CM are photo-autotrophs capable of ingesting prey using phagocytose when environmental conditions are not favourable (e.g., when nutrients limit growth). This subset includes nanoflagellates and dinoflagellates such as *Prymnesium parvum* and *Prorocentrum minimum,* respectively (Stoecker, 1998; Stoecker et al., 2017). NCM are phago-heterotrophs capable of photosynthesis to complement carbon

uptake. NCM temporarily acquire photosynthetic ability either by ingesting photosynthetic preys and sequestering their chloroplasts (kleptoplastidy) or by maintaining algal endosymbionts. NCM include ciliates and rhizaria such as *Laboea strobila, Strombidium capitatum* and *Collozoum spp* respectively (Stoecker, 1998; Mitra et al., 2016).

Mixotrophic protists play an important role in the marine carbon cycle. Due to their adaptability, these organisms are crucial for the transfer of matter and energy to the highest trophic levels, thus impacting the structure of planktonic communities by favouring the development of larger organisms (Ptacnick et al., 2004). Moreover, by switching the biomass maximum to larger organisms, carbon export increases in presence of mixotrophs. As instance, Ward and Follows (2016) compared the results from two food web models, only one accounted for mixotrophy, and showed that carbon export to depth increased by nearly 35% when mixotrophic protists were considered. By showing the significant effect of mixotrophic protists on the food web, these studies motivated their addition to current food web models (Jost et al., 2004; Mitra and Flynn, 2010).

In addition, mixotrophic protists are ubiquitous and can be found in various types of environments (Flynn et al., 2012; Hartmann et al., 2012; Stoecker et al., 2017). Some studies investigated mixotrophy in nutrient rich systems (eutrophized costal or estuarine systems) in the context of harmful algal blooms (HAB; Burkholder et al., 2008; Glibert et al., 2018). Typically, mixotrophy is studied in oligotrophic systems (Zubkhov and Tarran, 2008 ; Hartmann et al., 2012) including Mediterranean Sea. It was shown that Mediterranean Sea is highly oligotrophic especially in its Eastern Basin (Yacobi, 1995). Accordingly, some studies which aimed to investigate mixotrophy in protists have been conducted in the Mediterranean Sea. Several authors observed mixotrophic protists in both the Eastern and Western Basins, describe their distribution (Pitta and Giannakourou, 2000; Bernard and Rassoulzadegan, 1994) and quantify their effect on the ecosystem (Christaki et al., 1999; Dolan and Perez, 2000). However, few studies considered the effects of variable environmental parameters (i.e., temperature, salinity, pH, light and nutrients) on the spatial and temporal structuring of mixotrophic protists in the Mediterranean Sea.

Here we used a newly developed biogeochemical model (Eco3M_MIX-CarbOx, v1.0) to study the impact of light and nutrient limitations on the planktonic ecosystem structure in a Mediterranean coastal area, the Bay of Marseille (BoM) where we simulated a small volume of surface water (1 $m^3$). Eco3M_MIX-CarbOx contains a newly developed planktonic ecosystem model in which we consider mixotrophy. The mixotrophic compartment allow us to represent two types of mixotrophic protists: CM and NCM. We assessed it based on Stoecker's (1998) conceptual models of mixotrophy. Eco3m_MIX-CarbOx uses variable cellular quotas which allowed us to determine the nutritional state of the cell by comparing it to a reference quota. We conducted three specific case studies: (i) phytoplankton composition under typical forcings (light and nutrient concentrations as observed in the BoM) and specific events which all affect nutrient concentrations (Rhône River intrusions, water discharges from a local wastewater treatment plant and winter mixing), (ii) planktonic ecosystem composition under low light or nutrient conditions, paying special attention to the dynamics of mixotrophic protists, and (iii) comparing mixotrophic protists' C, N and P fluxes under limiting and non-limiting nutrients conditions.

Eco3M_MIX-CarbOx contains both a mixotrophy compartment and a representation of the carbonate system. The model description is split into two parts: (i) a description of how the organisms and their dynamics are represented in the model, with a particular focus on mixotrophic organisms, and (ii) a more detailed description of the carbonate module and the associated dynamics. While (i) is presented here, (ii) has been presented in a companion paper (Barré et al., 2023b).

## 2 Materials and methods

### 2.1 Study area

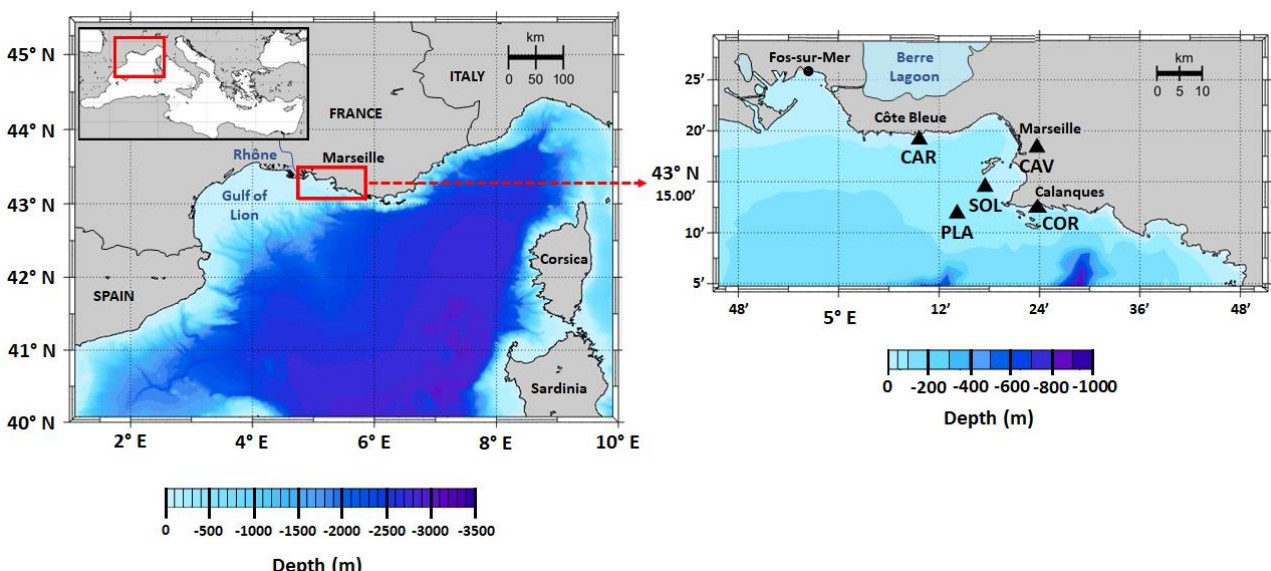

**Figure 1. Map of the study area showing the location of SOLEMIO station (SOL: 43°14.30' N, 5°17.30' E), Planier station (PLA: 43°11.96' N, 5°14.07' E), Carry buoy (CAR: 43°19.15' N, 5°09.64' E), Cinq Avenue station (CAV: 43°18.40' N, 5°23.70' E) and the Calanque de Cortiou (COR: 43°13.22' N, 5°25.40' E).**

The BoM is located in the North-Western (NW) Mediterranean Sea, in the eastern part of the Gulf of Lion near Marseille (Fig. 1). Due to this proximity to urbanized areas (e.g., Fos-sur-Mer and Berre Lagoon to the west, Fig. 1), it receives significant quantities of anthropogenic nutrients (especially ammonia and phosphate), chemical products, and organic matter from terrestrial and riverine sources and through atmospheric deposition (Djaoudi et al., 2017; Millet et al., 2018). Usually, significant inputs occur near the Calanque de Cortiou where wastewaters are discharged into the sea. During flood events, riverine and terrestrial runoff lead to significant inputs (Oursel et al., 2014). The biogeochemistry of the bay is also affected by its proximity to the Rhône River delta, located 35km to the west, as the Rhône River plume can be pushed eastwards under specific wind conditions which increases local productivity (Gatti et al., 2006; Fraysse et al., 2013, 2014). Other relevant processes that affect the biogeochemical functioning of the bay and add to its complex dynamics include strong

Mistral events (Yohia, 2017), upwelling events (Millot, 1990), eddies (Schaeffer et al., 2011) and intrusions of oligotrophic water masses via the Northern Current (Barrier et al., 2016; Ross et al., 2016).

In our model, environmental forcings are provided by in situ measurements of sea surface temperature (SST), salinity and atmospheric $p$CO$_2$ in combination with simulation data of wind speed and solar irradiance (Table 1). SST data was collected at the Planier station (PLA, Fig. 1) by the regional temperature observation network T-MEDNET (www.t-mednet.org, last access: 14 February 2023). Salinity data is from Carry buoy (CAR, Fig. 1) which forms part of the ROMARIN network (https://erddap.osupytheas.fr, last access: 14 February 2023). Atmospheric $p$CO$_2$ is recorded at the terrestrial station of Cinq Avenue (CAV, Fig. 1) by the AtmoSud regional atmospheric survey network (https://www.atmosud.org, last access: 14 February 2023), and AMC project (Aix-Marseille Carbon Pilot Study, https://www.otmed.fr/research-projects-and-results/result-2449, last access 14 February 2023). CAV station is located in the city Marseille and, the recorded $p$CO$_2$ values are representative of a highly urbanized environment, exhibiting strong maxima and large variations. Solar irradiance and wind speed were extracted from the WRF meteorological model (Yohia, 2017) for SOLEMIO station (Fig. 1).

To evaluate our model results, we compared the modelled total chlorophyll concentration to in situ measurements by using a dataset from the Service d'Observation en Milieu LITtoral (SOMLIT, https://www.somlit.fr/, last access 14 February 2023) which includes fortnightly measurements of total surface chlorophyll concentrations at SOLEMIO station.

**Table 1. Data types and their sources used to drive the environmental forcing during the 2017 model run.**

|  | Data type | Location | Time resolution |
|---|---|---|---|
| **SST** | Measurements | Planier station |  |
| **Salinity** | Measurements | Carry buoy |  |
| **Wind speed** | WRF model results | SOLEMIO station | Hourly |
| **Irradiance** | WRF model results | SOLEMIO station |  |
| **Atmospheric $p$CO$_2$** | Measurements | Cinq Avenues station |  |

## 2.2 Model description

We used the Eco3M_MIX-CarbOx model (v1.0) to simulate the food web using variable stoichiometry to study the evolution of the BoM ecosystem composition under light and nutrient limited conditions. The Eco3M_MIX-CarbOx model is a dimensionless (0D) model: we consider a volume of 1 m$^3$ of surface water at SOLEMIO station, in this volume the state variables only vary over time as the model is not coupled with a hydrodynamic model. Eco3M_MIX-CarbOx was developed to represent the dynamics of both mixotrophic protists (henceforth referred to as mixotrophs) and the carbonate system in the BoM. To obtain the present version of the Eco3M_MIX-CarbOx model, we developed a planktonic ecosystem model which contains mixotrophs, and added a modified version of the carbonate module from Lajaunie-Salla et al. (2021). The planktonic ecosystem model was developed using the Eco3M (Ecological Mechanistic and Molecular Modelling) platform (Baklouti et al., 2006a, b). The Eco3M platform allows the modelling of the first trophic levels by providing a process library

used to build different model configurations. It was developed in Fortran 90/95 and we used an Euler method to solve sink-source equation of each state variable. Based on results of previous studies (Jost et al., 2004; Mitra et al., 2014; Ward and Follows, 2016), we decided to represent mixotrophy and the carbonate cycle in the same model assuming that this would provide a more realistic representation of the carbonate cycle. In what follows we provide a brief description of Eco3M_MIX-CarbOx with a more detailed description of its mixotroph compartment. The carbonate system has been described in detailed in companion paper (Barré et al., 2023b).

Eco3M_MIX-CarbOx contains seven compartments, namely zooplankton, mixotrophs, phytoplankton, dissolved inorganic matter (DIM), labile dissolved organic matter (DOM), detrital particulate organic matter (POM) and heterotrophic bacteria, with a total of 37 variables (Fig. 2).

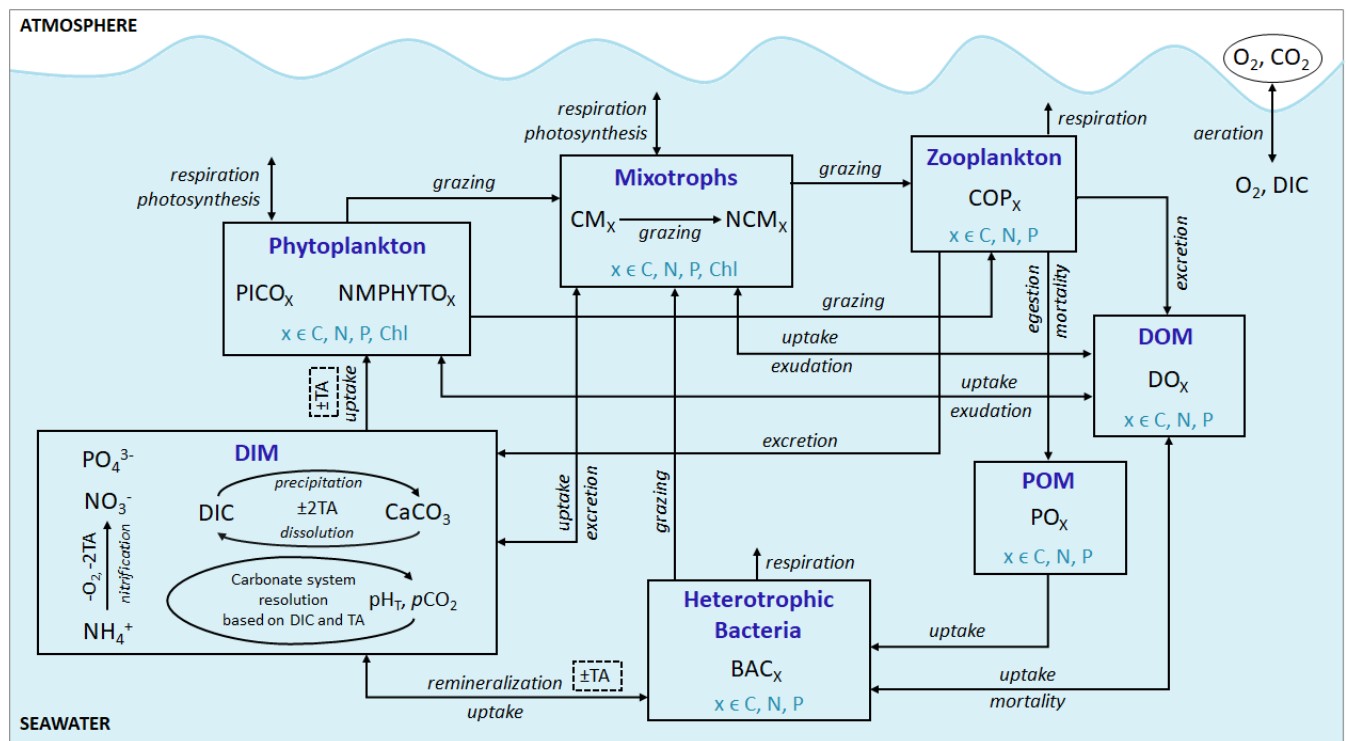

**Figure 2: Schematic representation of the Eco3M_MIX-CarbOx model. Each box represents a model compartment (DIM: dissolved inorganic matter, DOM: labile dissolved organic matter, POM: detrital particulate organic matter). State variables are indicated in black (COP: copepods, PICO: picophytoplankton, NMPHYTO: nano+micro-phytoplankton, $O_2$: dissolved oxygen, $CO_2$: dissolved carbon dioxide, DIC: dissolved inorganic carbon, TA: total alkalinity, $pCO_2$: partial pressure of $CO_2$, $CaCO_3$: calcium carbonate). Elements for which a state variable is expressed with a variable stoichiometry are shown in blue (C: carbon, N: nitrogen, P: phosphorus and, Chl: chlorophyll). Arrows represent processes between two state variables.**

### 2.2.1 Zooplankton

The zooplankton compartment represents copepod-type zooplankton (COP, organisms larger than 200 µm, Fig. 3) whose biomass depends on prey ingestion, respiration, excretion, egestion (faecal pellets), and predation by higher trophic levels. Copepod prey ingestion is represented using the formulation by Auger et al. (2011). Copepods ingest smaller prey and

grazing rates depend on prey type preference as well as on temperature and light due to their effect on prey abundance. Copepods feed with decreasing preference on NCM, nano+micro-phytoplankton (NMPHYTO), and CM (Verity and

145 Paffenhofer, 1996) and release ammonium ($NH_4^+$), phosphate ($PO_4^{3-}$), and dissolved organic carbon (DOC) through excretion, contributing to the POM compartment through egestion and mortality. Mortality due to predation by higher trophic levels represents a closure term (Fig. 2).

### 2.2.2 Phytoplankton

We considered two types of phytoplankton based on size (Fig. 3): picophytoplankton (PICO) and nano+micro-phytoplankton

(NMPHYTO). PICO includes autotrophic prokaryotic organisms such as *Prochlorococcus spp*. and S*ynechococcus spp* which are ubiquitous in the Mediterranean (Mella-flores et al., 2011). NMPHYTO aims to represent phytoplankton larger than 2 µm and smaller than 200 µm. It mainly includes diatoms and autotrophic nanoflagellates. As diatoms are an important component of Mediterranean spring blooms (Margalef, 1978, Leblanc et al., 2018) and cover wide size-range, we decided to consider them as representative of the NMPHYTO.

Both the NMPHYTO and PICO biomass are affected by photosynthesis, respiration, nutrient uptake, exudation, and grazing. Photosynthesis depends on light, nutrients, and temperature (based on Geider et al. (1998) formulation). Respiration depends on photosynthesis (a constant fraction of photosynthetically produced carbon) and nutrient uptake. Nutrient uptake is temperature dependent. NMPHYTO and PICO both consume nitrate ($NO_3^-$), $NH_4^+$, and $PO_4^{3-}$ while PICO also consumes dissolved organic nitrogen (DON) and dissolved organic phosphorus (DOP) (Duhamel et al., 2018). The uptake of DON and

DOP depends on temperature and the cell's nutritional state. If the cell is replete in N (P), then DON (DOP) uptake is null. Both phytoplankton groups exude DOC, DON, and DOP proportionally to their internal content in carbon (C), nitrogen (N) and phosphorus (P) (Fig. 2).

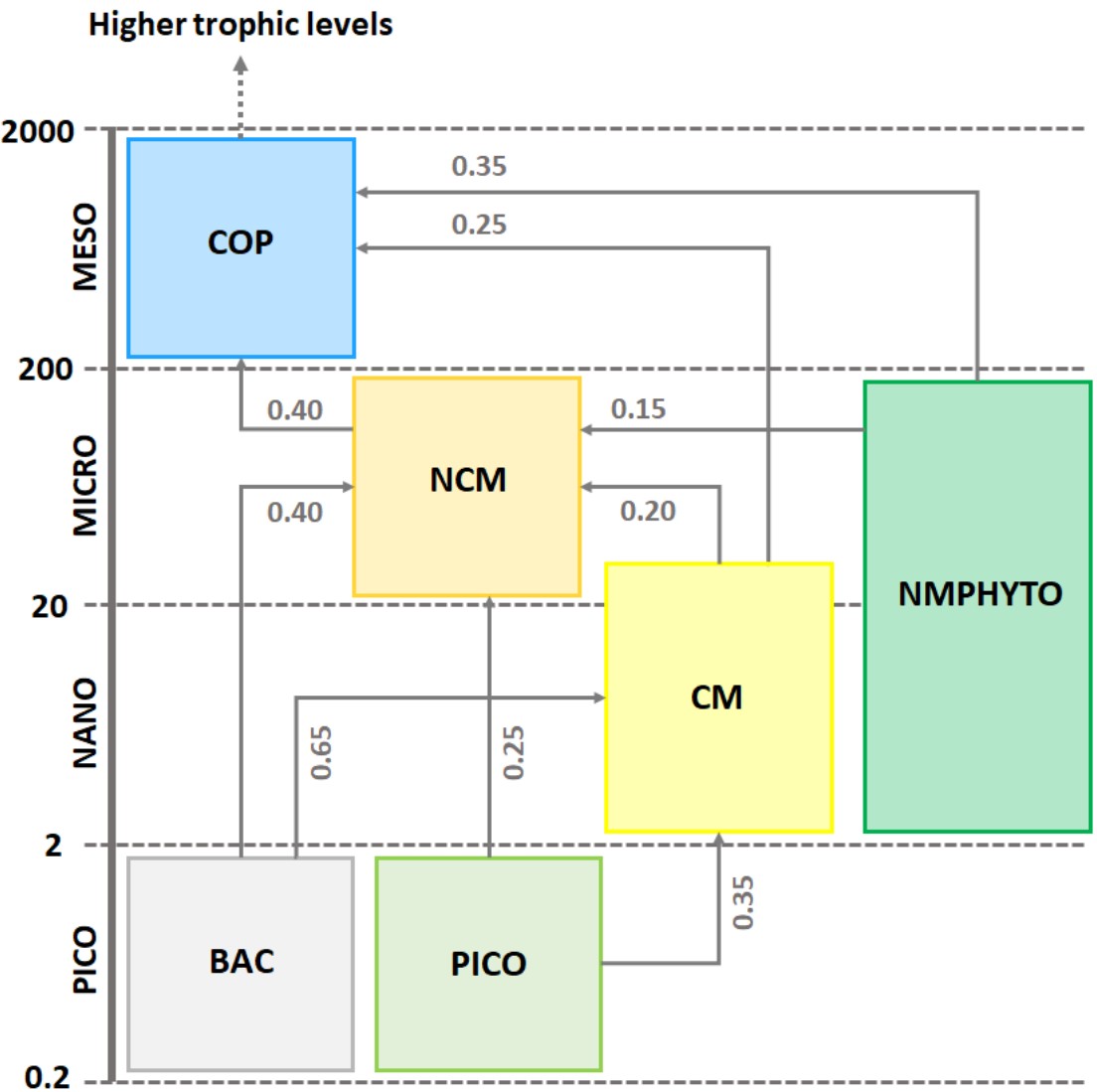

**Figure 3: Repartition of modelled organisms (COP: copepods, PICO: picophytoplankton, NMPHYTO: nano+micro-phytoplankton, and BAC: heterotrophic bacteria) in size classes and trophic interactions between them. Preference values are indicated in grey for copepods (Verity and Paffenhofer, 1996) and NCM (Epstein, 1992; Price & Turner, 1992 ; Christaki, 2009) and CM (Christaki et al., 2002 ; Zubkhov & Tarron, 2008, Millette et al., 2017 ; Livanou et al., 2019).**

### 2.2.3 Heterotrophic bacteria

Heterotrophic bacterial biomass results from balancing growth/losses due to bacterial production, respiration, nutrient uptake, remineralization, predators grazing and natural mortality (Kirchman, 2000 ; Faure et al., 2006). Bacterial production depends on DOC and particulate organic carbon (POC) and is limited by temperature and substrate availability. Heterotrophic bacteria consume particulate organic nitrogen (PON), particulate organic phosphorus (POP), DON, DOP,

$NH_4^+$, and $PO_4^{3-}$ which they remineralize to $NH_4^+$ and $PO_4^{3-}$. They contribute to the DOM pool through natural mortality which depends on temperature (Fig. 2).

### 2.2.4 Dissolved inorganic matter

The DIM compartment consists of the nutrients $NO_3^-$, $NH_4^+$, and $PO_4^{3-}$ as well as dissolved oxygen ($O_2$) and the carbonate system variables (total alkalinity: TA, dissolved inorganic carbon: DIC, $pH_T$, partial pressure of $CO_2$ : $pCO_2$, and calcium carbonate: $CaCO_3$). Nutrient concentrations are affected by heterotrophic bacterial remineralization, uptake, and excretion of organisms ($NH_4^+$ and $PO_4^{3-}$ only), and nitrification ($NO_3^-$ and $NH_4^+$ only). Nitrification (i.e., $NO_3^-$ production from $NH_4^+$) is temperature and $O_2$ dependent. $O_2$ concentration is calculated from photosynthesis, respiration, nitrification, and air-sea exchanges. The other variables included in the DIM compartment are the carbonate system variables (see Barré et al., 2023b for details).

### 2.2.5 Particulate and dissolved organic matter

In Eco3M_MIX-CarbOx, we only considered detrital POM and labile DOM. The POM and DOM compartments are affected by zooplankton, mixotrophs, phytoplankton and heterotrophic bacteria (see above and Fig. 2).

The state equations, process formulations, and associated parameters values for all compartments can be found in Appendices B to E.

### 2.3 Implementation and assessment of mixotrophs

Mixotrophy is defined as the ability of an organism to combine photoautotrophic and heterotrophic modes of nutrition (Riemann et al., 1995). While this implies that several types of mixotrophy exist in the ocean, we focused on a specific type of mixotrophy, namely the capability of a single-celled organism to employ photo- and phagotrophy. Based on Stoecker's (1998) classification, we included two types of mixotrophs in the model: a type IIIB non-constitutive mixotroph (NCM) and a type IIA constitutive mixotroph (CM).

### 2.3.1 Implementation of NCM

NCM (type IIIB) are defined as photosynthetic protozoa, i.e., they are primarily phagotrophic, but can complement their carbon uptake through photosynthesis (Stoecker, 1998). In Eco3M_MIX-CarbOx the NCM are based on ciliates and belong to microplankton (Esteban et al. 2010, Fig. 3). Their dynamics are governed by the following set of balance equations (see Appendix C for a more detailed description of each term).

$$\frac{\partial NCM_C}{\partial t} = \sum_{i=1}^{2}\left(Gra_{NCM_C}^{PHY_{C_i}}\right) + Gra_{NCM_C}^{CM_C} + Gra_{NCM_C}^{BAC_C} + Photo_{NCM_C}^{DIC} - Resp_{NCM_C}^{DIC} - Exu_{NCM_C}^{DOC} - Gra_{NCM_C}^{COP_C}$$

$$\frac{\partial NCM_N}{\partial t} = \sum_{i=1}^{2}\left(Gra_{NCM_N}^{PHY_{N_i}}\right) + Gra_{NCM_N}^{CM_N} + Gra_{NCM_N}^{BAC_N} - Exu_{NCM_N}^{DON} - Excr_{NCM_N}^{NH_4} - Gra_{NCM_N}^{COP_N}$$

$$\frac{\partial NCM_P}{\partial t} = \sum_{i=1}^{2}\left(Gra_{NCM_P}^{PHY_{P_i}}\right) + Gra_{NCM_P}^{CM_P} + Gra_{NCM_P}^{BAC_P} - Exu_{NCM_P}^{DOP} - Excr_{NCM_P}^{PO_4} - Gra_{NCM_P}^{COP_P}$$

$$\frac{\partial NCM_{CHL}}{\partial t} = \sum_{i=1}^{2}\left(Gra_{NCM_{Chl}}^{PHY_{Chl_i}}\right) + Gra_{NCM_{Chl}}^{CM_{Chl}} - Degrad_{NCM_{Chl}} - Gra_{NCM_{Chl}}^{COP_C}, \tag{1}$$

Being primarily phagotrophic, NCM grazing is implemented in a similar way to zooplankton grazing in that they can ingest preferentially smaller prey items while having certain preferences for different prey types. From most to least preferred prey, NCM feed on heterotrophic bacteria, picophytoplankton, CM and nano+micro-phytoplankton (Epstein, 1992 ; Price & Turner, 1992 ; Christaki, 1999). By ingesting photosynthetic prey, NCM acquire the capacity to photosynthesize by temporarily sequestering chloroplasts (Putt, 1990). This process is modelled as a grazing flux between the chlorophyll concentrations of photosynthetic prey and NCM (Eq. 2). The NCM capacity to photosynthesize degrades over time unless fresh chloroplasts are sequestered (Eq. 3, based on Leles et al., 2018).

$$Gra_{NCM_{Chl}}^{PREY_{Chl}} = G_{MAX} * \frac{(\Phi * PREY_C^2)}{K_{NCM} * \sum_{i=1}^{4}\left(\Phi_i * PREY_{C_i}\right) + \sum_{i=1}^{4}\left(\Phi_i * PREY_{C_i}^2\right)} * NCM_C * \frac{PREY_{Chl}}{PREY_C}, \tag{2}$$

$$Degrad_{NCM_{Chl}} = \left(\left(Gra_{NCM_{Chl}}^{PREY_{Chl}} * dt\right) + NCM_{Chl}\right) * k_{MORT,Chl}, \tag{3}$$

where PREY $\epsilon$ [CM, NMPHYTO, PICO], $G_{MAX}$, $K_{NCM}$, $\Phi$ and $k_{MORT,Chl}$ represent the maximum grazing rate, the grazing half saturation constant, the NCM preference for a specific prey type, and the loss rate of captured photosystems, respectively (see appendix E for details). $NCM_X$ and $PREY_X$ are the NCM and PREY concentrations of element X, respectively. $Gra_{NCM_{Chl}}^{PREY_{Chl}}$ and $Degrad_{NCM_{Chl}}$ are in mmol m$^{-3}$ s$^{-1}$.

As NCM photosynthesis depends on the sequestered chloroplasts from prey, we created a prey dependent formulation to represent it (Eq. 4). We based our formulation on Geider et al. (1998) which provide a photosynthesis flux nutrient, temperature, and light dependant. In this formulation, a maximum photosynthetic rate is first calculated ($P_{MAX}^C$) based on the C-specific photosynthetic rate at a reference temperature of the photosynthetic organism ($P_{REF}^C$). This rate is nutrient and temperature dependant and is next multiplied by light limitation function. We applied parameters of the prey except for the nutrient limitation which is calculated based on NCM internal content in N and P as the process takes place inside the NCM cells albeit using the prey's chloroplasts.

$$P_{MAX,NCM}^C = P_{REF,PREY}^C * f_{PREY}^T * f_{Q,NCM}^G$$

$$Photo_{NCM_C,PREY_C}^{DIC} = P_{MAX,NCM}^C * limI_{PREY} * NCM_C, \tag{4}$$

where PREY $\epsilon$ [CM, NMPHYTO, PICO], $P_{MAX}^C$ is the maximum photosynthetic rate in s$^{-1}$, and $Photo_{NCM_C,PREY_C}^{DIC}$ is the NCM photosynthetic flux associated to the chloroplast from the considered prey in mmol m$^{-3}$ s$^{-1}$. $P_{REF}^C$ is the C-specific photosynthetic rate at a reference temperature (see Appendix E for values for each prey). $f^T$, and limI are temperature and light limitation functions respectively (see Appendix C for detailed formulations). $f_Q^G$ is a nutrient limitation function which express the nutritional state of the cell and is based on X (X $\epsilon$ [N, P]) to C ratio (i.e., $NCM_X$ to $NCM_C$ in this case).

$\quad f_Q^G = \min \left( \frac{Q_C^N - Q_{C,min}^N}{Q_{C,max}^N - Q_{C,min}^N}, \frac{Q_C^P - Q_{C,min}^P}{Q_{C,max}^P - Q_{C,min}^P} \right),$ (5)

$f_Q^G$ is dimensionless. $Q_{c,min}^N$, $Q_{c,min}^P$, $Q_{c,max}^N$, and $Q_{c,max}^P$ represent the minima and maxima of the X to C ratios (see appendix E for values used for NCM). When the cellular C content is high relative to other elements, then $f_Q^G$ value approaches 0 and vice versa.

The photosynthetic fluxes from each prey type were weighted by NCM prey preference and summed according to:

$\quad Photo_{NCM_C}^{DIC} = \sum_{i=1}^{3} \left( \Phi * Photo_{NCM_C, PREY_{Ci}}^{DIC} \right),$ (6)

Where PREY $\epsilon$ [CM, NMPHYTO, PICO], $Photo_{NCM_C}^{DIC}$ is the NCM photosynthetic flux in mmol m$^{-3}$ s$^{-1}$, $\Phi$ is the NCM prey type preference (values in appendix E).

Finally, respiration, exudation, and excretion are based on grazing fluxes and nutrient limitations. Grazed C is consumed through respiration and excess C is exuded as DOC. The amount of respired or exuded C is determined by the cell's

nutritional state. Respiration and exudation fluxes are high when NCM C content is high relative to N or P and vice-versa. We used the same reasoning for grazed N (P) which is exuded as DON (DOP) or excreted as NH$_4^+$ (PO$_4^{3-}$) when NCM N (P) content is high (see Appendix C for details).

### 2.3.2 Implementation of CM

CM (type IIA) are defined as phagotrophic algae i.e., they are primarily phototrophic, but can ingest prey to obtain limiting

nutrients (Stoecker, 1998). CM are based on dinoflagellates which belong mainly to nanoplankton but can also be found in microplankton (Stoecker, 1999, Fig. 3). Their dynamics are governed by the following set of balance equations (see Appendix C for details).

$\frac{\partial CM_C}{\partial t} = Gra_{CM_C}^{PICO_C} + Gra_{CM_C}^{BAC_C} + Photo_{CM_C}^{DIC} - Resp_{CM_C}^{DIC} - Exu_{CM_C}^{DOC} - Gra_{CM_C}^{NCM_C} - Gra_{CM_C}^{COP_C}$

$\frac{\partial CM_N}{\partial t} = Gra_{CM_N}^{PICO_N} + Gra_{CM_N}^{BAC_N} + Upt_{CM_N}^{NO_3} + Upt_{CM_N}^{NH_4} + Upt_{CM_N}^{DON} - Exu_{CM_N}^{DON} - Gra_{CM_N}^{NCM_N} - Gra_{CM_N}^{COP_N}$

$\frac{\partial CM_P}{\partial t} = Gra_{CM_P}^{PICO_P} + Gra_{CM_P}^{BAC_P} + Upt_{CM_P}^{PO_4} + Upt_{CM_P}^{DOP} - Exu_{CM_P}^{DOP} - Gra_{CM_P}^{NCM_P} - Gra_{CM_P}^{COP_P}$

$\frac{\partial CM_{CHL}}{\partial t} = Syn_{CM_{Chl}} - Gra_{CM_{Chl}}^{NCM_{Chl}} - Gra_{CM_{Chl}}^{COP_C},$ (7)

CM photosynthesis is temperature, light and nutrient dependent following Geider et al. (1998):

$P_{MAX,CM}^C = P_{REF,CM}^C * f_{CM}^T * f_{Q,CM}^G$

$Photo_{CM_C}^{DIC} = P_{MAX,CM}^C * limI_{CM} * CM_C,$ (8)

where $P_{MAX}^C$ is the maximum photosynthetic rate in s$^{-1}$, $Photo_{CM_C}^{DIC}$ is the CM photosynthetic flux in mmol m$^{-3}$ s$^{-1}$, $P_{REF}^C$ is the C-specific photosynthetic rate at a reference temperature (see Appendix E for CM value). $f^T$, $f_Q^G$ and limI are temperature, nutrient and light limitation functions respectively (see Appendix C for detailed formulations of $f^T$ and limI, and Eq. 5 for the formulation of $f_Q^G$).

Like picophytoplankton, CM assimilate dissolved inorganic nutrients ($NO_3^-$, $NH_4^+$, and $PO_4^{3-}$) and DOM (DON and DOP).

Uptake fluxes are calculated by using a Michaelis-Menten equation and are limited by temperature. DOM uptake also depends on the nutritional state of the cell in that the higher cell's N (P) content the lower the DON (DOP) uptake.

When DIN and/or DIP is limiting the growth, CM can ingest smaller prey to supplement their N and/or P needs (Stoecker, 1997). CM feed on heterotrophic bacteria (preferred) and picophytoplankton (less preferred, Christaki et al., 2002 ; Zubkhov & Tarron, 2008, Millette et al., 2017 ; Livanou et al., 2019) and the same grazing formulation as for zooplankton and NCM

is used except that CM grazing is limited by DIN (DIP) concentration and light (Stoecker, 1997, 1998; Eq. 9).

$$Gra_{CM_C}^{PREY_C} = G_{MAX} * \frac{\Phi * PREY_C}{K_{CM} * \sum_{i=1}^{2}\left(\Phi_i * PREY_{C_i}\right) + \sum_{i=1}^{2}\left(\Phi_i * PREY_{C_i}^2\right)} * CM_C * f_{I,inhib}^{CM} * f_{NUT,inhib}^{CM} * \left(1 - f_{Q,CM}^G\right)$$

$$f_{I,inhib}^{CM} = 1 - \exp\left(\frac{-\alpha_{Chl} * Q_C^{Chl} * E_{PAR}}{P_{REF}^C}\right)$$

$$f_{NUT,inhib}^{CM} = \min\left(1 - \max\left(\frac{[NO_3^-]}{K_{NO_3^-} + [NO_3^-]}, \frac{[NH_4^+]}{K_{NH_4^+} + [NH_4^+]}\right), \frac{[PO_4^{3-}]}{K_{PO_4^{3-}} + [PO_4^{3-}]}\right),$$ (9)

where PREY $\epsilon$ [BAC, PICO], $Gra_{CM_C}^{PREY_C}$ is in mmol m$^{-3}$ s$^{-1}$. $f_{I,inhib}^{CM}$ and $f_{NUT,inhib}^{CM}$ are the (dimensionless) inhibitions of

270 grazing by light and nutrients, respectively. $G_{MAX}$, $K_{CM}$, $\Phi$, $\alpha_{Chl}$, $P_{REF}^C$, and $K_{NUT}$ represent the maximum grazing rate, the grazing half saturation constant, the CM prey preference, the chlorophyll-specific light absorption coefficient, the C-specific photosynthesis rate at a reference temperature, and the half saturation constant for the considered nutrient ($NO_3^-$, $NH_4^+$ or $PO_4^{3-}$), respectively (values in Appendix E). $Q_C^{Chl}$ is the chlorophyll-to-carbon ratio and $E_{PAR}$ the irradiance value. The grazing is also affected by CM internal content in N and P ($f_Q^G$ term, Eq. 5).

CM ingest prey to supplement their needs in N and P only, exuding grazed C as DOC (Stoecker, 1998; Eq.10). Hence, DOC is released through two metabolic pathways exudation of carbon acquired via : (i) photosynthesis, and (ii) grazing.

$$Exu_{CM_C}^{DOC} = \left(1 - frac_{resp}\right) * \left(Photo_{CM_C}^{DIC} * \left(1 - f_{Q,CM}^G\right)\right) + \sum_{i=1}^{2}\left(Gra_{CM_C}^{PREY_{C_i}}\right),$$ (10)

where PREY $\epsilon$ [BAC, PICO], $Exu_{CM_C}^{DOC}$ is in mmol m$^{-3}$ s$^{-1}$. $Photo_{CM_C}^{DIC}$ is the photosynthetic flux in mmol m$^{-3}$ s$^{-1}$ (Eq. 8) and $Gra_{CM_C}^{PREY_C}$ is the grazing flux for the considered prey in mmol m$^{-3}$ s$^{-1}$ (Eq. 9). $frac_{resp}$ represents the fraction of respired carbon

from photosynthesis (values and units in Appendix E).

The formulations for DON and DOP exudation are similar. Exudation only occurs on the N and P obtained from nutrient uptake. In other words, neither N nor P obtained from grazing are released through exudation. When DIN (DIP) concentration is limiting CM will ingest prey in addition to the uptake of nutrient. As their internal content in N (P) is particularly low, exudation of DON (DOP) is not allowed (equal to 0). When DIN (DIP) concentration is high, CM only

285 perform nutrient uptake (no grazing as it only supplements N and P needs in limiting conditions). Then, all the N (P) from uptake is exuded as the cell is already loaded in N (P) and as no grazing is performed, no N (P) from grazing is exuded in these conditions. Respiration uses the same formulation as phytoplankton i.e., a constant fraction of photosynthesis and nutrient uptake is respired (Section 2.2.2 and Appendix C).

**2.4 Designing numerical experiments**

**2.4.1 Assessment of mixotrophs**

To be considered as correctly represented by the model, NCM and CM must verify the properties listed in Table 2. These properties have been stated by Stoecker (1998) to provide conceptual models to represent the different types of mixotrophs.

**Table 2: Summary of NCM and CM properties based on Stoecker (1998). DIN represents the sum of $NO_3^-$ and $NH_4^+$ and DIP represents $PO_4^{3-}$. Food is represented by preys concentration.**

| NCM properties (Type IIIB, Stoecker, 1998) | |
|---|---|
| **Property number** | **Property description** |
| **NCMP1** | Grazing and DIN (DIP) concentration are independent |
| **NCMP2** | Photosynthesis and DIN (DIP) concentration are independent |
| **NCMP3** | Grazing and irradiance are independent |
| **NCMP4** | Photosynthesis increases when food concentration increases |
| **CM properties (Type IIA, Stoecker, 1998)** | |
| **Property number** | **Property description** |
| **CMP1** | Photosynthesis increases when food concentration increases |
| **CMP2** | Photosynthesis increases when DIN (DIP) concentration increases |
| **CMP3** | Grazing decreases when DIN (DIP) concentration increases |
| **CMP4** | Grazing increases when irradiance increases |

To verify these properties, we designed several numerical experiments (Table 3 and 4) in which we modify one of the following features: prey biomass, DIN and DIP concentrations or irradiance. We first ran a reference simulation (referred as Replete in Table 3) in which we set all the previous features to a maximum value during the entire simulation. Maximum prey biomass was obtained by multiply the initial condition by 2 (sum of the initial carbon prey biomass multiply by 2),

maximum DIN and DIP concentrations were chosen based on high values observed at SOLEMIO (Pujo-Pay et al., 2011) and maximum irradiance correspond to the mean value of simulated irradiance for the SOLEMIO station by the meteorological model WRF (Yohia, 2017). Next, we ran low nutrients (low nutrients values observed at SOLEMIO multiply by 0.1, low-nut simulation in Table 3 for NCM and CM), low prey concentration (maximum prey concentration multiplied by 0.5, low-food simulation in Table 3 for NCM and CM) and low light (maximum value multiplied by 0.05, low-light simulation in Table 3

for CM only). For NCM, to verify the light dependant property (NCMP3), it is also necessary to set the NCM concentration to a constant during the entire simulation, we performed another reference simulation and a low-light simulation in which NCM concentration is constant (initial condition, NCM replete with constant and NCM low light with constant in Table 4, respectively). Finally, we compare the simulations to their associated reference simulation.

**Table 3: Summary of the simulations performed to check NCM and CM properties (excluding NCMP3). For NCM, [PREY] stands for the sum of CM, nano+micro-phytoplankton, picophytoplankton and heterotrophic bacterial biomasses. For CM, [PREY] stand for the sum of picophytoplankton and heterotrophic bacterial biomasses.**

| NCM properties (Type IIIB, Stoecker, 1998) | | | | | |
|---|---|---|---|---|---|
| Simulation name | [PREY] (mmol C m$^{-3}$) | [DIN] (mmol N m$^{-3}$) | [DIP] (mmol P m$^{-3}$) | Irradiance (W m$^{-2}$) | Tested property |
| NCM Replete | 1.5 | 1.5 | 0.09 | 120 | Reference simulation |
| NCM Low-Nut | 1.5 | $7.5 \times 10^{-3}$ | $4.5 \times 10^{-4}$ | 120 | NCMP1 and NCMP2 |
| NCM Low-Food | 0.75 | 1.5 | 0.09 | 120 | NCMP4 |
| CM properties (Type IIA, Stoecker, 1998) | | | | | |
| Simulation name | [PREY] (mmol C m$^{-3}$) | [DIN] (mmol N m$^{-3}$) | [DIP] (mmol P m$^{-3}$) | Irradiance (W m$^{-2}$) | Tested property |
| CM Replete | 0.92 | 1.5 | 0.09 | 120 | Reference simulation |
| CM Low-Nut | 0.92 | $7.5 \times 10^{-3}$ | $4.5 \times 10^{-4}$ | 120 | CMP2 and CMP3 |
| CM Low-Light | 0.92 | 1.5 | 0.09 | 3 | CMP4 |
| CM Low-Food | 0.46 | 1.5 | 0.09 | 120 | CMP1 |

**Table 4: Summary of the simulations performed to NCMP3. Prey stands for the sum of CM, nano+micro-phytoplankton, picophytoplankton and heterotrophic bacterial biomasses.**

| Simulation name | [NCM] (mmol C m$^{-3}$) | [PREY] (mmol C m$^{-3}$) | [DIN] (mmol N m$^{-3}$) | [DIP] (mmol P m$^{-3}$) | Irradiance (W m$^{-2}$) | Tested property |
|---|---|---|---|---|---|---|
| NCM Replete with constant | 0.4 | 1.5 | 1.5 | 0.09 | 120 | Reference simulation |
| NCM Low-light with constant | 0.4 | 1.5 | 1.5 | 0.09 | 3 | NCMP3 |

## 2.4.2 Typical vs limited conditions

After verifying mixotrophs properties, we simulated three types of light and nutrient regimes for the BoM: typical, nutrient limited, and light limited (Table 5). With these three regimes, we aim to reproduce typical and limited conditions (i.e., nutrient and light limited) in the BoM. Simulations are run for 2017, at SOLEMIO station. Eco3M_MIX-CarbOx spin-up period is about 3 months. To avoid initial conditions impact on our results, we ran three years of simulation (i.e., repetition of 2017 three times) and we present the results for the second year of simulation.

For the typical scenario, light was modelled using the solar irradiance from the WRF meteorological model for SOLEMIO station (Table 1) and $NO_3^-$, $NH_4^+$, and $PO_4^{3-}$ concentrations were based on in situ observations at SOLEMIO during 2017 (values from SOMLIT) using a linear interpolation between fortnightly data points (Fig. 4). In these conditions, we also performed two simulations without mixotrophs as control simulations. These simulations correspond to two configurations of Eco3M_MIX-CarbOx without mixotrophs: a first one in which mixotrophs and their associated biogeochemical processes are simply deleted (D configuration in Table 5), and a second one in which we replaced mixotrophs by organisms with strict diets (R configuration in Table 5). These configurations and their results are presented in the supplementary material.

In the nutrient limited scenario, the ecosystem is limited by DIN and DIP concentrations only, using values 10 times lower than the minima observed at SOLEMIO, keeping both DIN (sum of $NO_3^-$ and $NH_4^+$, $6.75 \times 10^{-3}$ mmol m$^{-3}$ and $7.5 \times 10^{-4}$ mmol m$^{-3}$, respectively) and DIP constant for the duration of the simulation. The Eco3M_MIX-CarbOx model was initially developed to be run with low nutrient concentrations, representative of the Mediterranean Sea (Morel & Andre, 1991). To ensure that organisms were not limited by light, we multiplied the typical irradiance by 2.

In the light limited scenario, we only applied 5 % of the typical irradiance while DIN ($[NO_3^-] = 1.35$ mmol m$^{-3}$, $[NH_4^+] = 0.15$ mmol m$^{-3}$) and DIP concentrations were set to winter values at SOLEMIO.

For the three simulations, we used typical values of the BoM to represent temperature, salinity, wind speed and atmospheric $pCO_2$ as described in Table 1.

**Table 5: Summary of simulation properties. Configurations without mixotrophs are detailed in the supplementary material.**

| Simulation name | Mixotrophs | [DIN] | [DIP] | Irradiance |
|---|---|---|---|---|
| D configuration | Absent | SOLEMIO interpolation | SOLEMIO interpolation | WRF |
| R configuration | Absent | SOLEMIO interpolation | SOLEMIO interpolation | WRF |
| Typical | Present | SOLEMIO interpolation | SOLEMIO interpolation | WRF |
| Nutrient limited | Present | $7.5 \times 10^{-3}$ mmol N m$^{-3}$ | $4.5 \times 10^{-4}$ mmol P m$^{-3}$ | WRF $\times$ 2 |
| Light limited | Present | 1.5 mmol N m$^{-3}$ | 0.09 mmol P m$^{-3}$ | WRF $\times$ 0.05 |

**2.5 Ecosystem and phytoplankton composition**

We used the total carbon biomass which is calculated by summing daily average biomass of each organism to assess the ecosystem composition and its dynamics during different scenarios over a full year.

We used the total phytoplanktonic carbon biomass which is calculated by summing daily average carbon biomass of each phytoplanktonic organism, to assess the phytoplankton composition (given as percentages of nano+micro-phytoplankton, picophytoplankton and CM). We chose to include CM in phytoplankton composition since they are primarily phototrophic. The phytoplankton composition was examined for the typical scenario (see previous section) over a full year and during three specific events: (i) winter mixing, (ii) Rhône River intrusion, and (iii) Cortiou water intrusion (Fig. 4). Each of these events is associated with a nutrient maximum. The winter mixing event is associated with a peak in $PO_4^{3-}$ on 1 February (Fig. 4a), the Rhône River intrusion with a $NO_3^-$ maximum on 15 March (Fig. 4b), and the intrusion of Cortiou water with a $NH_4^+$ maximum (Fig. 4c). During these events, phytoplankton composition is calculated for a period of 11 days (day of the maximum and $\pm$ 5 days).

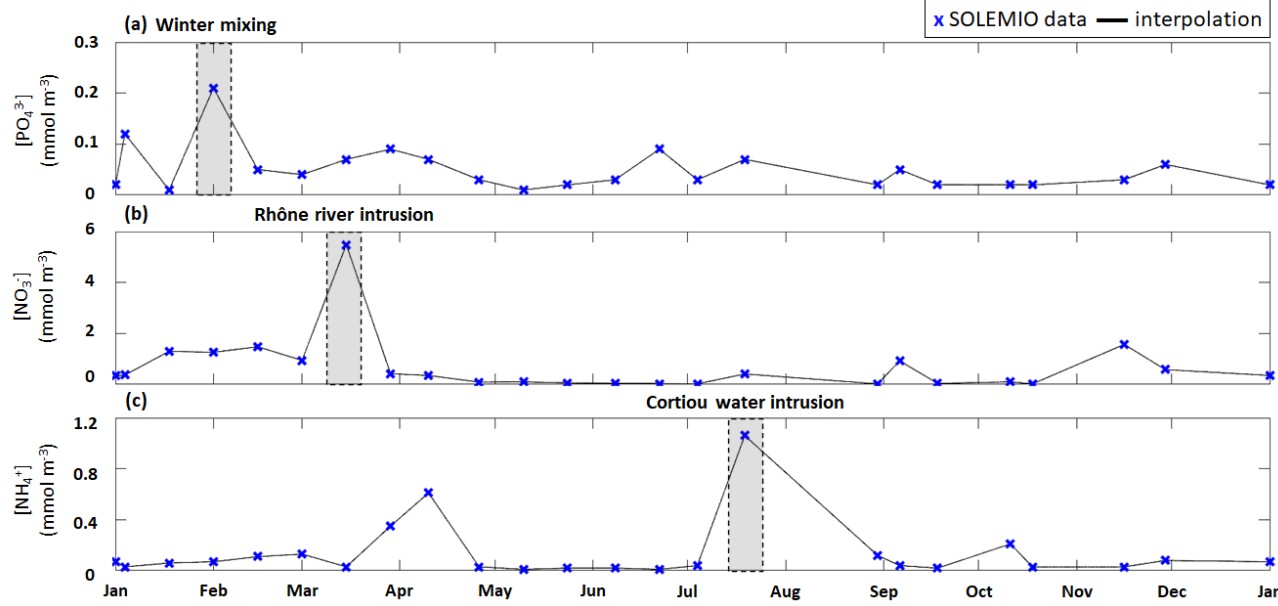

**Figure 4: Time series of interpolated surface (a) $PO_4^{3-}$ concentration, (b) $NO_3^-$ concentration, and (c) $NH_4^+$ concentration (lines) from fortnightly measurements at SOLEMIO data (markers) during 2017. The studied events are shaded in grey.**

## 3 Results

### 3.1 Representation of mixotrophs

To assess whether the mixotrophs were correctly represented in the model we compared the properties emerging during the simulation to those listed in Table 2 for the simulation described in Tables 3 and 4. Here, we present the yearly time-series of daily averaged grazing and photosynthesis fluxes (Fig. 5). Yearly mean values of grazing and photosynthesis for each simulation are presented in appendix F.

  The results show that, throughout the year, NCM grazing fluxes obtained in low and high DIM (DIN + DIP) conditions

remained constant (Fig. 5a) and seem independent of irradiance levels (Fig. 5c). Similarly, NCM photosynthesis in the model does not depend on DIM concentration (Fig. 5b). However, doubling the food led to an increase in NCM photosynthesis (Fig. 5d).

  For the CM the picture is different. CM photosynthesis slightly increases when food concentration increases while increasing DIM concentrations led to significantly increases in photosynthesis (Fig. 5e,f). Also, CM grazing depends on DIM

concentration and light (Fig. 5g,h), although the effect of the latter is less pronounced. Under low DIM concentrations, CM grazing was about one order of magnitude higher than with high DIM concentrations (maxima of $2.3 \times 10^{-7}$ mmol m$^{-3}$ s$^{-1}$ vs $5.0 \times 10^{-8}$ mmol m$^{-3}$ s$^{-1}$) (Fig. 5g). Increasing in light also led to significant increases in grazing (maxima of $5 \times 10^{-8}$ mmol m$^{-3}$ s$^{-1}$ vs $1 \times 10^{-9}$ mmol m$^{-3}$ s$^{-1}$ ; Fig. 5h).

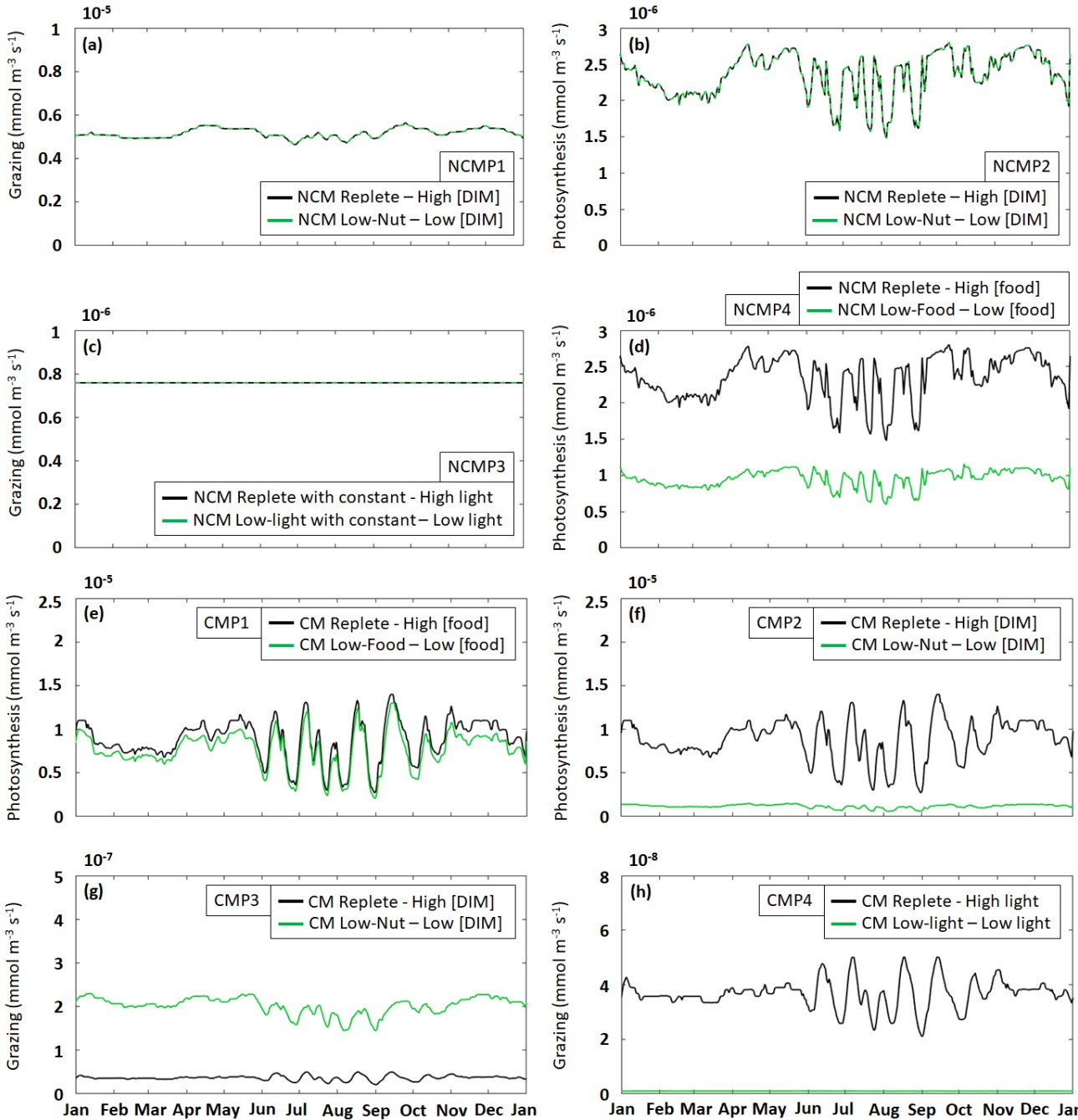

**Figure 5: Assessment of mixotrophs representation in the model. Each frame represents the test of a property stated by Stoecker (1998) for NCM Type IIIB: grazing is independent of (a) DIM concentration (NCMP1), (c) irradiance (NCMP3), and photosynthesis (b) is independent of DIM concentration (NCMP2), (d) increases with food concentration (NCMP4) ; and CM Type IIA: photosynthesis increases with (e) food concentration (CMP1), (f) DIM concentration (CMP2), and grazing (g) decreases when DIM concentration increases (CMP3), (h) increases with irradiance (CMP4). Properties are detailed in Table 2 and associated**

 simulations are detailed in Tables 3 and 4. Plotted values represent daily averaged grazing and photosynthesis. DIM: dissolved inorganic matter (sum of dissolved inorganic nitrogen and dissolved inorganic phosphorus).

## 3.2 Phytoplankton composition under typical forcing conditions and during specific events

We studied the phytoplankton composition throughout the entire year of 2017 (Fig. 6a) and during specific events, namely winter mixing event (Fig. 6b), a Rhône River intrusion (Fig. 6c), and a Cortiou water intrusion (Fig. 6d). The formulations 380 used to describe the limitation status are presented in Appendix D.

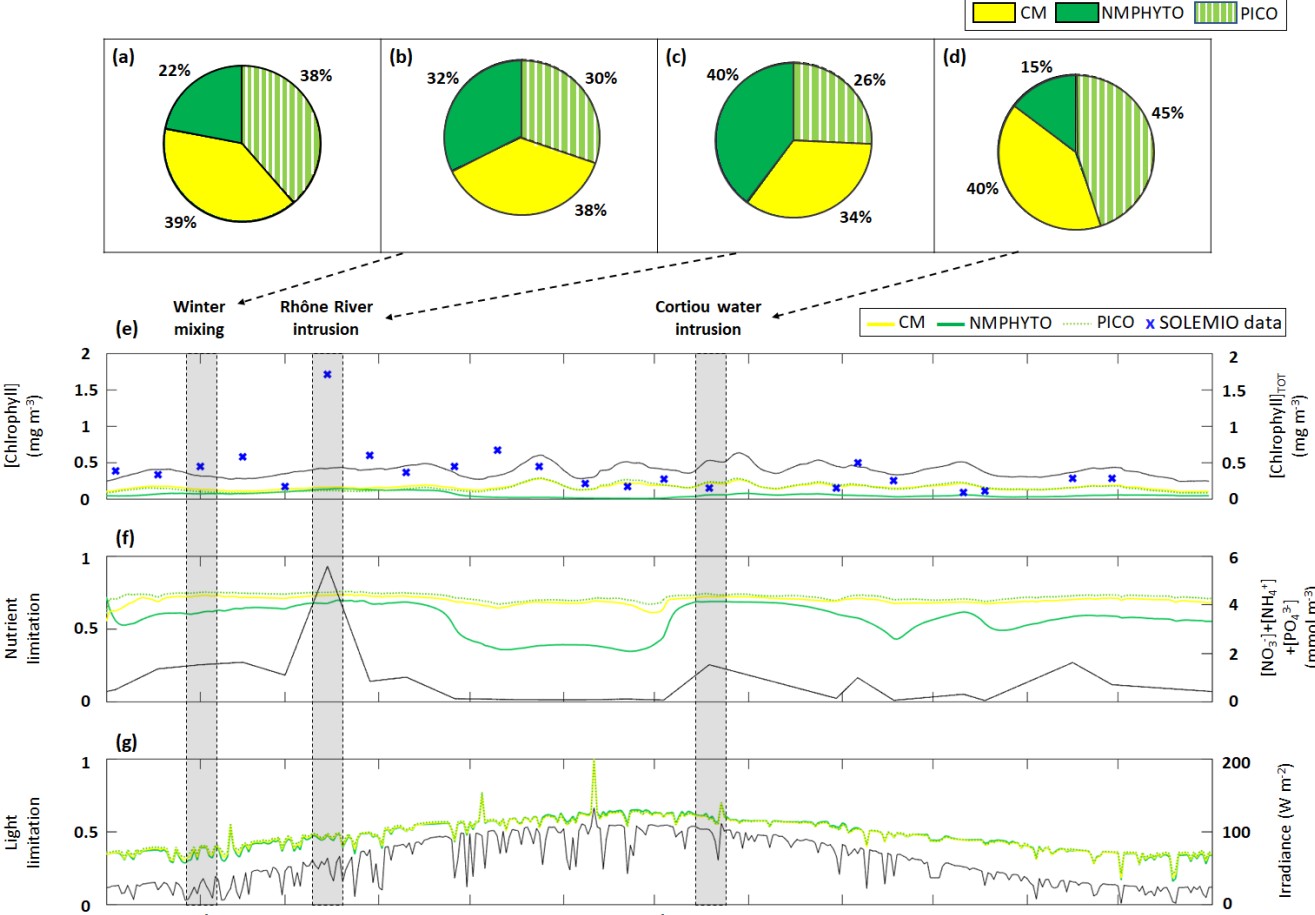

Figure 6: Phytoplankton composition as percentages of C biomass during (a) 2017, (b) a winter mixing event, (c) a Rhône River intrusion, and (d) a Cortiou water intrusion. Time series of daily averages of the three phytoplankton groups: (e) chlorophyll concentrations, (f) nutrient limitation status, and (g) light limitation status (a value of 1 means no limitation). The black line in 385 each panel show (e) total chlorophyll concentration (sum of daily average CM, NMPHYTO and PICO chlorophyll concentrations), (f) sum of nutrients ($[NO_3^-]+[NH_4^+]+[PO_4^{3-}]$), and (g) daily average irradiance, with the corresponding axes shown on the right. The markers in (f) represented in situ SOLEMIO data. Sections shaded in grey show when the three events occurred in time.

### 3.2.1 Annual scale

Through the year of 2017, phytoplankton biomass was dominated by CM, closely followed by PICO and at some distance by NMPHYTO (Fig 6a).

CM and PICO chlorophyll concentrations show similar patterns with values varying between 0.1 (on 18 February) and 0.3 mg Chl m$^{-3}$ (on 24 May). The highest variability occurred between May and October. NMPHYTO chlorophyll concentrations varied between 0.01 (on 25 June) and 0.16 mg Chl m$^{-3}$ (on 20 March), with the lowest values occurring

between May and July (Fig. 6e). The in situ values reached a maximum of 1.71 mg Chl m$^{-3}$ on 15 March, linked to the Rhône River intrusion event. Between June and November, in situ values were generally lower compared to the other months and a minimum of 0.1 mg Chl m$^{-3}$ was reached on 11 October. The modelled chlorophyll concentration shows less variations than the in situ data, especially since the model was unable to reproduce the maximum related to the Rhône intrusion on 15 March nor the minimum on 11 October. Nevertheless, the modelled values, ranging from 0.25 and 0.64 mg

Chl m$^{-3}$, are generally of the same order of magnitude as in situ observation. Both the model results and in situ data yielded the same mean chlorophyll concentrations of 0.4 mg Chl m$^{-3}$.

Total nutrients (Fig. 6f) varied between 0.08 mmol m$^{-3}$ (in summer and autumn) and 5.6 mmol m$^{-3}$ (reached on 15 March). CM and PICO nutrient limitation status remained fairly stable near the mean value of 0.71, however, organisms are more limited in late spring and summer (between May and July). NMPHYTO nutrient limitation status is more variable, showing

higher limitations in late spring and summer (between late April and July) and lesser limitation in early spring and late summer.

The light limitation status clearly reflects the diurnal and seasonal variations in incident irradiance (Fig. 6g). Throughout the year, all the three phytoplankton groups show nearly identical levels of limitation.

### 3.2.2 Winter mixing event

During the winter mixing event, a PO$_4^{3-}$ maximum was recorded at SOLEMIO station (0.21 mmol m$^{-3}$, Fig. 4a). In terms of C biomass, CM was most dominant, followed by NMPHYTO and PICO (Fig. 6b).

CM and PICO chlorophyll decreased slightly, while NMPHYTO chlorophyll remained constant (Fig. 6e). The decrease in CM and PICO chlorophyll is also visible in the total chlorophyll which dropped from 0.41 mg Chl m$^{-3}$ to 0.28 mg Chl m$^{-3}$ (Fig. 5e).

The nutrient limitation remained fairly stable for all phytoplankton groups (Fig 6.f).

During the event, irradiance was low (< 40 W m$^{-2}$, Fig. 6g) and decreased at the end of January due to bad weather. CM, NMPHYTO and PICO light limitation status remained similar throughout this event and at a relatively low value (0.3).

### 3.2.3 Rhône River intrusion

The Rhône River intrusion resulted in a $NO_3^-$ maximum at SOLEMIO station (5.48 mmol m$^{-3}$, Fig. 4b). Model results
indicate that during the event, phytoplankton was dominated by NMPHYTO followed by CM and PICO (Fig. 6c).

All three chlorophyll concentrations increased with the most significant increase occurring for NMPHYTO (from 0.11 to 0.15 mg Chl m$^{-3}$) which surpassed PICO at the beginning of the event (Fig. 6e).

The intrusion also led to a significant increase in modelled total nutrients (reaching 5.5 mmol m$^{-3}$). Nutrient limitation status was similar for all groups and remained between 0.67 and 0.75, showing no significant variations during the event (Fig. 5f).
While irradiance levels were moderate (around 60 W m$^{-2}$) all the three groups were still light limited (values of about 0.5, Fig. 6g).

### 3.2.4 Cortiou water intrusion

During the Cortiou water intrusion, in situ $NH_4^+$ concentration reached a maximum of 1.06 mmol m$^{-3}$ (Fig. 4c) at SOLEMIO station. In the model, phytoplankton composition was dominated by PICO and CM with NMPHYTO a distant third (Fig.
6d).

Chlorophyll increased in all groups resulting in an increase of total chlorophyll from 0.36 to 0.52 mg Chl m$^{-3}$ (Fig. 6e).

During the event, the sum of nutrients reached 1.53 mmol m$^{-3}$ with a clear $NH_4^+$ maximum. Nutrient limitation status was similar across groups and remained stable around 0.7 (Fig. 6f).

Irradiance levels were moderate (between 70 and 112 W m$^{-2}$) leading only to slight light limitation (values between 0.58 and
0.62, Fig. 6g).

## 3.3 Ecosystem composition under light and nutrient limitation

### 3.3.1 Nutrient limited conditions

In nutrient limited conditions, the modelled yearly total C biomass i.e., sum of daily C biomass of each organism, was 349.5 mmol C m$^{-3}$, divided between copepods (148.7 mmol C m$^{-3}$), NCM (129.8 mmol C m$^{-3}$) and heterotrophic bacteria (26.2
440   mmol C m$^{-3}$), followed by the three phytoplankton groups, of which NMPHYTO had the lowest biomass (4.5 mmol C m$^{-3}$, Fig. 7a).

Copepods and NCM dominated the ecosystem with copepods being more abundant between October to June, while NCM dominating during the other months of the year. In early June, NCM biomass started to increase and reached a maximum of 0.56 mmol C m$^{-3}$ on 18 July. Copepods biomass peaked shortly after (0.44 mmol C m$^{-3}$ on 1 September). Heterotrophic
bacteria biomass also started to increase in June and reaching a maximum of 0.12 mmol C m$^{-3}$ on 29 June. CM and PICO biomasses show similar dynamics, starting to increase in April and reaching a maximum in mid-June, before decreasing toward into September. NMPHYTO biomass remained low and close to its mean value of 0.01 mmol C m$^{-3}$ throughout the year (Fig. 7c).

### 3.3.2 Light limited conditions

In light limited conditions, the modelled yearly total C biomass was about 3 times higher than with nutrient limitation (1192.5 mmol C m$^{-3}$). NCM dominated the ecosystem (462.3 mmol C m$^{-3}$) followed by copepods (417.3 mmol C m$^{-3}$). NMPHYTO biomass was the lowest (59.2 mmol C m$^{-3}$, Fig. 7b).

Between late autumn and late spring copepods dominate while NCM become dominant in terms of biomass between mid-February and September. During this period, NCM biomass appears more variable compared to copepods and reaches a
maximum of 2.2 mmol C m$^{-3}$ on 14 June. Also heterotrophic bacteria showed a high variability particularly in summer, while remaining close to 0.15 mmol C m$^{-3}$ during the rest of the year. CM and PICO showed similar dynamics with their biomass starting to increase in early March before decreasing from mid-April and increasing again from mid-May till summer. They also showed their highest variability in summer. NMPHYTO biomass oscillated between 0.12 and 0.2 mmol C m$^{-3}$ showing a similar overall behaviour to CM and PICO except that the NMPHYTO maximum was reached on 23 April and not in
summer (Fig. 7d).

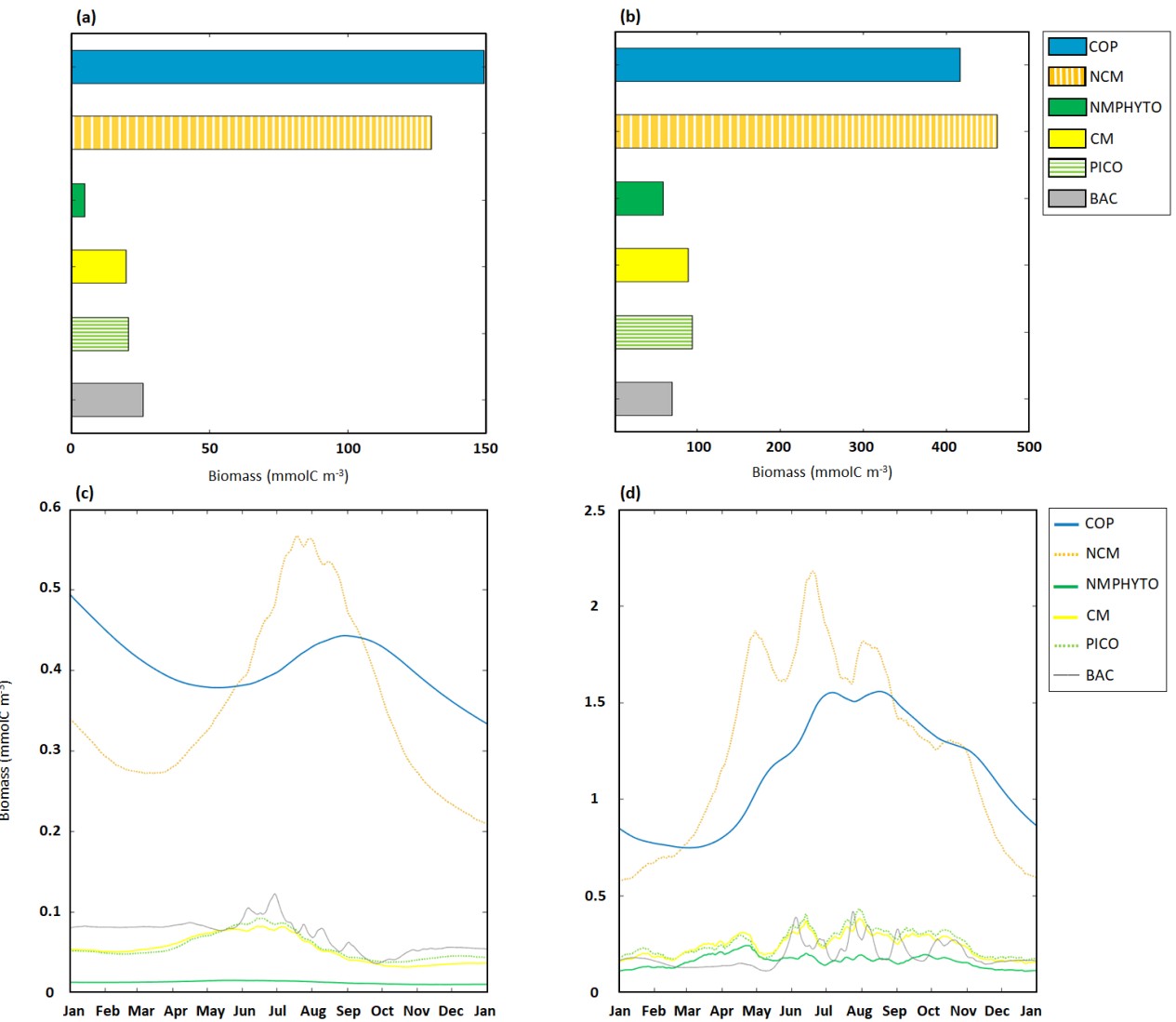

**Figure 7 : Yearly ecosystem C biomass composition and dynamics for copepods (COP), NCM, nano+micro-phytoplankton (NMPHYTO), CM, picophytoplankton (PICO) and heterotrophic bacteria (BAC). Yearly totals under (a) nutrient, and (b) light limited conditions. Time series of daily averages under (c) nutrient and (d) light limited conditions. Note the different scales on panels (a) and (b) as well as (c) and (d).**

## 3.4 Carbon, nitrogen, and phosphorus fluxes of mixotrophs

### 3.4.1 Carbon fluxes

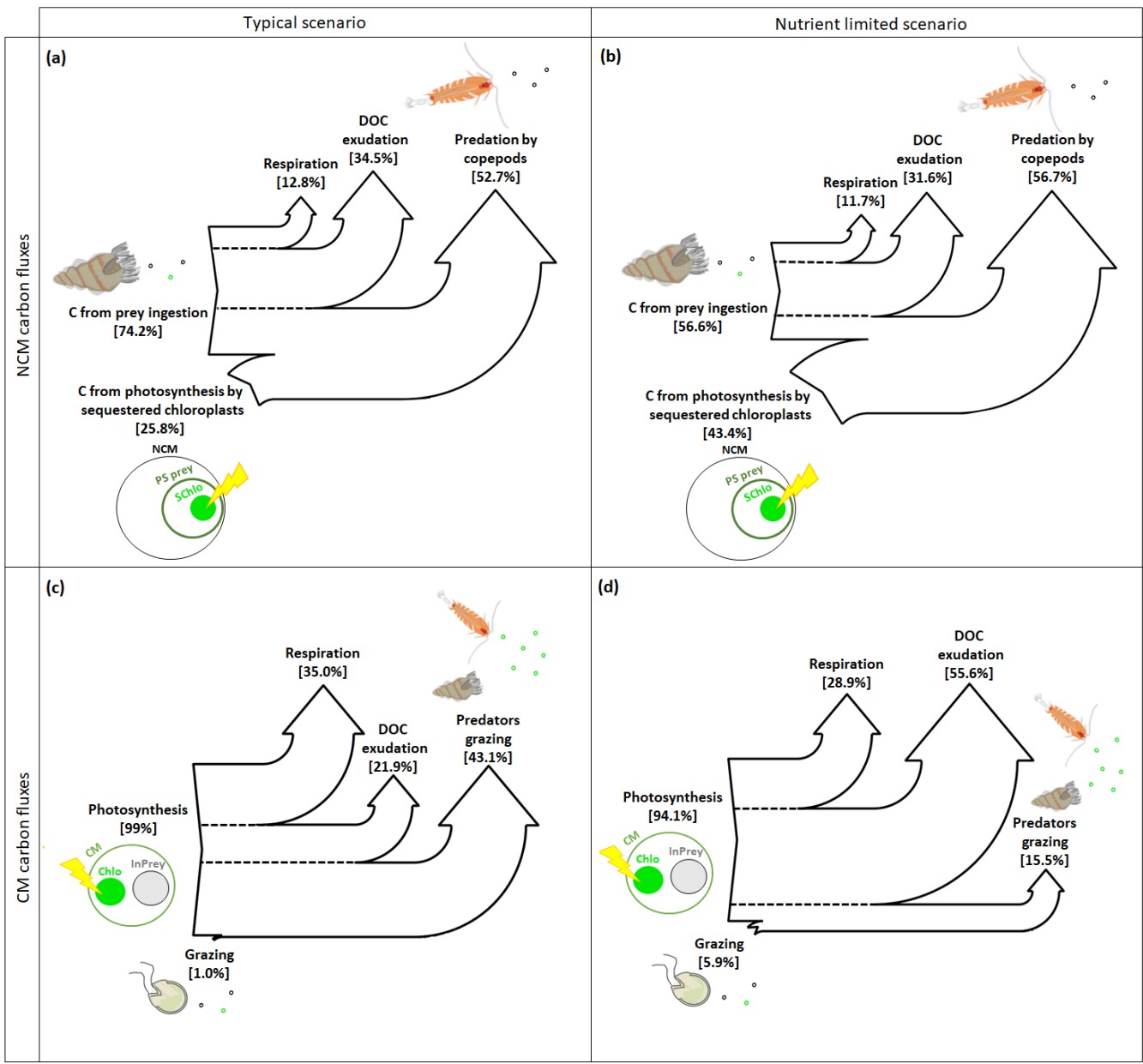

**Figure 8: Sankey diagrams showing the carbon (C) fluxes for NCM (a, b) and CM (c, d) in typical (a, c) and nutrient limited (b, d) scenarios. Numbers represent the yearly averaged C fluxes. PS prey: photosynthetic prey, InPrey: ingested prey, SChlo: sequestered chloroplast, Chlo: chloroplast.**

In typical and nutrient limited conditions, NCM can meet their metabolic needs by ingesting prey and by photosynthesizing using sequestered chloroplasts. In typical conditions (Fig. 8a), NCM obtained about three quarters of their C through prey

ingestion (74.2 %) and the remaining quarter through photosynthesis (25.8 %). The most significant loss terms are, in descending order, grazing by copepods, exudation of DOC, and respiration. In nutrient limited conditions (Fig. 8b), C uptake by photosynthesis and predation are more balanced (43.4% and 56.6 %, respectively) while the losses are similar to the typical scenario.

In contrast, when CM find themselves in typical conditions, they meet their metabolic needs almost through photosynthesis while grazing is almost negligible (Fig. 8c). The most important loss terms are grazing, followed by respiration, and DOC exudation. In nutrient limited conditions the role of grazing increases but only slightly and photosynthesis remains the dominant source of C (Fig. 8d). Interestingly, C loss terms change considerably under nutrient limitation: predation decreased significantly to become the least important loss term while more than half losses now occur via DOC exudation, while respiration decreased slightly.

### 3.4.2 Nitrogen and phosphorus fluxes

CM can complement their normal N and P uptake, i.e., DIM and DOM uptake (referred as total N or P uptake in Figure 9), by grazing. In typical conditions, grazing is insignificant to both N and P uptake (Fig. 9a, c), while losses occur predominantly through exudation of DON and DOP with predation representing only about one third.

In nutrient limited conditions (Fig. 9b, d), the role of grazing has increased substantially and now provides about 40 % of the N and a quarter of the P requirements. Also the loss terms have changed considerably, with N losses occurring almost exclusively due to grazing (Fig. 9b) while P losses appear equally split between DOP exudation and grazing (Fig. 9d).

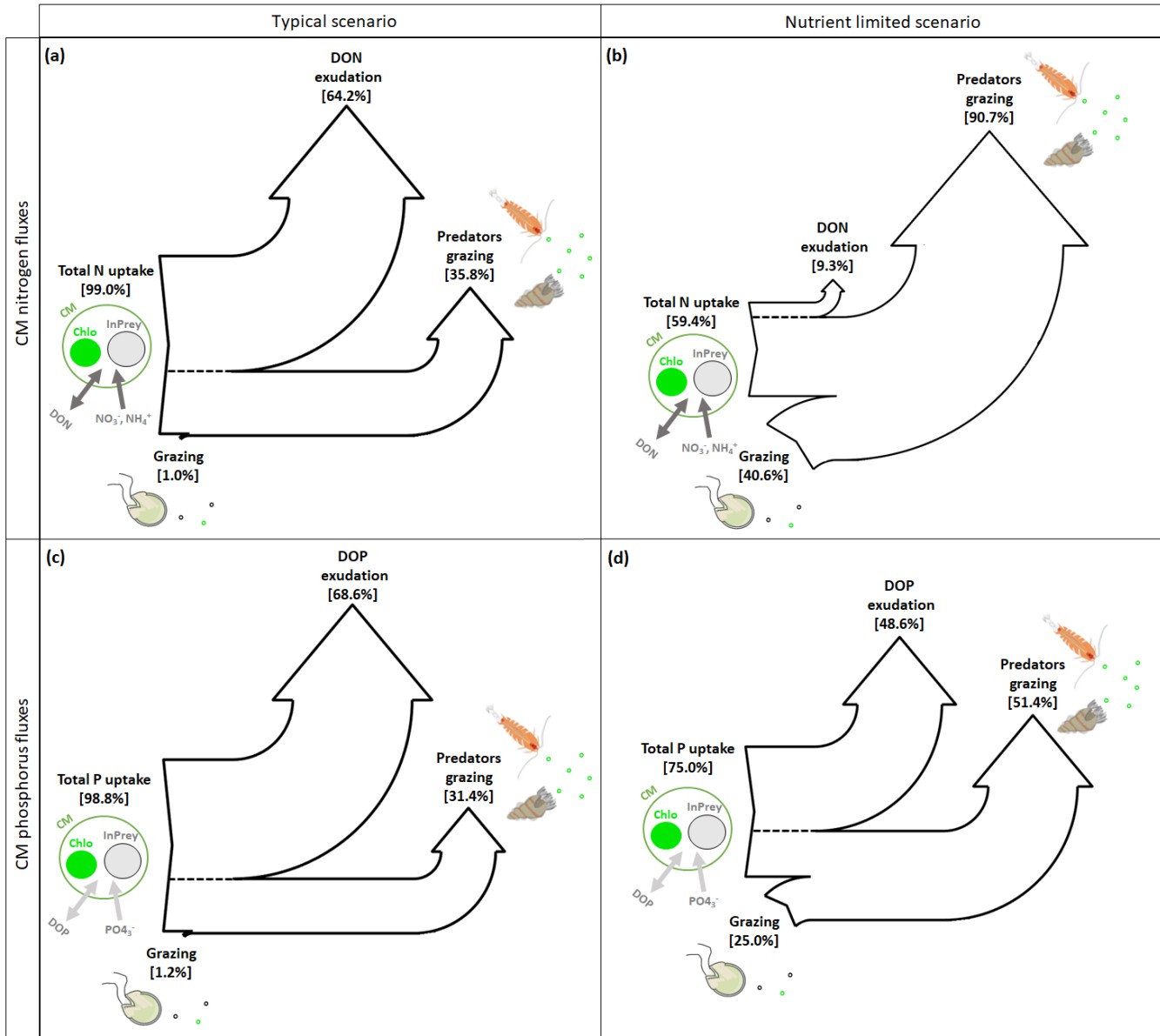

**Figure 9: Sankey diagrams showing (a, b) nitrogen (N), and (c, d) phosphorus (P) fluxes for CM in (a, c) typical and (b, d) nutrient limited conditions. Numbers represent the yearly averaged fluxes. InPrey: ingested prey and Chlo: chloroplast. Total N (P) represents the sum of DIN (DIP) and DON (DOP) uptakes.**

## 4 Discussion

Our results allowed us to determine the conditions which lead to the emergence of mixotrophs in the BoM. We show that mixotrophs are significantly impacted by nutrient limited conditions. In addition, the biogeochemical fluxes associated with

NCM and CM, showed that grazing and photosynthesis are strongly dependent on environmental conditions and can provide them with real competitive advantages.

In the following discussion, we decided to focus on CM as they are significant contributors to overall primary production (33 % of the total photosynthesis is performed by CM). Moreover, CM mixotrophy can significantly modify C, N, and P fluxes depending on environmental conditions.

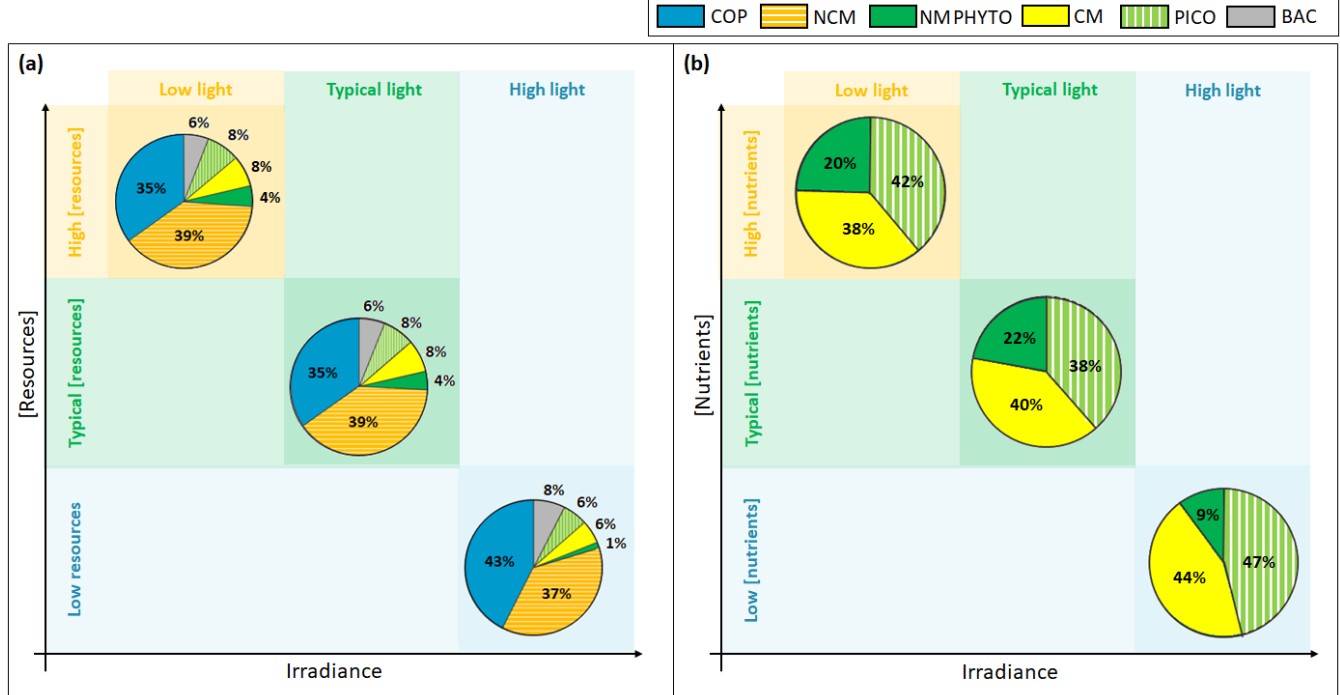

**Figure 10: Yearly (a) ecosystem and (b) phytoplankton composition in percentages of C biomass, in light limited, typical and**
**nutrient limited conditions. The term resources stands for both nutrients and preys.**

### 4.1 Mixotrophs representation assessment

As biomass measurements were not available for our location, we performed the assessment of these organisms based on properties listed in Table 2. We showed that NCM and CM properties were all well reproduced by the model (Fig. 5). The third NCM property : grazing and irradiance are independent (NCMP3, Table 2), required a constant NCM concentration to
be verified (Table 4). When irradiance increases, the NCM concentration increases. This feature is only due to the photosynthesis process which become less limited by light. NCM photosynthesis includes a prey dependant (i.e., based on preys' parameters) light limitation function (the closer the function is to 1, the less limited the organisms) which tends to 1 when irradiance increases. Grazing formulation does not include a term of direct dependence on light but includes NCM biomass which explains the increase of grazing when NCM biomass is not set to a constant. It seems difficult to avoid this
feature as photosynthesis is known to increase up to a certain value of irradiance which depends on species (Platt et al., 1980 ; Geider, 2013).

Regardless of the simulation we modelled close percentage of C biomass for NCM (ciliates) and copepods (difference maximum of 6% Fig. 10a). These percentages are always significantly higher than phytoplankton and heterotrophic bacteria ones. Even if, in the Gulf of Lion and especially in low salinity water from the Rhône River, oligotrich ciliates have been found abundant (Christaki et al., 2009), we do not exclude that, by only considering copepods as predator of NCM, we can underestimate the grazing that occurs on this type of organisms. In the actual model, we do not consider strict heterotrophs which belong to the nano and micro size classes. These organisms can be important competitors of ciliates, and certain species can even consume ciliates (Stoecker and Capuzzo, 1990 ; Johansson et al., 2004). The adding of these organisms could improve the representation of NCM dynamics and, accordingly, of the ecosystem and then, will be considered for an improved version of the model. Moreover, we do not consider a mortality term for NCM. Montagnes (1996) showed that mortality rates for two species of the genus *Strombidium* and two species of the genus *Strombilidium* were rapid. Accordingly, adding this term to the model could allow to represent a more realistic NCM biomass.

Regardless of the simulation, CM percentage in C biomass remains close to the phytoplankton one (Fig. 10a). We performed the assessment of phytoplankton for the typical simulation, by using SOLEMIO chlorophyll measurements (Fig. 5e, statistical analysis presented in Appendix G). According to statistic indicators, Eco3M_MIX-CarbOx reproduced well measured chlorophyl (cost function below 1 and RMSD close to 0). Especially, the model provided values in the same range than observations with relatively close mean (0.40 for the model and 0.39 for observations). Observed chlorophyll reached a maximum value in mid-March, linked to the Rhône River intrusion which is not reproduced by the model. This maximum can be linked to an input of allochthonous chlorophyll (i.e., phytoplankton development near the nutrients loaded Rhône River plume, which is brought to SOLEMIO by currents, Fraysse et al., 2014). As Eco3M_MIX-CarbOx is dimensionless (only time derivation), we do not represent this input which can explain that we are not able to reproduce this chlorophyll maximum. However, during this event, we reproduced the development and dominance of large cells (NMPHYTO) commonly observed in these cases (Fraysse et al., 2014).

## 4.2 Impact of limiting factors on ecosystem and phytoplankton composition

### 4.2.1 Light

In our light limited scenario, nutrient levels were kept artificially elevated throughout the year to prevent nutrients from becoming limiting and affecting the results. Light limitation had a considerable effect on total C biomass which was almost halved under low light compared to typical conditions (1192.5 mmol C m$^{-3}$ vs 2016.3 mmol C m$^{-3}$).

Ecosystem composition remained almost identical between light limited and typical conditions (Fig. 10a). In fact, light limitation only directly impacts the three phytoplankton groups, while copepods and NCM are only impacted indirectly through the effect of light on their prey. Heterotrophic bacteria do not become light limited in our model (Appendix C).

Considering that nutrients were kept artificially elevated in the light limited scenario, it is not surprising CM nutrition is almost entirely based on photosynthesis (99 %, result not shown), i.e., they behaved like strict autotrophs and their

mixotrophy did not represent a competitive advantage in this case. As instance, Stoecker et al. (1997) showed that in low light and high nutrient conditions, the CM *Prorocentrum minimum,* tend to photosynthesize rather than feed on prey as this latter mechanism only becomes relevant when inorganic nutrients are limiting. Thus, in light limited conditions, the phytoplankton arrangement only depends on the organism's ability to photosynthesize.

Although CM biomass remains high in low light, its share of the pie decreases in favour of NMPHYTO which seem to gain a slight edge. While the share of NMPHYTO increases slightly under low light PICO appears to be unaffected (Fig. 10a, b). In this simulation nutrient levels were kept artificially high to prevent nutrient limitation. By lifting the nutrient limitation NMPHYTO which is particularly sensitive to nutrients concentration, can grow more easily. In addition, NMPHYTO includes mainly diatoms which are known to be advantaged in low light environment (Fisher and Halsey, 2016). CM are more affected by low light and are not able to use mixotrophy in these conditions (nutrient concentration is high). The low effect of light on PICO agrees with observations by Timmermans et al. (2005) who showed that when nutrients are not co-limiting picophytoplankton still developed well.

The winter mixing event is a useful example that illustrates the impact of light on phytoplankton. During this event, the weather was particularly cloudy yielding low levels of ambient light and several decreases. These decreases in light level are reflected in the three phytoplankton groups limitation status which also decreased (which indicates an increase in limitation) (Fig. 6g).

### 4.2.2 Nutrients concentration

When nutrients are limiting, the shares of NCM, CM, PICO, and NMPHYTO decrease while copepods and heterotrophic bacteria show a relative increase (Fig. 10a). We found that when nutrient concentration was low, the ability of NCM to photosynthesize was particularly useful as it provided nearly half their C uptake (Fig. 8). Nevertheless, NCM yearly total biomass do not exceed the copepods one (Figs 7a, 10a). In fact, despite their ability to photosynthesize, NCM remained highly dependent on prey abundance. To prove this strong dependence of NCM on their prey, Mitra et al. (2016) performed several simulations involving different planktonic communities such as heterotrophic bacteria, phytoplankton, and NCM. They found that NCM biomass quickly increased but once the available prey was consumed, it dropped just as quickly. Due to this strong prey dependency, NCM cannot dominate the ecosystem throughout the year. Instead, we found that NCM biomass increased in summer (even exceeding copepods, Fig. 7c), right after CM and PICO biomass had increased, which in turn replenished the prey concentration.

Our modelled phytoplankton showed significant reactions to changes in nutrient concentration. While low nutrients led to an almost complete disappearance of NMPHYTO (Fig. 10a), CM and PICO appeared to handle low nutrient concentrations more easily. On the one hand, PICO are known to be able to cope with nutrient limited environments more efficiency than larger cells, mainly due to their small size which results in higher nutrient affinity (Agawin et al., 2000). On the other hand, nutrient limitation allowed CM to take full advantage of mixotrophy, which allows them to compensate a lack in DIN and DIP by grazing. Thus, by using two different competitive strategies, both PICO and CM can tolerate low nutrient conditions

allowed them to become the dominant phytoplankton groups in this scenario. Leles et al. (2018) also found relative increase in CM when nutrient concentration decreased.

The Rhône River and Cortiou water intrusions are useful examples that illustrate the impact of nutrient concentrations on the ecosystem and phytoplankton compositions. The Rhône River intrusion led to high $NO_3^-$ concentrations which in turn led to increased NMPHYTO growth, illustrating their high sensitivity to nutrient concentrations. NCM also fared well in this scenario and reached a dominant 39 % of the total C biomass (results not shown). In these conditions, NCM nutrition is mainly based on grazing (75.3 %) due to the high prey concentration but photosynthesis still represents a high percentage of the nutrition. In fact, some mixotrophic ciliates (*e.g., Laboea strobila*) are known to be highly dependent on photosynthesis (Stoecker et al., 1988; Sanders, 1991; Esteban et al., 2010). Stoecker et al. (1988) calculated that, in this case, photosynthesis via sequestered chloroplasts could contribute up to 37 % of the ciliate's total carbon demand in resources-rich conditions. The Cortiou water intrusion led to high $NH_4^+$ concentrations, alleviating the nutrient limitation for the three phytoplankton groups, particularly in NMPHYTO (Fig. 6f). In fact, immediately before this intrusion event, the ambient nutrient concentration was very low which explains the sudden response of phytoplankton. However, NMPHYTO still only represented 15 % of the total phytoplanktonic C biomass at the time (Fig. 6d), indicating that other factors are at play as well. As the Cortiou water intrusion took place during the summer upwelling period, we can hypothesize that temperature also have played a role in shaping the phytoplankton composition.

**4.3 Mixotrophy as: a strategy to overcome nutrient limitation in highly limited environments**

Several authors studied the functioning of food webs in oligotrophic environments, including subtropical gyres which cover about 40 % of the planet's surface and exhibit low production rates (Polovina et al., 2008). Mixotrophy is commonly observed in these gyres and has been recognized as crucial for plankton to survive in these environments (Zubkov and Tarran, 2008; Hartmann et al., 2012; Stoecker et al., 2017). Focusing on the Mediterranean Sea, several authors remarked the omnipresence of mixotrophic organisms (Pitta and Giannakouru, 2000; Christaki et al., 1999; Unrein et al., 2010), highlighting its importance in nutrient depleted areas. Using observations, Oikomonou et al. (2020) emphasized that mixotrophy was crucial in P-limited conditions and showed that mixotrophic flagellates grazed more on heterotrophic bacteria than the heterotrophic flagellates in these conditions. Moreover, both Oikomonou et al. (2020) and Christaki et al. (1999) observed that adding P to areas with P-limitation led to an immediate and pronounced reduction of grazing by mixotrophs. Livanou et al. (2021) drew similar conclusions using a modelling approach showing that, in a P-limited environment, organisms can meet about 90 % of their P requirements through grazing. This percentage drops to 17 % after P addition, as the organisms switch to uptake of DIP.

In agreement with these earlier studies, our model results indicated that the grazing component of mixotrophy increased when nutrients became limiting. This increase was significant for N and P as the percentage of grazing in the nutrition of CM was 40-fold higher for N and 25-fold higher for P. Despite these increases, the grazing percentages for P predicted by our model were still 3.5 times below the values in Livanou et al. (2021). In fact, in our nutrient limited simulation, CM were

mainly limited by N which explains why limitation had an even more pronounced effect on N fluxes. We can assume that when CM are mainly limited by P, the effect on P fluxes is more pronounced. Moreover, while we defined mixotrophy as the capability of a cell to use photo- and phagotrophy, other forms of mixotrophy exist in the ocean, e.g., osmotrophy which denotes an organism's ability to feed on dissolved organic compounds. Osmotrophy has been observed in a large variety of organisms and appears ubiquitous among phagotrophic phytoplankton (Sanders, 1991; Burkholder et al., 2008). Our model

can account for two forms of CM mixotrophy namely prey ingestion and DON/DOP uptake when DIN/DIP become limiting. In the nutrient limited simulation, CM osmotrophy represented a significant part of their N uptake as 43 % originated from DON. In typical sceanrio, this percentage dropped to 20 % which highlight the importance of osmotrophy as a source of N in low nutrients conditions. These results agree with observations which showed that osmotrophy can be a significant source of N and P for some microorganisms (Graneli et al., 1999; Lewitus, 2006). Also some HAB species obtained about 35 % of

their N uptake from DON (Glibert and Legrand, 2006). In contrast to the increase in grazing to supplement N and P nutrition in nutrient limited conditions, C uptake due to grazing remained low but still CM grazing fluxes on heterotrophic bacteria and PICO remained in the same ranges as observed by Livanou et al. (2019) for the ultra-oligotrophic Eastern Mediterranean Sea (Table 6). Other fluxes in C and especially DOC exudation were affected by the change in nutrient concentrations. DOC exudation reached about 56 % of the total C losses in nutrient limited conditions which is close to the percentage obtained by

Livanou et al. (2021) for DOC exudation before P addition (59 %). In low nutrient conditions, a small part of the C taken by CM was provided by grazing on heterotrophic bacteria. This C is released to the environment as DOC, as CM are unable to use organic C from their prey. The remaining C is provided by photosynthesis, but due to the low internal N:C and P:C ratios, CM release a large part to the environment as DOC. This released DOC can be used by heterotrophic bacteria unless they are limited by N and/or P (Thingstad et al., 1997).

**Table 6: Comparing modelled yearly CM grazing rates from the typical and nutrient limited scenarios to observations obtained by Livanou et al. (2019).**

|  | Typical | Nutrient limited | Livanou et al. (2019) |
|---|---|---|---|
| **Grazing by CM on heterotrophic bacteria (BAC CM$^{-1}$ h$^{-1}$)** | 0.03 | 0.1 | [0.04; 0.65] |
| **Grazing by CM on picophytoplankton (PICO CM$^{-1}$ h$^{-1}$)** | 0.02 | 0.03 | [0.006; 0.104] |

### 4.4 Why is it important to consider mixotrophy ?

An increasing number of studies has been investigating the impact of mixotrophs on their environment and were able to highlight the crucial role played by these organisms in the food web (Mitra et al., 2016; Ward and Follows, 2016; Ghyoot et

al., 2017 ; Stoecker et al., 2017). For instance, once Ward and Follows (2016) started to consider consider mixotrophs in their food web model, the biomass maximum switched to larger organisms which in turn led to an increase in carbon export to depth due to the production of larger carbon-enriched detritus. Still using a modelling approach (MIRO model), Ghyoot

and al. (2017) investigated the impact of the introduction of three forms of mixotrophy (osmotrophy, non-constitutive mixotrophy and constitutive mixotrophy) on trophic dynamics in the Southern North Sea. They showed that these three types of mixotrophy have different impact on system dynamics: while results showed that constitutive mixotrophy did not significantly affect the functioning of the ecosystem, osmotrophy increased gross primary production (GPP), sedimentation and bacterial production and non-constitutive mixotrophy also increased remineralisation and transfer to higher trophic level under high irradiance. Mixotrophy was also shown to play an important role in harmful algal blooms (Kempton et al., 2002 ; Burkholder et al., 2008). Accordingly, the need of developing models which include mixotrophy to represent and predict such events has been raised by several authors (Burkholder et al., 2008 ; McGillycuddy, 2010 ; Mitra & Flynn, 2010 ; Flynn & McGillicuddy, 2018). Moreover, as climate and anthropogenic changes could disrupt ecosystem functioning, some authors have highlighted that mixotrophs would occupy a central place in future ecosystems. Mitra et al. (2014) indicated that in future conditions of increased water column stability, and changed nutrient regimes, mixotrophs would have an increasing competitive advantage over strict autotrophs and heterotrophs.

Despite the central role that mixotrophs could play in ecosystems of the future, only few studies have investigated the impact of environmental forcings on these organisms. While some authors used in situ observations, mainly mesocosm experiments, to study the impact of light (Ptacknick et al., 2016), temperature (Wilken et al., 2013) or of a specific nutrient such as $PO_4^{3-}$ (Oikonomou et al., 2020) others, have chosen modelling approach to be able to study a wider range of parameters. For instance, Leles et al. (2018) investigated the impact of light and nutrient on mixotrophs and on their strict autotrophic and heterotrophic competitors modelling. They showed that changes in light and nutrients resulted in significant changes in ecosystem composition: while strict autotrophs and heterotrophs increased in relative importance in the transition from nutrient to light limitation, nutrient poor conditions favoured the development of mixotrophs. Still using modelling, Schneider et al. (2021) investigate the hypothesis that the biogeochemical gradient of inorganic nutrient and suspended sediment concentrations drives the observed occurrence of constitutive mixoplankton in the Dutch Southern North Sea. They showed that dissolved inorganic phosphate and silica concentration drive the occurrence of constitutive mixoplankton. Due to the scarcity of measurements and lack of spatial coverage, modelling approaches appear a viable and necessary alternative to gain further insight of mixotroph activity (particularly photosynthesis and grazing rates) and abundance as well as more detailed descriptions of mixotrophs characteristics which can be used for model validation, as was done here.

In the present work, we provided a relatively simple model (reduced number of compartments, 0D reasoning) to represent mixotrophy in the BoM. Even though we showed that we reproduced well the two types of mixotrophs modelled (all properties from Stoecker, 1998 were verified), Eco3M_MIX-CarbOx could still be improved. When developing Eco3M_MIX-CarbOx, we considered a simplify food web with a reduced number of compartments, consequently we made the choice to not consider strict heterotrophs which belong to the nano and micro size classes. This choice can affect the representation of NCM biomass as these organisms are known to compete with ciliates for resources. Some species can even ingest ciliates (Stoecker and Capuzzo, 1990 ; Johansson et al., 2004). Moreover, in the current version of the model, we do not take into account the possible increasing metabolic cost associated with mixotrophy (i.e., maintenance of both

autotrophic and heterotrophic apparatus). Raven (1997) suggested that the cost of maintaining phagotrophic apparatus for a primarily phototrophic organism remain low, but the cost of maintaining a phototrophic apparatus for a primarily phagotrophic organism can be significant and often resulting in lower growth rates than strict heterotrophs. It might be interesting to consider it as it could improve the representation of the NCM biomass.

For the particular location studied here, the Bay of Marseille (BoM), Eco3M_MIX-CarbOx is the first biogeochemical model to include an explicit compartment for mixotrophy in its representation of the food web. Eco3M_MIX-CarbOx used variable stoichiometry which allowed us to determine the nutritional state of the cell including potential nutrient limitation. This feature is even more important in the BoM where nutrient limitation has been shown to alternate between N and P several times during the year (Fraysse et al., 2013). We provided new insights regarding the conditions that lead to the emergence of mixotrophs in the BoM. Especially, we showed that, in the BoM, mixotrophy could represent a significant advantage when nutrients were limiting, particularly for CM. Even though Eco3M_MIX-CarbOx was developed and used in the BoM, it is easily adaptable to other coastal environments if environmental forcings are provided. This feature makes it a particularly suitable tool to perform long term studies and prediction of mixotrophy dynamics in coastal environments.

In this study, we focussed on the representation of mixotrophs in the model and on elucidating how different nutrient and light regimes affected the balance between mixotrophic uptake processes. However, other factors such as temperature and pH could also affect mixotrophs (Wilken et al., 2013; Razzak et al., 2015). Considering the effect of global change on these environmental forcings, it seems imperative to gain a better understanding of their effects on mixotrophs. Moreover, a modelling approach is particularly relevant to conduct when it comes to long-term studies and especially forecasts. As a next step, Eco3M_MIX-CarbOx will be coupled to a 3D hydrodynamic model which will allow us to study the effect of mixotrophs on the carbonate system as well as the impact of changes in the carbonate system on the emergence of mixotrophs. More generally, the coupled model should enable us to study the impacts of climate change on coastal ecosystem composition and on C fluxes.

## 5 Conclusions

Here we developed a new dimensionless biogeochemical model, Eco3M_MIX-CarbOx v1.0 to simulate the food web using variable stoichiometry in order to investigate the impact of light and nutrient limitations on the structuring of the planktonic ecosystem in a Mediterranean coastal area: the Bay of Marseille, France (BoM). In addition to the typical compartment for zooplankton, phytoplankton, and heterotrophic bacteria, Eco3M_MIX-CarbOx also contains a newly developed compartment to represent two types of mixotrophs: non-constitutive mixotrophs (NCM) and constitutive mixotrophs (CM). Due to the scarcity of actual measurements, we used the conceptual models from Stoecker (1998) to assess whether our model successfully reproduced the defining characteristics of mixotrophs. This could be demonstrated through a series of simulations involving changing light, nutrient and prey regimes in which the physiological traits of NCM and CM, were well reproduced by our model. We also ran a set of simulations to investigate (i) the evolution of phytoplankton composition in

typical light and nutrient conditions for the BoM, and especially during winter mixing, a Rhône River and Cortiou water intrusion, (ii) the evolution of the ecosystem composition under light and nutrient limited conditions and (iii) the evolution of C, N and P fluxes of NCM and CM once nutrients became limiting.

During the Rhône River and the Cortiou water intrusions, phytoplankton composition was mostly affected by changes in nutrient concentrations associated to these events. During the winter mixing event, variability in nutrients and light availability affected the organisms. Comparing the effects of light and nutrient limitation, nutrients had a more significant effect on ecosystem composition than light, although the limitation of either resource resulted in a decrease in overall C biomass. Regarding mixotrophs dynamic, the following trends emerged: (i) the portion of the ecosystem in percentage of C biomass occupied by NCM decreased when resources (prey and nutrients) decreased, (ii) the portion of the ecosystem in percentage of C biomass occupied by CM increased when nutrients decreased. We showed that when resource concentrations decreased, the contribution of photosynthesis to the C uptake of NCM increased, allowing them to maintain a carbon biomass almost as significant as the copepods one despite limiting conditions. When nutrients decreased, CM strongly increased the grazing component of their N and P uptake (by factors of 40 and 25, respectively). These results agree with previous studies which have shown that mixotrophy can represent a real competitive advantage in low nutrient (resource) conditions.

This work also provided new insights regarding the conditions that lead to the emergence of mixotrophs in the BoM. On a more general note, the model represents a new tool to perform long-term studies and predictions of mixotroph dynamics in coastal environments, particularly under different environmental forcings caused by global change where mixotrophs are expected to play a central role in future ecosystems. It is therefore important to gain a better understanding of how these organisms will respond to future light, nutrient, temperature, and pH scenario for example.

## Appendix A: State variables description and initial conditions values

**Table A1 : Summery of state variables description and initial condition values.**

| Compartments | State variables | Description | Initial condition | Units |
|---|---|---|---|---|
| Zooplankton | COP$_X$ | Copepod biomass in X <br> $X \in [C, N, P]$ | 0.700 <br> 0.106 <br> 0.007 | mmol X m$^{-3}$ |
| Mixotrophs | NCM$_X$ | Non constitutive mixotrophs biomass in X <br> $X \in [C, N, P]$ | 0.400 <br> 0.060 <br> 0.004 | mmol X m$^{-3}$ |
| | NCM$_{Chl}$ | Non constitutive mixotrophs chlorophyll concentration | 0.003 | mg Chl m$^{-3}$ |
| | CM$_X$ | Constitutive mixotrophs biomass in X <br> $X \in [C, N, P]$ | 0.200 <br> 0.030 <br> 0.002 | mmol X m$^{-3}$ |
| | CM$_{Chl}$ | Constitutive mixotrophs chlorophyll concentration | 0.080 | mg Chl m$^{-3}$ |
| Phytoplankton | NMPHYTO$_X$ | Nano+micro-phytoplankton biomass in X <br> $X \in [C, N, P]$ | 0.088 <br> 0.013 <br> 0.001 | mmol X m$^{-3}$ |
| | NMPHYTO$_{Chl}$ | Nano+micro-phytoplankton chlorophyll concentration | 0.020 | mg Chl m$^{-3}$ |
| | PICO$_X$ | Picophytoplankton biomass in X <br> $X \in [C, N, P]$ | 0.352 <br> 0.060 <br> 0.004 | mmol X m$^{-3}$ |
| | PICO$_{Chl}$ | Picophytoplankton chlorophyll concentration | 0.080 | mg Chl m$^{-3}$ |
| Heterotrophic bacteria | BAC$_X$ | Heterotrophic bacteria biomass in X <br> $X \in [C, N, P]$ | 0.108 <br> 0.025 <br> 0.002 | mmol X m$^{-3}$ |
| Dissolved Organic Matter (DOM) | DOX | Concentration of dissolved organic matter in X <br> $X \in [C, N, P]$ | 1.600 <br> 0.100 <br> 0.002 | mmol X m$^{-3}$ |
| Particulate Organic Matter (POM) | POX | Concentration of particulate organic matter in X <br> $X \in [C, N, P]$ | 5.700 <br> 0.700 <br> 0.050 | mmol X m$^{-3}$ |
| Dissolved Inorganic Matter (DIM) | NO$_3$ | Nitrate concentration | 0.700 | mmol N m$^{-3}$ |
| | NH$_4$ | Ammonium concentration | 0.060 | mmol N m$^{-3}$ |
| | PO$_4$ | Phosphate concentration | 0.030 | mmol P m$^{-3}$ |
| | O$_2$ | Oxygen concentration | 247.416 | mmol O m$^{-3}$ |

| | | | |
|---|---|---|---|
| TA | Total Alkalinity | 2660.496 | $\mu mol\ kg^{-1}$ |
| DIC | Dissolved Inorganic Carbon | 2358.430 | $\mu mol\ kg^{-1}$ |
| $pCO_2$ | Seawater $CO_2$ partial pressure | 371.283 | $\mu atm$ |
| $pH_T$ | pH on total scale | 8.110 | ø |
| $CaCO_3$ | Calcium carbonate concentration | 3.109 | $mmol\ m^{-3}$ |

## Appendix B: Balance equations

**Table B1: Balance equations**

| Compartments | Variables | Balance equations |
|---|---|---|
| Zooplankton | $COP_X$ $X \in [C, N, P]$ | $$\frac{\partial COP_C}{\partial t} = Gra_{COP_C}^{NCM_C} + Gra_{COP_C}^{NMPHYTO_C} + Gra_{COP_C}^{CM_C} - Resp_{COP_C}^{DIC} - Excr_{COP_C}^{DOC} - E_{COP_C}^{POC}$$ $$- Predation_{COP_C}^{POC}$$ $$\frac{\partial COP_N}{\partial t} = Gra_{COP_N}^{NCM_N} + Gra_{COP_N}^{NMPHYTO_N} + Gra_{COP_N}^{CM_N} - Excr_{COP_N}^{NH_4} - E_{COP_N}^{PON}$$ $$- Predation_{COP_N}^{PON}$$ $$\frac{\partial COP_P}{\partial t} = Gra_{COP_P}^{NCM_P} + Gra_{COP_P}^{NMPHYTO_P} + Gra_{COP_P}^{CM_P} - Excr_{COP_P}^{PO_4} - E_{COP_P}^{POP}$$ $$- Predation_{COP_P}^{POP}$$ |
| Mixotrophs | $NCM_X$ $X \in [C, N, P, Chl]$ | $$\frac{\partial NCM_C}{\partial t} = \sum_{i=1}^{2} \left( Gra_{NCM_C}^{PHY_{C_i}} \right) + Gra_{NCM_C}^{CM_C} + Gra_{NCM_C}^{BAC_C} + Photo_{NCM_C}^{DIC} - Resp_{NCM_C}^{DIC}$$ $$- Exu_{NCM_C}^{DOC} - Gra_{NCM_C}^{COP_C}$$ $$\frac{\partial NCM_N}{\partial t} = \sum_{i=1}^{2} \left( Gra_{NCM_N}^{PHY_{N_i}} \right) + Gra_{NCM_N}^{CM_N} + Gra_{NCM_N}^{BAC_N} - Exu_{NCM_N}^{DON} - Excr_{NCM_N}^{NH_4}$$ $$- Gra_{NCM_N}^{COP_N}$$ $$\frac{\partial NCM_P}{\partial t} = \sum_{i=1}^{2} \left( Gra_{NCM_P}^{PHY_{P_i}} \right) + Gra_{NCM_P}^{CM_P} + Gra_{NCM_P}^{BAC_P} - Exu_{NCM_P}^{DOP} - Excr_{NCM_P}^{PO_4}$$ $$- Gra_{NCM_P}^{COP_P}$$ $$\frac{\partial NCM_{CHL}}{\partial t} = \sum_{i=1}^{2} \left( Gra_{NCM_{Chl}}^{PHY_{Chl_i}} \right) + Gra_{NCM_{Chl}}^{CM_{Chl}} - Degrad_{NCM_{Chl}} - Gra_{NCM_{Chl}}^{COP_C}$$ $$PHY \in [NMPHYTO, PICO]$$ |
| | $CM_X$ $X \in [C, N, P, Chl]$ | $$\frac{\partial CM_C}{\partial t} = Gra_{CM_C}^{PICO_C} + Gra_{CM_C}^{BAC_C} + Photo_{CM_C}^{DIC} - Resp_{CM_C}^{DIC} - Exu_{CM_C}^{DOC} - Gra_{CM_C}^{NCM_C}$$ $$- Gra_{CM_C}^{COP_C}$$ $$\frac{\partial CM_N}{\partial t} = Gra_{CM_N}^{PICO_N} + Gra_{CM_N}^{BAC_N} + Upt_{CM_N}^{NO_3} + Upt_{CM_N}^{NH_4} + Upt_{CM_N}^{DON} - Exu_{CM_N}^{DON}$$ $$- Gra_{CM_N}^{NCM_N} - Gra_{CM_N}^{COP_N}$$ $$\frac{\partial CM_P}{\partial t} = Gra_{CM_P}^{PICO_P} + Gra_{CM_P}^{BAC_P} + Upt_{CM_P}^{PO_4} + Upt_{CM_P}^{DOP} - Exu_{CM_P}^{DOP} - Gra_{CM_P}^{NCM_P}$$ $$- Gra_{CM_P}^{COP_P}$$ $$\frac{\partial CM_{CHL}}{\partial t} = Syn_{CM_{Chl}} - Gra_{CM_{Chl}}^{NCM_{Chl}} - Gra_{CM_{Chl}}^{COP_C}$$ |

| | | |
|---|---|---|
| Phytoplankton | NMPHYTO$_X$<br>X ∈ [C, N, P, Chl] | $$\frac{\partial \text{NMPHYTO}_C}{\partial t} = \text{Photo}_{\text{NMPHYTO}_C}^{\text{DIC}} - \text{Resp}_{\text{NMPHYTO}_C}^{\text{DIC}} - \text{Exu}_{\text{NMPHYTO}_C}^{\text{DOC}} \\ - \text{Gra}_{\text{NMPHYTO}_C}^{\text{NCM}_C} - \text{Gra}_{\text{NMPHYTO}_C}^{\text{COP}_C}$$ $$\frac{\partial \text{NMPHYTO}_N}{\partial t} = \text{Upt}_{\text{NMPHYTO}_N}^{\text{NO}_3} + \text{Upt}_{\text{NMPHYTO}_N}^{\text{NH}_4} - \text{Exu}_{\text{NMPHYTO}_N}^{\text{DON}} - \text{Gra}_{\text{NMPHYTO}_N}^{\text{NCM}_N} \\ - \text{Gra}_{\text{NMPHYTO}_N}^{\text{COP}_N}$$ $$\frac{\partial \text{NMPHYTO}_P}{\partial t} = \text{Upt}_{\text{NMPHYTO}_P}^{\text{PO}_4} - \text{Exu}_{\text{NMPHYTO}_P}^{\text{DOP}} - \text{Gra}_{\text{NMPHYTO}_P}^{\text{NCM}_P} - \text{Gra}_{\text{NMPHYTO}_P}^{\text{COP}_P}$$ $$\frac{\partial \text{NMPHYTO}_{\text{Chl}}}{\partial t} = \text{Syn}_{\text{NMPHYTO}_{\text{Chl}}} - \text{Gra}_{\text{NMPHYTO}_{\text{Chl}}}^{\text{NCM}_{\text{Chl}}} - \text{Gra}_{\text{NMPHYTO}_{\text{Chl}}}^{\text{COP}_C}$$ |

| | | |
|---|---|---|
| | PICO$_X$<br>X ∈ [C, N, P, Chl] | $$\frac{\partial \text{PICO}_C}{\partial t} = \text{Photo}_{\text{PICO}_C}^{\text{DIC}} - \text{Resp}_{\text{PICO}_C}^{\text{DIC}} - \text{Exu}_{\text{PICO}_C}^{\text{DOC}} - \sum_{i=1}^{2} \left( \text{Gra}_{\text{PICO}_C}^{\text{MIX}_{C_i}} \right)$$ $$\frac{\partial \text{PICO}_N}{\partial t} = \text{Upt}_{\text{PICO}_N}^{\text{NO}_3} + \text{Upt}_{\text{PICO}_N}^{\text{NH}_4} + \text{Upt}_{\text{PICO}_N}^{\text{DON}} - \text{Exu}_{\text{PICO}_N}^{\text{DON}} - \sum_{i=1}^{2} \left( \text{Gra}_{\text{PICO}_N}^{\text{MIX}_{N_i}} \right)$$ $$\frac{\partial \text{PICO}_P}{\partial t} = \text{Upt}_{\text{PICO}_P}^{\text{PO}_4} + \text{Upt}_{\text{PICO}_P}^{\text{DOP}} - \text{Exu}_{\text{PICO}_P}^{\text{DOP}} - \sum_{i=1}^{2} \left( \text{Gra}_{\text{PICO}_P}^{\text{MIX}_{P_i}} \right)$$ $$\frac{\partial \text{PICO}_{\text{CHL}}}{\partial t} = \text{Syn}_{\text{PICO}_{\text{Chl}}} - \sum_{i=1}^{2} \left( \text{Gra}_{\text{PICO}_{\text{Chl}}}^{\text{MIX}_{\text{Chl}_i}} \right)$$ MIX ∈ [NCM, CM] |

| | | |
|---|---|---|
| Heterotrophic bacteria | BAC$_X$<br>X ∈ [C, N, P, Chl] | $$\frac{\partial \text{BAC}_C}{\partial t} = \text{BP}_{\text{BAC}_C}^{\text{DOC}} + \text{BP}_{\text{BAC}_C}^{\text{POC}} - \text{BR}_{\text{BAC}_C}^{\text{DIC}} - \text{Mort}_{\text{BAC}_C}^{\text{DOC}} - \sum_{i=1}^{2} \left( \text{Gra}_{\text{BAC}_C}^{\text{MIX}_{C_i}} \right)$$ $$\frac{\partial \text{BAC}_N}{\partial t} = \text{Upt}_{\text{BAC}_N}^{\text{NH}_4} + \text{Upt}_{\text{BAC}_N}^{\text{DON}} + \text{Upt}_{\text{BAC}_N}^{\text{PON}} - \text{Remin}_{\text{BAC}_N}^{\text{NH}_4} - \text{Mort}_{\text{BAC}_N}^{\text{DON}} \\ - \sum_{i=1}^{2} \left( \text{Gra}_{\text{BAC}_N}^{\text{MIX}_{N_i}} \right)$$ $$\frac{\partial \text{BAC}_P}{\partial t} = \text{Upt}_{\text{BAC}_P}^{\text{PO}_4} + \text{Upt}_{\text{BAC}_P}^{\text{DOP}} + \text{Upt}_{\text{BAC}_P}^{\text{POP}} - \text{Remin}_{\text{BAC}_P}^{\text{PO}_4} - \text{Mort}_{\text{BAC}_P}^{\text{DOP}} \\ - \sum_{i=1}^{2} \left( \text{Gra}_{\text{BAC}_P}^{\text{MIX}_{P_i}} \right)$$ MIX ∈ [NCM, CM] |

| | | |
|---|---|---|
| DOM | DOX $X \in$ [C, N, P] | $$\frac{\partial DOC}{\partial t} = \sum_{i=1}^{2}\left(Exu_{DOC}^{PHY_{C_i}}\right) + \sum_{i=1}^{2}\left(Exu_{DOC}^{MIX_{C_i}}\right) + Excr_{DOC}^{COP_C} + Mort_{DOC}^{BAC_C} - BP_{DOC}^{BAC_C}$$ $$\frac{\partial DON}{\partial t} = \sum_{i=1}^{2}\left(Exu_{DON}^{PHY_{N_i}}\right) + \sum_{i=1}^{2}\left(Exu_{DON}^{MIX_{N_i}}\right) + Mort_{DON}^{BAC_N} - Upt_{DON}^{CM_N} - Upt_{DON}^{PICO_N}$$ $$- Upt_{DON}^{BAC_N}$$ $$\frac{\partial DOP}{\partial t} = \sum_{i=1}^{2}\left(Exu_{DOP}^{PHY_{P_i}}\right) + \sum_{i=1}^{2}\left(Exu_{DOP}^{MIX_{P_i}}\right) + Mort_{DOP}^{BAC_P} - Upt_{DOP}^{CM_P} - Upt_{DOP}^{PICO_P}$$ $$- Upt_{DOP}^{BAC_P}$$ PHY $\in$ [NMPHYTO, PICO], MIX $\in$ [NCM, CM] |
| POM | POX $X \in$ [C, N, P] | $$\frac{\partial POC}{\partial t} = E_{POC}^{COP_C} + Predation_{POX}^{COP_X} - BP_{POC}^{BAC_C}$$ $$\frac{\partial PON}{\partial t} = E_{PON}^{COP_N} + Predation_{PON}^{COP_N} - Upt_{PON}^{BAC_N}$$ $$\frac{\partial POP}{\partial t} = E_{POP}^{COP_P} + Predation_{POP}^{COP_P} - Upt_{POP}^{BAC_P}$$ |
| MID | $NO_3$ | $$\frac{\partial NO_3}{\partial t} = Nitrif_{NO_3}^{NH_4} - \sum_{i=1}^{2} Upt_{NO_3}^{Phy_{N_i}} - Upt_{NO_3}^{CM_{N_i}}$$ PHY $\in$ [NMPHYTO, PICO] |
| | $NH_4$ | $$\frac{\partial NH_4}{\partial t} = Excr_{NH_4}^{COP_{N_i}} + Excr_{NH_4}^{NCM_{N_i}} + Remin_{NH_4}^{BAC_N} - \sum_{i=1}^{2}\left(Upt_{NH_4}^{Phy_{N_i}}\right) - Upt_{NH_4}^{CM_N}$$ $$- Upt_{NH_4}^{BAC_N} - Nitrif_{NH_4}^{NO_3}$$ PHY $\in$ [NMPHYTO, PICO] |
| | $PO_4$ | $$\frac{\partial PO_4}{\partial t} = Excr_{PO_4}^{COP_{P_i}} + Excr_{PO_4}^{NCM_{P_i}} + Remin_{PO_4}^{BAC_P} - \sum_{i=1}^{2}\left(Upt_{PO_4}^{PHY_{P_i}}\right) - Upt_{PO_4}^{CM_P}$$ $$- Upt_{PO_4}^{BAC_P}$$ PHY $\in$ [NMPHYTO, PICO] |
| | $CaCO_3$ | $$\frac{\partial CaCO_3}{\partial t} = Prec_{DIC}^{CaCO_3} - Diss_{DIC}^{CaCO_3}$$ |

$$\frac{\partial O_2}{\partial t} = \left(\frac{O}{C}\right)_{PP} * \sum_{i=1}^{2}\left(Photo_{O_2}^{PHY_i}\right) + \left(\frac{O}{C}\right)_{PP} \cdot \sum_{i=1}^{2}\left(Photo_{O_2}^{Mix_i}\right) + Aera_{O_2}$$

O_2

$$- \sum_{i=1}^{2}\left(Resp_{O_2}^{Phy_i}\right) - \sum_{i=1}^{2}\left(Resp_{O_2}^{MIX_i}\right) - Resp_{O_2}^{COP} - BR_{O_2}^{BAC}$$

$$- \left(\frac{O}{C}\right)_{NITRIF} \cdot Nitrif_{O_2}$$

PHY $\epsilon$ [NMPHYTO, PICO], MIX $\epsilon$ [NCM, CM]

MID

$$\frac{\partial TA}{\partial t} = 2.\,Diss_{TA}^{CaCO_3} + \sum_{i=1}^{2}\left(Upt_{NO_3}^{Phy_{N_i}}\right) + Upt_{NO_3}^{CM_N} + \sum_{i=1}^{2}\left(Upt_{PO_4}^{PHY_{P_i}}\right) + Upt_{PO_4}^{CM_P}$$

TA

$$+ Remin_{NH_4}^{BAC_N} - \sum_{i=1}^{2}\left(Upt_{NH_4}^{PHY_{N_i}}\right) - Upt_{NH_4}^{CM_N} - Remin_{PO_4}^{BAC_P}$$

$$- 2.\,Prec_{TA}^{CaCO_3} - 2.\,Nitrif_{TA}$$

PHY $\epsilon$ [NMPHYTO, PICO]

$$\frac{\partial DIC}{\partial t} = \sum_{i=1}^{2}\left(Resp_{DIC}^{PHY_{C_i}}\right) + \sum_{i=1}^{2}\left(Resp_{DIC}^{MIX_{C_i}}\right) + Resp_{DIC}^{COP_C} + BR_{DIC}^{BAC_C} + Aera_{DIC}$$

DIC

$$+ Diss_{DIC}^{CaCO_3} - \sum_{i=1}^{2}\left(Photo_{DIC}^{PHY_{C_i}}\right) - \sum_{i=1}^{2}\left(Photo_{DIC}^{MIX_{C_i}}\right)$$

$$- Prec_{DIC}^{CaCO_3}$$

PHY $\epsilon$ [NMPHYTO, PICO], MIX $\epsilon$ [NCM, CM]

## Appendix C: Processes descriptions, formulations, and units

**Table C1: Biogeochemical processes simulated by Eco3M_MIX-CarbOx for zooplankton**

| Notation | Description | Formulation | Units |
|---|---|---|---|
| **Zooplankton** | | | |
| $Gra_{COP_C}^{PREY_C}$ <br> *PREY $\epsilon$ [NCM, CM, NMPHYTO] | Copepods grazing on $PREY_C$ | $Gra_{COP_C}^{PREY_C}$ $= G_{MAX} * \dfrac{(\Phi * PREY_C^2)}{K_{COP} * \sum_{i=1}^{3}(\Phi * PREY_{C_i}) + \sum_{i=1}^{3}(\Phi * PREY_{C_i}^2)}$ $* COP_C$ | mmol C m$^{-3}$ s$^{-1}$ |
| $Gra_{COP_X}^{PREY_X}$ <br> *PREY $\epsilon$ [NCM, CM,NMPHYTO] <br> *X $\epsilon$ [N, P] | Copepods grazing on $PREY_X$ | $Gra_{COP_X}^{PREY_X} = Gra_{COP_C}^{PREY_C} * \dfrac{PREY_X}{PREY_C}$ | mmol X m$^{-3}$ s$^{-1}$ |
| $Resp_{COP_C}^{DIC}$ | Copepods respiration | $Resp_{COP_C}^{DIC} = \sum_{i=1}^{3}\left(frac_{resp} * \left(Gra_{COP_C}^{PREY_{C_i}} * (1 - f_Q^G)\right)\right)$ | mmol C m$^{-3}$ s$^{-1}$ |
| $Excr_{COP_C}^{DOC}$ | Copepods excretion of DOC | $Excr_{COP_C}^{DOC} = \sum_{i=1}^{3}\left((1 - frac_{resp}) * (1 - f_{Q,PREY_{Ci}}^G)\right.$ $\left. * \left(Gra_{COP_C}^{PREY_{C_i}} * (1 - f_Q^G)\right)\right)$ | mmol C m$^{-3}$ s$^{-1}$ |
| $Excr_{COP_X}^{Nut_X}$ <br> *Nut$_X$ $\epsilon$ [NH$_4^+$, PO$_4^{3-}$] <br> *X $\epsilon$ [N, P] | Copepods excretion of Nut$_X$ | $Excr_{COP_X}^{Nut_X} = \sum_{i=1}^{3}\left((1 - f_{Q,PREY_{Ci}}^G) * \left(Gra_{COP_X}^{PREY_{X_i}} * (1 - f_Q^U)\right)\right)$ | mmol X m$^{-3}$ s$^{-1}$ |
| $E_{COP_C}^{POC}$ | Copepods egestion of POC | $E_{COP_C}^{POC} = \sum_{i=1}^{3}\left((1 - frac_{Resp})\right.$ $\left. * \left(f_{Q,PREY_{Ci}}^G * Gra_{COP_C}^{PREY_{C_i}} * (1 - f_Q^G)\right)\right)$ | mmol C m$^{-3}$ s$^{-1}$ |
| $E_{COP_X}^{POX}$ <br> *X $\epsilon$ [N, P] | Copepods egestion of POX | $E_{COP_X}^{POX} = \sum_{i=1}^{3}\left(f_{Q,PREY_{Xi}}^G * \left(Gra_{COP_X}^{PREY_{X_i}} * (1 - f_Q^U)\right)\right)$ | mmol X m$^{-3}$ s$^{-1}$ |
| $Predation_{COP_X}^{POX}$ <br> *X $\epsilon$ [C, N, P] | Higher trophic levels predation on copepods | $Predation_{COP_X}^{POX} = k_{mort} * COP_X^2$ | mmol X m$^{-3}$ s$^{-1}$ |

 **Table C2: Biogeochemical processes simulated by Eco3M_MIX-CarbOx for non-constitutive mixotrophs**

| Notation | Description | Formulation | Units |
|---|---|---|---|
| **MIXOTROPHS (Non-constitutive mixotrophs)** | | | |
| $Gra_{NCM_C}^{PREY_C}$ <br><br> *PREY $\epsilon$ [CM, NMPHYTO, PICO, BAC] | NCM grazing on $PREY_C$ | $$Gra_{NCM_C}^{PREY_C} = G_{MAX} * \frac{(\Phi * PREY_C^2)}{K_{NCM} * \sum_{i=1}^{4}(\Phi * PREY_{C_i}) + \sum_{i=1}^{4}(\Phi * PREY_{C_i}^2)} * NCM_C$$ | mmol C m$^{-3}$ s$^{-1}$ |
| $Gra_{NCM_{Chl}}^{PREY_{Chl}}$ <br><br> *PREY $\epsilon$ [CM, NMPHYTO, PICO] | NCM grazing on $PREY_{Chl}$ | $$Gra_{NCM_{Chl}}^{PREY_{Chl}} = Gra_{NCM_C}^{PREY_C} * \frac{PREY_{Chl}}{PREY_C}$$ | mg Chl m$^{-3}$ s$^{-1}$ |
| $Gra_{NCM_X}^{PREY_X}$ <br><br> *PREY $\epsilon$ [CM, NMPHYTO, PICO, BAC] <br><br> *X $\epsilon$ [N, P] | NCM grazing on $PREY_X$ | $$Gra_{NCM_X}^{PREY_X} = Gra_{NCM_C}^{PREY_C} * \frac{PREY_X}{PREY_C}$$ | mmol X m$^{-3}$ s$^{-1}$ |
| $Photo_{NCM_C}^{DIC}$ | NCM photosynthesis | $$Photo_{DIC}^{NCM_C} = \sum_{i=1}^{3}(\Phi_i * P_{Ref,PREY_i}^{C} * f_{PREY_i}^{T} * f_Q^{G} * limI_{PREY_i} * NCM_C)$$ | mmol C m$^{-3}$ s$^{-1}$ |
| $Resp_{NCM_C}^{DIC}$ | NCM respiration | $$Resp_{NCM_C}^{DIC} = \sum_{i=1}^{4}\left(frac_{resp} * \left(Gra_{NCM_C}^{PREY_{C_i}} * (1 - f_Q^{G})\right)\right)$$ | mmol C m$^{-3}$ s$^{-1}$ |
| $Exu_{NCM_C}^{DOC}$ | NCM exudation of DOC | $$Exu_{NCM_C}^{DOC} = \sum_{i=1}^{4}\left((1 - frac_{Resp}) * Gra_{NCM_C}^{PREY_{C_i}} * (1 - f_Q^{G})\right)$$ | mmol C m$^{-3}$ s$^{-1}$ |
| $Exu_{NCM_X}^{DOX}$ <br><br> *X $\epsilon$ [N, P] | NCM exudation of DOX | $$Exu_{NCM_X}^{DOX} = \sum_{i=1}^{4}\left(frac_{MOD} * Gra_{NCM_X}^{PREY_{X_i}} * (1 - f_Q^{U})\right)$$ | mmol X m$^{-3}$ s$^{-1}$ |
| $Excr_{NCM_X}^{Nut_X}$ <br><br> *Nut$_X$ $\epsilon$ [NH$_4^+$, PO$_4^{3-}$] <br><br> *X $\epsilon$ [N, P] | NCM excretion of $Nut_X$ | $$Excr_{NCM_X}^{Nut_X} = \sum_{i=1}^{4}\left((1 - frac_{MOD}) * Gra_{NCM_X}^{PREY_{X_i}} * (1 - f_Q^{U})\right)$$ | mmol X m$^{-3}$ s$^{-1}$ |
| $Degrad_{NCM_{Chl}}$ | NCM chlorophyll degradation | $$Degrad_{NCM_{Chl}} = \left(\left(Gra_{NCM_{Chl}}^{PREY_{Chl}} * dt\right) + NCM_{Chl}\right) * k_{MORT,Chl}$$ | mg Chl m$^{-3}$ s$^{-1}$ |

**Table C3: Biogeochemical processes simulated by Eco3M_MIX-CarbOx for constitutive mixotrophs**

| Notation | Description | Formulation | Units |
|---|---|---|---|
| **MIXOTROPHS (Constitutive mixotrophs)** | | | |
| $\text{Gra}_{CM_C}^{PREY_C}$ <br> *PREY $\epsilon$ [PICO, BAC] | CM grazing of $PREY_C$ | $$\text{Gra}_{CM_C}^{PREY_C} = \left(\left(G_{MAX} * \frac{(\Phi * PREY_C^2)}{K_{CM} * \sum_{i=1}^{2}(\Phi_i * PREY_{C_i}) + \sum_{i=1}^{2}(\Phi_i * PREY_{C_i}^2)}\right) * \left(1 - \exp\left(\frac{-\alpha_{Chl} * Q_C^{Chl} * E_{PAR}}{P_{Ref}^C}\right)\right) * f_{inhib}^{CM}\right)$$ | mmol C m$^{-3}$ s$^{-1}$ |
| $\text{Photo}_{CM_C}^{DIC}$ | CM photosynthesis | $$\text{Photo}_{CM_C}^{DIC} = P_{MAX}^C * \text{limI} * CM_C$$ | mmol C m$^{-3}$ s$^{-1}$ |
| $\text{Resp}_{CM_C}^{DIC}$ | CM respiration | $$\text{Resp}_{CM_C}^{DIC} = \sum_{i=1}^{3}\left(\text{cout}_{resp}^{Nut_X} * \mu_{PPB}^{NR} * Q_{C,max}^X * \frac{Nut_{X_i}}{Nut_{X_i} + K_{Nut_{X_i}}} * CM_C\right) + \text{frac}_{resp} * \text{Photo}_{CM_C}^{DIC}$$ <br><br> *Nut$_X$ $\epsilon$ [NO$_3^-$, NH$_4^+$, PO$_4^{3-}$] | mmol C m$^{-3}$ s$^{-1}$ |
| $\text{Upt}_{CM_X}^{Nut_X}$ <br> *Nut$_X$ $\epsilon$ [NO$_3^-$, NH$_4^+$, PO$_4^{3-}$] <br> *X $\epsilon$ [N, P] | CM uptake of Nut$_X$ | $$\text{Upt}_{CM_X}^{Nut_X} = \mu_{NR}^{PPB} * Q_{C,max}^X * \frac{Nut_X}{Nut_X + K_{Nut_X}} * CM_C$$ | mmol X m$^{-3}$ s$^{-1}$ |
| $\text{Upt}_{CM_X}^{DOX}$ <br> *X $\epsilon$ [N, P] | CM uptake of DOX | $$\text{Upt}_{CM_X}^{DOX} = \mu_{NR}^{PPB} * Q_{C,max}^X * \frac{DOX}{DOX + K_{DOX}} * CM_C * f_Q^U$$ | mmol X m$^{-3}$ s$^{-1}$ |
| $\text{Exu}_{CM_C}^{DOC}$ | CM exudation of DOC | $$\text{Exu}_{CM_C}^{DOC} = (1 - \text{frac}_{resp}) * \left(\text{Photo}_{CM_C}^{DIC} * (1 - f_Q^G)\right) + \sum_{i=1}^{2}\left(\text{Gra}_{CM_C}^{PREY_{C_i}}\right)$$ | mmol C m$^{-3}$ s$^{-1}$ |
| $\text{Exu}_{CM_N}^{DON}$ | CM exudation of DON | $$\text{Exu}_{CM_N}^{DON} = \sum_{i=1}^{2}\left(\left(\mu_{PPB}^{NR} * Q_{C,max}^N * \frac{Nut_{X_i}}{Nut_{X_i} + K_{Nut_{X_i}}} * CM_C\right) * (1 - f_Q^U)\right)$$ <br><br> *Nut$_X$ $\epsilon$ [NO$_3^-$, NH$_4^+$] | mmol N m$^{-3}$ s$^{-1}$ |
| $\text{Exu}_{CM_P}^{DOP}$ | CM exudation of DOP | $$\text{Exu}_{CM_P}^{DOP} = \mu_{PPB}^{NR} * Q_{C,max}^P * \frac{PO_4^{3-}}{PO_4^{3-} + K_{PO_4}} * CM_C * (1 - f_Q^U)$$ | mmol P m$^{-3}$ s$^{-1}$ |
| $\text{Syn}_{CM_{Chl}}$ | CM chlorophyll synthesis | $$\text{Syn}_{CM_{Chl}} = Q_C^N * \left(Q_{N,min}^{Chl} + f_Q^N * \left(Q_{N,max}^{Chl} - Q_{N,min}^{Chl}\right)\right) * CM_C$$ | mg Chl m$^{-3}$ s$^{-1}$ |

 **Table C4: Biogeochemical processes simulated by Eco3M_MIX-CarbOx for phytoplankton**

| Notation | Description | Formulation | Units |
|---|---|---|---|
| **PHYTOPLANKTON (nano+micro-phytoplankton and picophytoplankton)** | | | |
| $\text{Photo}_{\text{PHY}_C}^{\text{DIC}}$<br><br>*PHY $\epsilon$ [NMPHYTO, PICO] | Phytoplankton photosynthesis | $$\text{Photo}_{\text{PHY}_C}^{\text{DIC}} = P_{\text{MAX}}^C * \text{limI} * \text{PHY}_C$$ | mmol C m$^{-3}$ s$^{-1}$ |
| $\text{Resp}_{\text{PHY}_C}^{\text{DIC}}$<br><br>*PHY $\epsilon$ [NMPHYTO, PICO] | Phytoplankton respiration | $$\text{Resp}_{\text{PHY}_C}^{\text{DIC}} = \sum_{i=1}^{3} \left( \text{cout}_{\text{resp}}^{\text{Nut}_X} * \mu_{\text{PPB}}^{\text{NR}} * Q_{C,\text{max}}^X * \frac{\text{Nut}_{X_i}}{\text{Nut}_{X_i} + K_{\text{Nut}_{X_i}}} * \text{PHY}_C \right) + \text{frac}_{\text{resp}} * \text{Photo}_{\text{PHY}_C}^{\text{DIC}}$$<br><br>*Nut$_X \epsilon$ [NO$_3^-$, NH$_4^+$, PO$_4^{3-}$] | mmol C m$^{-3}$ s$^{-1}$ |
| $\text{Upt}_{\text{PHY}_X}^{\text{Nut}_X}$<br><br>*PHY $\epsilon$ [NMPHYTO, PICO]<br>*X $\epsilon$ [N, P]<br><br>*Nut$_X \epsilon$ [NO$_3^-$, NH$_4^+$, PO$_4^{3-}$] | Phytoplankton uptake of Nut$_X$ | $$\text{Upt}_{\text{PHY}_X}^{\text{Nut}_X} = \mu_{\text{PPB}}^{\text{NR}} * Q_{C,\text{max}}^X * \frac{\text{Nut}_X}{\text{Nut}_X + K_{\text{Nut}_X}} * \text{PHY}_C$$ | mmol X m$^{-3}$ s$^{-1}$ |
| $\text{Exu}_{\text{PHY}_C}^{\text{DOC}}$<br><br>*PHY $\epsilon$ NMPHYTO, PICO] | Phytoplankton exudation of DOC | $$\text{Exu}_{\text{PHY}_C}^{\text{DOC}} = (1 - \text{frac}_{\text{resp}}) * \left( \text{Photo}_{\text{PHY}_C}^{\text{DIC}} * (1 - f_Q^G) \right)$$ | mmol C m$^{-3}$ s$^{-1}$ |
| $\text{Exu}_{\text{PHY}_N}^{\text{DON}}$<br><br>*PHY $\epsilon$ [NMPHYTO, PICO] | Phytoplankton exudation of DON | $$\text{Exu}_{\text{PHY}_N}^{\text{DON}} = \sum_{i=1}^{2} \left( \left( \mu_{\text{PPB}}^{\text{NR}} * Q_{C,\text{max}}^X * \frac{\text{Nut}_{X_i}}{\text{Nut}_{X_i} + K_{\text{Nut}_{X_i}}} * \text{PHY}_C \right) * (1 - f_Q^U) \right)$$<br><br>*NutX $\epsilon$ [NO$_3^-$, NH$_4^+$] | mmol N m$^{-3}$ s$^{-1}$ |
| $\text{Exu}_{\text{PHY}_P}^{\text{DOP}}$<br><br>*PHY $\epsilon$ [NMPHYTO, PICO] | Phytoplankton exudation of DOP | $$\text{Exu}_{\text{PHY}_P}^{\text{DOP}} = \mu_{\text{PPB}}^{\text{NR}} * Q_{C,\text{max}}^P * \frac{\text{PO}_4^{3-}}{\text{PO}_4^{3-} + K_{\text{PO}_4}} * \text{PHY}_C * (1 - f_Q^U)$$ | mmol P m$^{-3}$ s$^{-1}$ |
| $\text{Syn}_{\text{Phy}_{\text{Chl}}}$<br><br>*PHY $\epsilon$ [NMPHYTO, PICO] | Phytoplankton chlorophyll synthesis | $$\text{Syn}_{\text{Phy}_{\text{Chl}}} = Q_C^N * \left( Q_{N,\text{min}}^{\text{Chl}} + f_Q^N \left( Q_{N,\text{max}}^{\text{Chl}} - Q_{N,\text{min}}^{\text{Chl}} \right) \right) * \text{PHY}_C$$ | mg Chl m$^{-3}$ s$^{-1}$ |
| **PHYTOPLANKTON (Picophytoplankton only)** | | | |

| $Upt_{PICO_X}^{DOX}$ $*X \in [N, P]$ | Picophytoplankton uptake of DOX | $Upt_{PICO_X}^{DOX} = \mu_{PPB}^{NR} * Q_{C,max}^{X} * \dfrac{DOX}{DOX + K_{DOX}} * PICO_C * f_Q^U$ | mmol X m$^{-3}$ s$^{-1}$ |
|---|---|---|---|

**Table C5: Biogeochemical processes simulated by Eco3M_MIX-CarbOx for heterotrophic bacteria**

| Notation | Description | Formulation | Units |
|---|---|---|---|
| | | **HETEROTROPHIC BACTERIA** | |
| $BP_{BAC_C}^{DOC}$ | Bacterial production on DOC | $BP_{BAC_C}^{DOC} = \mu_{MAX}^{BAC} * \dfrac{DOC}{DOC + K_{DOC}} * BAC_C * f_{Q_{10}}^{T} * f_{Q}^{G}$ | mmol C m$^{-3}$ s$^{-1}$ |
| $BP_{BAC_C}^{POC}$ | Bacterial production on POC | $BP_{BAC_C}^{POC} = \mu_{MAX}^{BAC} * \dfrac{POC}{POC + K_{POC}} * BAC_C * f_{Q_{10}}^{T}$ | mmol C m$^{-3}$ s$^{-1}$ |
| $BR_{BAC_C}^{DIC}$ | Bacterial respiration | $BR_{BAC_C}^{DIC} = (1 - bge)$ $* \left( \sum\limits_{i=1}^{2} \left( \mu_{MAX}^{BAC} * \dfrac{X_i}{X_i + K_{X_i}} * BAC_C * f_{Q_{10}}^{T} * f_{Q}^{G} \right) \right)$ $*X \in [DOC, POC]$ | mmol C m$^{-3}$ s$^{-1}$ |
| $Upt_{BAC_X}^{Element_X}$ *Element$_X$ $\in$ [NH$_4^+$, PO$_4^{3-}$, DON, DOP, PON, POP] *X $\in$ [N, P] | Element$_X$ uptake by heterotrophic bacteria | $Upt_{BAC_X}^{Element_X} = \mu_{MAX}^{BAC} * Q_{C,max}^{X} * \dfrac{Element_X}{Element_X + K_{Element_X}} * BAC_C * f_{Q_{10}}^{T}$ | mmol X m$^{-3}$ s$^{-1}$ |
| $Remin_{BAC_N}^{NH_4}$ | NH$_4^+$ remineralisation by heterotrophic bacteria | $Remin_{BAC_N}^{NH_4} = \sum\limits_{i=1}^{3} \left( Upt_{BAC_N}^{Element_{N_i}} * f_{Q_{10}}^{T} * (1 - f_{Q}^{U}) \right)$ ElementN $\in$ [NH$_4^+$, DON, PON] | mmol N m$^{-3}$ s$^{-1}$ |
| $Remin_{BAC_P}^{PO_4}$ | PO$_4^{3-}$ remineralisation by heterotrophic bacteria | $Remin_{BAC_P}^{PO_4} = \sum\limits_{i=1}^{3} \left( Upt_{BAC_P}^{Element_{P_i}} * f_{Q_{10}}^{T} * (1 - f_{Q}^{U}) \right)$ ElementP $\in$ [PO$_4^{3-}$, DOP, POP] | mmol P m$^{-3}$ s$^{-1}$ |
| $Mort_{BAC_X}^{DOX}$ *X $\in$ [C, N, P] | Natural mortality | $Mort_{BAC_X}^{DOX} = k_{mort} * BAC_X * f_{Q_{10}}^{T}$ | mmol X m$^{-3}$ s$^{-1}$ |

**Table C6: Biogeochemical processes simulated by Eco3M_MIX-CarbOx for dissolved inorganic matter (DIM)**

| Notation | Description | Formulation | Units |
|----------|-------------|-------------|-------|
| | | **DIM** | |
| $\text{Nitrif}_{NH_4}^{NO_3}$ | Nitrification | $\text{Nitrif}_{NH_4}^{NO_3} = tx_{NITRIF} * NH_4 * f_{Q_{10},nitrif}^T * \dfrac{O_2}{O_2 + K_{O_2}}$ | mmol N m$^{-3}$ s$^{-1}$ |
| $\text{Aera}^{DIC}$ | Aeration on DIC | $\text{Aera}^{DIC} = \dfrac{K_{ex}}{H} * \alpha * \left(pCO_{2,sea} - pCO_{2,atm}\right)$ | mmol C m$^{-3}$ s$^{-1}$ |
| $\text{Aera}^{O_2}$ | Aeration on O$_2$ | $\text{Aera}^{O_2} = \dfrac{K_{ex}}{H} * \left(DO_{sea} - DO_{atm}\right)$ | mmol O m$^{-3}$ s$^{-1}$ |
| $\text{Prec}_{DIC}^{CaCO_3}$ | CaCO$_3$ precipitation | $\text{Prec}_{DIC}^{CaCO_3} = \displaystyle\sum_{i=1}^{2} \left(\text{Photo}_{PHY_{C_i}}^{DIC} - \text{Resp}_{PHY_{C_i}}^{DIC}\right)$ $+ \displaystyle\sum_{i=1}^{2} \left(\text{Photo}_{MIX_{C_i}}^{DIC} - \text{Resp}_{MIX_{C_i}}^{DIC}\right) * f_{precip}$  *PHY $\epsilon$ [NMPHYTO, PICO] *MIX $\epsilon$ [NCM, CM] | mmol C m$^{-3}$ s$^{-1}$ |
| $\text{Diss}_{DIC}^{CaCO_3}$ | CaCO$_3$ dissolution | $\text{Diss}_{DIC}^{CaCO_3} = f_{diss}$ | mmol C m$^{-3}$ s$^{-1}$ |

## Appendix D: Detailed function formulation

**Table D1: Summary of functions formulations**

| Notation | Description | Formulation | Units |
|----------|-------------|-------------|-------|
| $f_Q^G$ | Growth quota function | $f_Q^G = \min\left(\dfrac{Q_c^N - Q_{c,min}^N}{Q_{c,max}^N - Q_{c,min}^N}, \dfrac{Q_c^P - Q_{c,min}^P}{Q_{c,max}^P - Q_{c,min}^P}\right)$ | ∅ |
| $f_Q^U$ | Uptake quota function | $f_Q^U = \min\left(1, \left(\dfrac{Q_{C,max}^X - Q_C^X}{Q_{C,max}^X - Q_{C,min}^X}\right)^n\right)$ | ∅ |
| $f_Q^N$ | Nitrogen quota function | $f_Q^N = \left(\dfrac{Q_c^N - Q_{c,min}^N}{Q_{c,max}^N - Q_{c,min}^N}\right)$ | ∅ |
| $f^T$ | Temperature function | $f^T = \dfrac{2*(1-\beta)*\dfrac{(T - T_{LET})}{(T_{OPT} - T_{LET})}}{\left(\dfrac{(T - T_{LET})}{(T_{OPT} - T_{LET})}\right)^2 + 2*(-\beta)\dfrac{(T - T_{LET})}{(T_{OPT} - T_{LET})} - 1}$ | ∅ |
| $f_{Q_{10}}^T$ | $Q_{10}$ temperature function | $f_{Q_{10}}^T = Q_{10}^{\frac{T-20}{10}}$ | ∅ |
| $f_{Q_{10},nitrif}^T$ | $Q_{10}$ temperature function for nitrification | $f_{Q_{10},nitrif}^T = Q_{10,nitrif}^{\frac{T-10}{10}}$ | ∅ |
| $f_{Inhib}^{CM}$ | CM grazing inhibition function | $f_{Inhib}^{CM} = \min\left(1 - \max\left(\dfrac{NO_3}{NO_3 + K_{NO_3}}, \dfrac{NH_4}{NH_4 + K_{NH_4}}\right), 1 - \dfrac{PO_4}{PO_4 + K_{PO_4}}\right)$ | ∅ |
| $P_{MAX}^C$ | Maximum photosynthesis rate | $P_{MAX}^C = P_{Ref}^C * f^T * f_Q^G$ | $s^{-1}$ |
| $limI$ | Light limitation function | $limI = 1 - \exp\left(\dfrac{-\alpha_{Chl} * Q_C^{Chl} * E_{PAR}}{P_{MAX}^C}\right)$ | ∅ |
| $\mu_{PPB}^{NR}$ | Nutrient replete photosynthesis rate | $\mu_{PPB}^{NR} = P_{Ref}^C * f^T * limI$ | $s^{-1}$ |
| $K_{ex}$ | Exchange coefficient | $K_{ex} = 0.251 * U_{10}^2 * \left(\dfrac{660}{Sc}\right)^{\left(\frac{1}{2}\right)}$ | cm h$^{-1}$ |
| $f_{precip}$ | CaCO$_3$ precipitation function | $f_{Precip} = K_{Precip} * \dfrac{\Omega-1}{K_C+\Omega-1}$ si $\Omega - 1 > 0$ <br> $f_{Precip} = 0$ si $\Omega - 1 < 0$ | ∅ |
| $f_{diss}$ | CaCO$_3$ dissolution function | $f_{Diss} = K_{Diss} * (1 - \Omega)$ si $\Omega - 1 < 0$ | $s^{-1}$ |

$$f_{Diss} = 0 \text{ si } \Omega - 1 > 0$$

| | | | |
|---|---|---|---|
| $\Omega$ | CaCO$_3$ saturation state | $\Omega = \dfrac{[CO_3^{2-}]_{mes} * [Ca^{2+}]_{mes}}{[CO_3^{2-}]_{sat} * [Ca^{2+}]_{sat}}$ | ø |

## Appendix E: Parameters descriptions, values, and units

Table E1 : Parameters values. (1) Campbell et al., 2013, (2) Stickney et al., 2000, (3) Auger et al., 2011, (4) Gaudy & Botha, 2007, (5) Banaru et al., 2019, (6) Leles et al., 2018, (7) Grosky et al., 1988, (8) Ghyoot et al., 2017, (9) Nielsen, 1997, (10) Thornley & Cannell, 2000, (11) Leblanc et al., 2018, (12) Sarthou et al., 2005, (13) Lacroix & Gregoire, 2002, (14) Lajaunie-Salla et al., 2021, (15) Tett, 1990, (16) Marty et al., 2002, (17) Gehlen et al., 2007, (18) Vrede et al., 2002, (19) Wanninkhof, 2014, (*) Calibrated.

| Notation | Description | Value | | Units | Reference |
|---|---|---|---|---|---|
| | | **COP** | **NCM** | | |
| $G_{MAX}$ | Maximum grazing rate | 1.296 | 3.024 | $d^{-1}$ | 1, 2* |
| $K_{PRED}$ | Grazing half-saturation constant | 20 | 8.5 | mol C m$^{-3}$ | 1, 3 |
| $frac_{resp}$ | Fraction of C allocated to respiration process | 0.27 | 0.27 | - | 4 |
| $frac_{MOD}$ | Fraction of N (P) released as MOD | - | 0.53 | - | 1 |
| $K_{mort}$ | Mortality rate | 0.033 | - | $d^{-1}$ | 1, 5 |
| $K_{mort,Chl}$ | Loss rate of captured chloroplasts | - | 0.4 | $d^{-1}$ | 6 |
| $Q_{C,min}^{N}$ | Minimum N:C ratio | 0.12 | 0.066 | mol N mol C$^{-1}$ | 7*, 1 |
| $Q_{C,max}^{N}$ | Maximum N:C ratio | 0.25 | 0.214 | mol N mol C$^{-1}$ | 7*, 1 |
| $Q_{C,min}^{P}$ | Minimum P:C ratio | 0.006 | 0.0037 | mol P mol C$^{-1}$ | 6 |
| $Q_{C,max}^{P}$ | Maximum P:C ratio | 0.016 | 0.0119 | mol P mol C$^{-1}$ | 6 |
| n | Curve shape factor | 2 | 2 | - | * |

**Constitutive mixotrophs (CM) and phytoplankton (NMPHYTO and PICO)**

| Notation | Description | CM | NM PHYTO | PICO | Units | Reference |
|---|---|---|---|---|---|---|
| $G_{MAX}$ | Maximum grazing rate | 2.160 | - | - | $d^{-1}$ | 2, 8 |
| $K_{PRED}$ | Grazing half-saturation constant | 5.0 | - | - | mol C m$^{-3}$ | 1 |
| $frac_{resp}$ | Fraction of C allocated to respiration process | 0.300 | 0.200 | 0.320 | - | 9, 10 |
| $cout_{resp}^{NO_3}$ | NO$_3^-$ respiration coast | 0.397 | 0.397 | 0.397 | - | 3 |
| $cout_{resp}^{NH_4}$ | NH$_4^+$ respiration coast | 0.198 | 0.198 | 0.198 | - | 3 |
| $cout_{resp}^{PO_4}$ | PO$_4^{3-}$ respiration coast | 0.350 | 0.350 | 0.350 | - | 11 |
| $\alpha_{Chl}$ | Chlorophyll-specific light absorption coefficient | $5.4\times10^{-6}$ | $3.83\times10^{-6}$ | $8.2\times10^{-6}$ | (mol C m$^{-2}$)(g Chl J$^{-1}$)$^{-1}$ | 11*, 6 |
| $P_{ref}^{C}$ | C-specific photosynthesis rate at temperature Tref | 1.55 | 1.05 | 1.81 | $d^{-1}$ | 12* |
| $\beta$ | Temperature curve shape factor | 0.6 | 0.8 | 0.5 | - | 13* |
| $T_{OPT}$ | Growth optimal temperature | 16.0 | 14.0 | 17.0 | °C | 1* |
| $T_{LET}$ | Lethal temperature | 10.0 | 9.0 | 11.0 | °C | 1* |
| $Q_{C,min}^{N}$ | Minimum N:C ratio | 0.100 | 0.050 | 0.115 | mol N mol C$^{-1}$ | 11 |
| $Q_{C,max}^{N}$ | Maximum N:C ratio | 0.215 | 0.170 | 0.229 | mol N mol C$^{-1}$ | 11 |
| $Q_{C,min}^{P}$ | Minimum P:C ratio | 0.0062 | 0.0031 | 0.0071 | mol P mol C$^{-1}$ | 11 |
| $Q_{C,max}^{P}$ | Maximum P:C ratio | 0.0130 | 0.0100 | 0.0143 | mol P mol C$^{-1}$ | 11 |

| Symbol | Description | | | | Units | Ref. |
|---|---|---|---|---|---|---|
| $K_{NO_3}$ | $NO_3^-$ half-saturation constant | 1.5 | 3.5 | 0.73 | mmol N m$^{-3}$ | 11 |
| $K_{NH_4}$ | $NH_4^+$ half-saturation constant | 0.12 | 0.18 | 0.07 | mmol N m$^{-3}$ | 11 |
| $K_{PO_4}$ | $PO_4^{3-}$ half-saturation constant | 0.008 | 0.01 | 0.005 | mmol P m$^{-3}$ | 1,*,14 |
| $K_{DON}$ | DON half-saturation constant | 1.5 | - | 0.85 | mmol N m$^{-3}$ | 11 |
| $K_{DOP}$ | DOP half-saturation constant | 0.155 | - | 0.085 | mmol P m$^{-3}$ | 11 |
| $Q_{Chl,min}^{N}$ | Minimum N:Chl ratio | 1.0 | 1.0 | 1.0 | mol N g Chl$^{-1}$ | 14 |
| $Q_{Chl,max}^{N}$ | Maximum N:Chl ratio | 2.55 | 3.0 | 2.2 | mol N g Chl$^{-1}$ | 11 |
| n | Curve shape factor | 1 | 1 | 1 | - | * |

**Heterotrophic bacteria**

| Symbol | Description | Value | Units | Ref. |
|---|---|---|---|---|
| bge | Bacteria growth efficiency | 0.8 | - | 1 |
| $Q_{10}$ | Temperature coefficient | 2.95 | - | 3 |
| $\mu_{MAX,NH_4}^{BAC}$ | Maximum rate of $NH_4^+$ uptake | 1.218 | d$^{-1}$ | 14 |
| $\mu_{MAX,PO_4}^{BAC}$ | Maximum rate of $PO_4^{3-}$ uptake | 1.209 | d$^{-1}$ | 14 |
| $\mu_{MAX,DOC}^{BAC}$ | Maximum rate of DOC uptake | 8.372 | d$^{-1}$ | 1 |
| $\mu_{MAX,DON}^{BAC}$ | Maximum rate of DON uptake | 1.218 | d$^{-1}$ | 14 |
| $\mu_{MAX,DOP}^{BAC}$ | Maximum rate of DOP uptake | 17.28 | d$^{-1}$ | 14 |
| $\mu_{MAX,POC}^{BAC}$ | Maximum rate of POC uptake | 0.665 | d$^{-1}$ | * |
| $\mu_{MAX,PON}^{BAC}$ | Maximum rate of PON uptake | 0.190 | d$^{-1}$ | 1 |
| $\mu_{MAX,POP}^{BAC}$ | Maximum rate of POP uptake | 0.359 | d$^{-1}$ | 1 |
| $K_{NH_4}$ | NH4+ half-saturation constant | 0.15 | mmol N m$^{-3}$ | 14 |
| $K_{PO_4}$ | PO43- half-saturation constant | 0.02 | mmol P m$^{-3}$ | 14 |
| $K_{DOC}$ | DOC half-saturation constant | 25.0 | mmol C m$^{-3}$ | 14 |
| $K_{DON}$ | DON half-saturation constant | 0.5 | mmol N m$^{-3}$ | 14 |
| $K_{DOP}$ | DOP half-saturation constant | 0.08 | mmol P m$^{-3}$ | 14 |
| $K_{POC}$ | POC half-saturation constant | 5.0 | mmol C m$^{-3}$ | * |
| $K_{PON}$ | PON half-saturation constant | 0.5 | mmol N m$^{-3}$ | 14 |
| $K_{POP}$ | POP half-saturation constant | 0.08 | mmol P m$^{-3}$ | 14 |
| $Q_{C,min}^{N}$ | Minimum N:C ratio | 0.168 | mol N mol C$^{-1}$ | 11,18 |
| $Q_{C,max}^{N}$ | Maximum N:C ratio | 0.264 | mol N mol C$^{-1}$ | 11, 18 |
| $Q_{C,min}^{P}$ | Minimum P:C ratio | 0.0083 | mol P mol C$^{-1}$ | 11, 18 |
| $Q_{C,max}^{P}$ | Maximum P:C ratio | 0.0278 | mol P mol C$^{-1}$ | 11, 18 |
| $K_{mort}$ | Mortality rate | 0.0432 | d$^{-1}$ | 13 |
| n | Curve shape factor | 1 | - | * |

**Dissolved inorganic matter**

| Symbol | Description | Value | Units | Ref. |
|---|---|---|---|---|
| $tx_{nitrif}$ | Nitrification rate | 0.050 | d$^{-1}$ | 13 |
| $K_{O_2}$ | Dissolved oxygen half-saturation constant | 30 | mmol O$_2$ m$^{-3}$ | 15 |
| $Q_{10,nitrif}$ | Temperature coefficient for nitrification | 2.37 | - | 3 |
| $K_{precip}$ | Fraction of PIC to LPOC | 0.02 | - | 16 |
| $K_c$ | CaCO$_3$ half-saturation constant | 0.4 | (µmol kg$^{-1}$)$^2$ | 16 |
| $K_{Diss}$ | Dissolution rate | 10.8 | d$^{-1}$ | 17 |

| | | | | Value | Units | Ref |
|---|---|---|---|---|---|---|
| $K_{ex}$ | Exchange coefficient | | | 0.251 | cm h$^{-1}$ m$^{-2}$ | 19 |
| H | Depth | | | 1 | m | - |
| $\left(\dfrac{O}{C}\right)_{PP}$ | Primary production O:C ratio | | | 1.10 | - | - |
| m1 | Fraction of the solar energy flux photosynthetically available | | | 0.43 | - | 15 |
| m2 | Sea surface reflection | | | 0.95 | - | 15 |
| m3 | More rapid attenuation of polychromatic light near the sea surface | | | 0.75 | - | 15 |

**Table E2: Predator preference for their preys (COP: copepods, NMPHYTO: nano+micro-phytoplankton, PICO: picophytoplankton and BACT: heterotrophic bacteria). (20) Verity and Paffenhofer (1996), (21) Price & Turner, 1992, (22) Christaki et al., 2009, (23) Epstein et al., 1992, (24) : Christaki et al., 2002, (25) Zubkhov & Tarron, 2008, (26) Millette et al., 2017, (27) Livanou et al., 2019, (\*) Calibrated.**

| | | PREYS | | | | | References |
|---|---|---|---|---|---|---|---|
| | | NCM | CM | NMPHYTO | PICO | BAC | |
| | COP | 0.4 | 0.25 | 0.35 | | | 20, * |
| PRED | NCM | | 0.20 | 0.15 | 0.25 | 0.40 | 21, 22, 23, * |
| | CM | | | | 0.35 | 0.65 | 24, 25, 26, 27 * |

 **Appendix F: Yearly mean values of photosynthesis and grazing for NCM and CM properties verification simulations**

**Table F1: Yearly mean values of grazing and photosynthesis for NCM and CM properties verification simulations (Table 2).**

| NCM | | |
|---|---|---|
| **Simulation** | **Yearly mean grazing (mmolC m$^{-3}$ s$^{-1}$)** | **Yearly mean photosynthesis (mmolC m$^{-3}$ s$^{-1}$)** |
| **NCM-Replete** | $5.16 \times 10^{-6}$ | $2.35 \times 10^{-6}$ |
| **NCM-Low Nut** | $5.16 \times 10^{-6}$ | $2.35 \times 10^{-6}$ |
| **NCM-Low Food** | $1.50 \times 10^{-6}$ | $9.54 \times 10^{-7}$ |
| **NCM-Replete Constant** | $7.60 \times 10^{-7}$ | $1.12 \times 10^{-6}$ |
| **NCM-Low light Constant** | $7.60 \times 10^{-7}$ | $3.70 \times 10^{-7}$ |
| **CM** | | |
| **Simulation** | **Yearly mean grazing (mmolC m$^{-3}$ s$^{-1}$)** | **Yearly mean photosynthesis (mmolC m$^{-3}$ s$^{-1}$)** |
| **CM-Replete** | $3.67 \times 10^{-8}$ | $8.81 \times 10^{-6}$ |
| **CM-Low Nut** | $2.02 \times 10^{-7}$ | $1.18 \times 10^{-6}$ |
| **CM-Low Light** | $1.00 \times 10^{-9}$ | $2.70 \times 10^{-7}$ |
| **CM-Low Food** | $1.60 \times 10^{-8}$ | $7.60 \times 10^{-6}$ |

## Appendix G: Statistical analysis

We calculated three statistical indicators for the comparison between modelled chlorophyll and chlorophyll measurements performed at SOLEMIO station: the percent bias (%BIAS), the cost function (CF) and the root mean square deviation (RMSD).

%BIAS is calculated according to Allen et al. (2007). A positive %BIAS means that the model underestimated the in situ observations and vice versa. We interpreted %BIAS according to Marechal (2004) ( excellent if %BIAS < 10 %, very good

if 10 % ≤ %BIAS < 20 %, good if 20 % ≤ %BIAS < 40 % and poor otherwise). We use the absolute values of %BIAS, to assess the overall agreement between the model results and observations.

The cost function is calculated based on Allen et al. (2007). According to Radach and Moll (2006), CF < 1 is considered very good, 1 ≤ CF < 2 is good, 2 ≤ CF < 3 is reasonable, while CF ≥ 3 is poor.

RMSD quantifies the difference between model results and observations (Allen et al., 2007). The closer RMSD is to 0, the

795 more reliable the model.

**Table G1: Statistic indicator calculated for observed and modelled chlorophyl.**

|  | Model | Observations |
|---|---|---|
| **Mean (mg Chl m$^{-3}$)** | 0.40 | 0.49 |
| **Range of values (mg Chl m$^{-3}$)** | [0.08 ; 0.90] | [0.1 ; 1.71] |
| **Standard deviation (mg Chl m$^{-3}$)** | 0.21 | 0.33 |
| **CF** | 0.85 | |
| **RMSD (mg Chl m$^{-3}$)** | 0.41 | |
| **%BIAS (%)** | -1.33 | |

**Appendix H: User manual**

The version of Eco3M_MIX-CarbOX used in this article can be downloaded from the Zenodo website (https://zenodo.org/record/7669658#.Y_dAJ0NKg2w, last access: 23 February 2023, Barré Lucille, Diaz Frédéric, Wagener Thibaut, Van Wambeke France, Mazoyer Camille, Yohia Christophe, & Pinazo Christel. (2022). Eco3M_MIX-CarbOx (v1.0). Zenodo. https://doi.org/10.5281/zenodo.7669658). To run Eco3M_MIX-CarbOX, the whole archive must be uploaded.

-   Time, time step and save time of simulated state variables can be defined in the file config.ini (path: MIX-CarbOx_0D_v1.0/BIO/).

      -   Boundary conditions, initial conditions values of state variables and forcing data are stocked in DATA directory (path: MIX-CarbOx_0D_v1.0/BIO/DATA/)

      -   Biogeochemical processes formulations are stocked in F_PROCESS directory (path: MIX-
CarbOx_0D_v1.0/BIO/F_PROCESS/).

      -   Results files and MALTAB routines to visualize them are stocked in SORTIES directory (path: MIX-CarbOx_0D_v1.0/BIO/SORTIES/).

      To run Eco3M_MIX-CarbOx v1.0 :

      gmake !This command creates two executable files : eco3M_ini.exe and eco3M.exe.

For further information, please contact Lucille Barré (lucille.barre@mio.osupytheas.fr).

## Code availability

The current version of Eco3M_MIX-CarbOx is available from the Zenodo website (https://zenodo.org/record/7669658#.Y_dAJ0NKg2w, last access: 23 February 2023) under the Creative Commons Attribution 4.0 international licence. The exact version of the model used to produce the results in this paper is archived on Zenodo (Barré Lucille, Diaz Frédéric, Wagener Thibaut, Van Wambeke France, Mazoyer Camille, Yohia Christophe, & Pinazo Christel. (2022). Eco3M_MIX-CarbOx (v1.0). Zenodo. https://doi.org/10.5281/zenodo.7669658) as are input data and scripts to run the model and produce the plots for all the simulation presented in this paper.

## Data availability

Surface total chlorophyll concentration data are available on request on https://www.somlit.fr/. Temperature data is available on www.t-mednet.org by filling out the request form for station and years pre-selected. Salinity data is available on https://erddap.osupytheas.fr. The non-processed atmospheric $p$CO$_2$ data can be found on https://servicedata.atmosud.org/donnees-stations. Request for processed atmospheric $p$CO$_2$ data should be addressed to alexandre.armengaud@airpaca.org and irene.xueref-remy@imbe.fr.

## Author contribution

LB conceptualized this study, developed the Eco3M_MIX-CarbOx model v1.0, designed the numerical experiments, developed MATLAB software to visualize and process the model results, processed, and analysed the model results, wrote the initial draft, revised the manuscript, and answered reviewers. FD provided the initial version of the model code (without carbonate module and with an initial implementation of the mixotrophs) and helped to develop the Eco3M_MIX-CarbOx v1.0. CP acquired the fundings, participated to the conceptualization of this study and supervised it, participated to the model development, designed the numerical experiments, analysed the model results, reviewed, and edited the initial draft and helped to answer reviewers' comments. FvW helped to design the numerical experiments and with the analysis of model results, reviewed and edited the initial draft. CM helped in the model development process by giving expertise on the code development to reduce calculation time. CY provided the wind and irradiance data, maintained computing resources. TW participated to the conceptualization of this study, helped to design the numerical experiments, analysed the model results, reviewed, and edited the initial draft and helped to answer reviewers' comments.

## Competing interests

The authors declare that they have no conflict of interest.

**Acknowledgements**

We thank the National Service d'Observation en MILieu Littoral (SOMLIT) for its permission to use SOLEMIO data. We would like to thank the crew members of the RV Antedon II, operated by the DT-INSU, for making these samplings possible, the team of the SAM platform (Service Atmosphère Mer) of the MIO for help with the field work. We also thank Michel Lafont and Véronique Lagadec of the PACEM (Plateforme Analytique de Chimie des Environnements Marins) platform of the MIO. We acknowledge the TMEDNet team for its permission to use the Planier-Souquet temperature data.

We thank the ROMARIN network team for its permission to use the salinity data from Carry buoy. We thank the observatoire de la qualité de l'air en Région Sud Provence-Alpes-Côte d'Azur (ATMOSUD) in particular, Alexandre Armengaud, and the AMC (Aix-Marseille Carbon Pilot Study) project leaders, Irène Xueref-Remy and Dominique Lefèvre for providing the atmospheric $CO_2$ data at the Cinq Avenue station. We acknowledge the staff of the "Cluster de calcul intensif HPC" platform of the OSU Institut PYTHEAS (Aix–Marseille Université, INSU-CNRS) for providing the

computing facilities. We would like to thank Julien Lecubin from the Service Informatique de l'OSU Institut Pytheas for its technical assistance. We thank XpertScientific team for the manuscript correction. We thank the two anonymous reviewers for their helpful comments to improve this paper.

**Fundings**

This work takes part of the IAMM project (Évaluer l'Impact de la métropole Aix-Marseille sur l'Acidification de la baie de

Marseille et les conséquences sur les microorganismes marins, approche par Modélisation) funded by the public establishment of the Ministry of the Environment, l'Agence de l'eau Rhône Mediterranée Corse.

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
