# Peer review of "Implementation and assessment of a model including mixotrophs and the carbonate cycle (Eco3M\_MIX-CarbOx v1.0) in a highly dynamic Mediterranean coastal environment (Bay of Marseille, France) (Part I): Evolution of ecosystem composition under limited light and nutrient conditions"

_Geoscientific Model Development, 2023_

## Author Comment (AC1)

**Referee #1**

First, we would like to thank Referee #1 for his/her careful evaluation of our manuscript. We believe that his/her comments will help to improve the manuscript. Please, find hereafter our responses to the concerns raised by Referee #1.

**Specific comments**

**Lines 23-26: What the authors mean with the phrase "portion of the ecosystem"? Is it in terms of carbon? This needs to be rephrased appropriately. Also, the phrase "relatively high carbon biomass" could be more specific, i.e. relative to what and how much?**

We agree with this comment. To clarify, we changed the phrase :

[In addition, we investigate the carbon, nitrogen and phosphorus fluxes associated with mixotrophic protists and showed that: (i) the portion of the ecosystem occupied by NCM decreases when resources (nutrient and prey concentrations) decrease, although their mixotrophy allows them to maintain a relatively high carbon biomass as photosynthesis increase as food source; (ii) the portion of the ecosystem occupied by CM increases when nutrient concentrations decrease, due to their capability to ingest prey to supplement their N and P needs.]

to:

[In addition, we investigate the carbon, nitrogen and phosphorus fluxes associated with mixotrophic protists and showed that: (i) the portion of the ecosystem in percentage of carbon biomass occupied by NCM decreases when resources (nutrient and prey concentrations) decrease, although their mixotrophy allows them to maintain a carbon biomass almost as significant as the copepods one (129.8 and 148.7 mmolC m$^{-3}$, respectively), as photosynthesis increase as food source; (ii) the portion of the ecosystem in percentage of carbon biomass occupied by CM increases when nutrient concentrations decrease, due to their capability to ingest prey to supplement their N and P needs.].

**Lines 47-48: "Studies are … heterotrophs". This sentence needs rephrasing. What's the message here? That there are fewer modelling studies compared to experimental ones?**

Yes, we wanted to point out that, as a lot of models still consider a food web divided into strict phototrophs and heterotrophs, there are fewer modelling studies comparing to experimental ones. We modified it to avoid confusion (l.48-54).

[Mixotrophic protists played an important role in the marine carbon cycle. Due to their adaptability, these organisms are crucial for the transfer of matter and energy to the highest trophic levels, thus impacting the structure of planktonic communities by favouring the development of larger organisms (Ptacnick et al., 2004). Moreover, by switching the biomass maximum to larger organisms, carbon export increases in presence of mixotrophs. As instance, Ward and Follows (2016) compared the results from two food web models, only one accounted for mixotrophy, and showed that carbon export to depth increased by nearly 35% when mixotrophic protists were considered. By showing the significant effect of mixotrophic protists on the food web, these studies motivated their addition to current food web models (Jost et al., 2004; Mitra and Flynn, 2010).]

**Line 65: "Unlike most other models..." There are also a lot of models that use variable stoichiometry so this phrase is somewhat misleading and should be rephrased.**

We modified (l.68-69) :

[Unlike most other models, Eco3m_MIX-CarbOx uses variable cellular quotas which allowed us to determine the nutritional state of the cell by comparing it to a reference quota]

to:

[Eco3m_MIX-CarbOx uses variable cellular quotas which allowed us to determine the nutritional state of the cell by comparing it to a reference quota].

**Figure 2: The explanation of the abbreviation TA is missing.**

Thank you for pointing this out, we added the explanation of the abbreviation TA.

**In addition, it appears strange that there is no heterotrophic protists compartment in the model. Heterotrophic protists are an important component of the marine microbial food web. Thus, the authors should comment on why they chose not to include them in their model formulation.**

We agree that heterotrophic protists could play an important role in the food web assemblage. We chose to not consider them because, by adding mixotrophs, we have considerably complexified the architecture of the model. To simplify the transition to 3D (coupling to a hydrodynamic model) and to limit the calculation time, we decided not to consider them. We propose to discuss it in a new section : 4.1 Mixotrophs representation assessment; as a possible improvement of the 0D model in the discussion (l.501-505).

[In the present model, we do not consider strict heterotrophs which belong to the nano and micro size classes. These organisms can be important competitors of ciliates, and certain species can even consume ciliates (Stoecker and Capuzzo, 1990 ; Johansson et al., 2004). The adding of these organisms could improve the representation of NCM dynamics and, accordingly, of the ecosystem and is then considered for an improved version of the model.]

**Moreover, I believe that some explanation should be provided regarding the fact that there is no mortality for phytoplankton and mixotrophs but there is mortality for bacteria.**

We add a point about mortality of NCM in the discussion (l.505-507). We think that adding mortality could improve the representation of NCM dynamics, and we consider it for an improved version of the model.

[Moreover, we do not consider a mortality term for NCM. Montagnes (1996) showed that mortality rates for two species of the genus *Strombidium* and two species of the genus *Strombilidium* were rapid. Accordingly, adding this term to the model could allow to represent a more realistic NCM biomass.]

For phytoplankton and CM, we performed sensitivity tests to add a mortality term when developing the model, but none were conclusive as our phytoplankton compartment and CM variable was already balanced. We believe that this is due to the fact that predators exert a strong top-down control on phytoplankton and CM populations.

**Lines 112-114: To the best of my knowledge the Baklouti et al. 2006 a, b papers present the formulations only for the phytoplankton compartment. For zooplankton the authors provide the reference of Auger et al. (2011). However, some background information on the formulation of bacteria compartment is missing. I advise the authors to provide this background information and the relevant references on which they have based the formulation for bacteria.**

We added the references for heterotrophic bacteria formulation (Kirchman, 2000 ; Faure et al., 2006) (l.169).

**Line 130: Why copepods feed with different preference on nanophytoplankton and CM? Most of the CM belong to nanoflagellates so I don't see why copepods prefer one group more than the other.**

To clarify this, we add Figure 3 which illustrate organisms' repartition in size classes and trophic interactions between them. In our model, the variable NANO aims to represent the phytoplankton

larger than 2 µm and smaller than 2000 µm (nanophytoplankton and microphytoplankton). This variable stands for diatoms, autotrophic dinoflagellates. In the northwestern Mediterranean Sea, diatoms are an important component of phytoplankton assemblage especially during the spring bloom (Margalef, 1978, Leblanc et al., 2018) and cover wide size-range, we decided to consider them as representative of the variable. To avoid confusion, we switch the name NANO to NMPHYTO (for nano+micro-phytoplankton).

We modified the lines 136 to 141:

[We considered two types of phytoplankton based on size: nanophytoplankton (NANO) and picophytoplankton (PICO). Nanophytoplankton includes autotrophic flagellates and small diatoms. We used *Minidiscus spp*. as the representative species of nanophytoplankton as the *minidiscus* genus proliferates throughout the NW Miterranean when light and nutrients are less limiting (Leblanc et al., 2018). Picophytoplankton includes autotrophic prokaryotic organisms such as *Prochlorococcus spp*. and *Synechococcus spp*. The *Synechococcus* genus is ubiquitous in the Mediterranean (Mella-flores et al., 2011) and was therefore considered the representative genus of picophytoplankton in the model.]

to:

[We considered two types of phytoplankton based on size (Fig. 3): picophytoplankton (PICO) and nano+micro-phytoplankton (NMPHYTO). PICO includes autotrophic prokaryotic organisms such as Prochlorococcus spp. and *Synechococcus spp* which are ubiquitous in the Mediterranean (Mella-flores et al., 2011). NMPHYTO aims to represent phytoplankton larger than 2 µm and smaller than 200 µm. It mainly includes diatoms and autotrophic nanoflagellates. As diatoms are an important component of Mediterranean spring blooms (Margalef, 1978, Leblanc et al., 2018) and cover wide size-range, we decided to consider them as representative of the NMPHYTO]

and the rest of the manuscript accordingly.

Accordingly, we assumed that copepods feed on strictly smaller sized preys with the strongest preference for the largest organism: NCM. NCM are considered to be ciliates, copepods preferential ingestion of ciliates in environments where diatoms and dinoflagellate are present has been demonstrated by Verity (1996). They also represent a preferential food source for the reproduction of some copepod species (Dutz & Peters, 2008). Next, we decided to apply a strongest preference on diatoms as they cover a largest size range with possibly bigger organisms than CM.

**Line 145: Why only PICO can consume DON and DOP but not PHYTO. As far as I know most phytoplankton species can consume DON and DOP. Therefore, the authors should provide some justification on this assumption. Moreover, dissolved organic phosphorus and nitrogen should be stoichiometrically coupled with dissolved organic carbon, accounting of course for variable stoichiometry of dissolved organic matter. That is PICO as well as heterotrophic bacteria and mixotrophs will consume a mol of dissolved organic matter containing x mols of DON and y mols of DOP, i.e. by definition DON and DOP is coupled to organic carbon. The way that the consumption of dissolved organic matter is formulated in the current model does not account for the uptake of dissolved organic carbon along with DON and DOP.**

Thank you for this interesting comment. We decided to consider osmotrophy only for the smaller organisms (PICO and CM, Duhamel et al. (2018) ; Glibert and Legrand, (2006)). However, we agree that recent studies show that some diatoms (which are representative of NMPHYTO in our model) are able to perform dissolved organic compounds uptake (Villanova and Spetea, 2021). Then, we will consider it as possible improvement for next versions of the model.

We agree with the fact that DON, DOP and DOC are coupled, as an example we consider the three uptakes for heterotrophic bacteria. In fact, we decided to add only DON and DOP uptake for CM and PICO because we assumed that, as a type of mixotrophy, these uptakes are done to supplement N and P needs when NO3-, NH4+ and PO43- are limiting the growth. We represent it by limiting these uptakes

by nutrients concentration (by considering the internal content of the cell in N and P). When N (P) content of the cell is high the uptake is close to 0 and vice versa. In the model, carbon is entirely provided by photosynthesis and organisms are rarely limited by this element which explain that we do not consider DOC uptake.

**Line 185: Why the NCM as well as the CM have different preferences for the different types of prey. This is somewhat arbitrary and some support for this assumption should be provided.**

For NCM: We made the choice to prioritize the ingestion of smaller organisms by considering bacteria and picophytoplankton as the preys with the highest preference as it is the case for small ciliates (Rassoulzadegan et al. 1988, Price & Turner, 1992, Christaki et al., 1999). We then prioritize nanophytoplankton which has been shown to be a great food source for ciliates in the Gulf of Lion (Christaki et al., 2009). We apply the lowest preference to NMPHYTO as they cover a wide range of size (they can be as large as NCM) and species including diatoms which are associated with smaller ciliates grazing rates (Epstein et al., 1992).

For CM: We assumed that CM consume strictly smaller sized preys. They are known to consume bacteria and picophytoplankton (Christaki et al., 2002 ; Zubkhov & Tarron, 2008, Millette et al., 2017, Livanou et al., 2019).

We provided references in the text (l.204 and l.260), the caption of the new figure 3:

[From most to least preferred prey, NCM feed on heterotrophic bacteria, picophytoplankton, CM and nano+micro-phytoplankton (Verity, 1991 ; Price & Turner, 1992 ; Christaki, 1999).]

[CM feed on heterotrophic bacteria (preferred) and picophytoplankton (less preferred, Christaki et al., 2002 ; Zubkhov & Tarron, 2008, Millette et al., 2017 ; Livanou et al., 2019) and the same grazing formulation as for zooplankton and NCM is used except that CM grazing is limited by DIN (DIP) concentration and light (Stoecker, 1997, 1998; Eq. 9).]

[Figure 3: Repartition of modelled organisms (COP: copepods, PICO: picophytoplankton, NMPHYTO: nano+micro-phytoplankton, and BACT: heterotrophic bacteria) in size classes and trophic interactions between them. Preference values are indicated in grey for copepods (Verity and Paffenhofer, 1996) and NCM (Epstein, 1992; Price & Turner, 1992 ; Christaki, 2009) and CM (Christaki et al., 2002 ; Zubkhov & Tarron, 2008, Millette et al., 2017 ; Livanou et al., 2019).]

and added a reference column to the Table E2.

**Line 210: It is not clear to me why the photosynthetic flux is weighted by the prey preference. Shouldn't it be analogous to the grazing rate to each of the different prey since prey should be first consumed and then it can be used for photosynthesis by the mixotroph?**

In the present formulation we do not directly consider the ingested chlorophyl for each prey. It is another possibility to represent this process. We chose the present formulation as it was less complex and more adapted to our model. However, we would like to point out that the present formulation considers the benefits of grazing through the calculation of a quota function ($f_{Q,NCM}^{G}$). Prey dependence is then added through photosynthesis calculation parameters (i.e., we used prey parameters to calculate temperature ($f^{T}$) and light limitation (limI) functions and based the calculation of the maximum photosynthetic rate on the C-specific photosynthetic rate of the prey at a reference temperature ($P_{REF,PREY}^{C}$)) and the portion of each photosynthetic prey through their associated preference value.

**Lines 252-253: Authors should provide further information regarding this statement. To what evidence is this assumption based on?**

It is not a formulation choice but a result from the CM grazing representation. When DIN (DIP) concentration is limiting CM will ingest prey in addition to the uptake of nutrient. As their internal content in N (P) is particularly low, exudation of DON (DOP) is not allowed (equal to 0). When DIN (DIP) concentration is high, CM only perform nutrient uptake (no grazing as it only supplements N and P needs in limiting conditions). Then, all the N (P) from uptake is exuded as the cell is already loaded in N (P) and as no grazing is performed, no N (P) from grazing is exuded in these conditions.

We changed (l.277):

[The formulations for DON and DOP exudation are similar except neither N nor P obtained from grazing are released, only N and P obtained from nutrient uptake if the cell's N and P content is high are released. Respiration uses the same formulation as for phytoplankton i.e., a constant fraction of photosynthesis and nutrient uptake is respired (Section 2.2.2 and Appendix C).]

to:

[The formulations for DON and DOP exudation are similar. Exudation only occurs on the N and P obtained from nutrient uptake. In other words, neither N or P obtained from grazing are released through exudation. When DIN (DIP) concentration is limiting CM will ingest prey in addition to the uptake of nutrient. As their internal content in N (P) is particularly low, exudation of DON (DOP) is not allowed (equal to 0). When DIN (DIP) concentration is high, CM only perform nutrient uptake (no grazing as it only supplements N and P needs in limiting conditions). Then, all the N (P) from uptake is exuded as the cell is already loaded in N (P) and as no grazing is performed, no N (P) from grazing is exuded in these conditions.]

**Lines 431-433: Mixotrophic organisms must invest in the synthesis and maintenance of both a phototrophic and a phagotrophic apparatus, which can lead to an increased metabolic cost. Trade-offs of mixotrophy are not taken into account in the current model, however the authors should comment on the potential effect of trade-offs of mixotrophy on the competitive advantage of mixotrophs.**

We thank the referee for this interesting comment. We reorganised the last point of our discussion to included it (l.635-647):

[In the present work, we provided a relatively simple model (reduced number of compartments, 0D reasoning) to represent mixotrophy in the BoM. Even though we showed that we reproduced well the two types of mixotrophs modelled (all properties from Stoecker, 1998 were verified), Eco3M_MIX-CarbOx could still be improved. When developing Eco3M_MIX-CarbOx, we considered a simplify food web with a reduced number of compartments, consequently we made the choice to not consider strict heterotrophs which belong to the nano and micro size classes. This choice can affect the representation of NCM biomass as these organisms are known to compete with ciliates for resources. Some species can even ingest ciliates (Stoecker and Capuzzo, 1990 ; Johansson et al., 2004). Moreover, in the current version of the model, we do not take into account the possible increasing metabolic cost associated with mixotrophy (i.e., maintenance of both autotrophic and heterotrophic apparatus). Raven (1997) shown that the cost of maintaining phagotrophic apparatus for a primarily phototrophic organism remains low, but the cost of maintaining a phototrophic apparatus for a primarily phagotrophic organism can be significant and often resulting in lower growth rates than strict heterotrophs. It might be interesting to consider it as it could improve the representation of the NCM biomass.]

**Lines 454-455: "Although CM biomass… not co-limiting": These findings should be further explained. i.e. What are the model assumptions and formulations that give rise to these results?**

These results are explained by the high nutrient concentration applied to this simulation and the parameters used to calculate light limitation for each organism. By lifting the nutrient limitation, NMPHYTO which is particularly sensitive to nutrient concentration, can grow more easily. In addition,

NMPHYTO include mainly diatoms which are known to be advantaged in low light environment (Fisher and Halsey, 2016) we then chose the parameters for light limitation calculation accordingly. Due to the chosen parameters, CM are more affected by low light, in addition they do not perform grazing in these conditions as nutrient concentration is high.

We modified:

[Although CM biomass remains high in low light, its share of the pie decreases in favour of NANO which seem to gain a slight edge. While the share of NANO increases slightly under low light PICO appears to be unaffected which is in agreement with observations by Timmermans et al. (2005) for when nutrients are not co-limiting (Fig. 9a, b).]

to (l.545):

[Although CM biomass remains high in low light, its share of the pie decreases in favour of NANO which seem to gain a slight edge. While the share of NANO increases slightly under low light PICO appears to be unaffected (Fig. 10a, b). In this simulation nutrient levels were kept artificially high to prevent nutrient limitation. By lifting the nutrient limitation NMPHYTO which is particularly sensitive to nutrients concentration, can grow more easily. In addition, NMPHYTO includes mainly diatoms which are known to be advantaged in low light environment (Fisher and Halsey, 2016). CM are more affected by low light and are not able to use mixotrophy in these conditions (nutrient concentration is high). The low effect of light on PICO agrees with observations by Timmermans et al. (2005) who showed that when nutrients are not co-limiting picophytoplankton still developed well.]

**Technical corrections:**

Thank you for this, we took into account all these corrections.

**Figures :**

[Figure]

**Figure 3**: Repartition of modelled organisms (COP: copepods, PICO: picophytoplankton, NMPHYTO: nano+micro-phytoplankton and BACT: heterotrophic bacteria) in size classes and trophic interactions between them. Preference values are indicated in grey for copepods (Verity, 1996) and NCM (Epstein, 1992; Price & Turner, 1992 ; Christaki, 2009) and CM (Christaki et al., 2002 ; Zubkhov & Tarron, 2008, Millet et al., 2017 ; Livanou et al., 2019). **[added to the manuscript]**

**Tables:**

**Table E2**: Predator preference for their preys (COP: copepods, NMPHYTO: nano+micro-phytoplankton, PICO: picophytoplankton and BACT: heterotrophic bacteria). (20) Verity and Paffenhofer (1996), (21) Price & Turner, 1992, (22) Christaki et al., 2009, (23) Epstein et al., 1992, (24) : Christaki et al., 2002, (25) Zubkhov & Tarron, 2008, (26) Millette et al., 2017, (27) Livanou et al., 2019, (*) Calibrated.

|      |      | PREYS | | | | | References |
|------|------|-----|------|---------|------|------|------------|
|      |      | NCM | CM | NMPHYTO | PICO | BACT |            |
|      | COP  | 0.4 | 0.25 | 0.35  |      |      | 20, *      |
| PRED | NCM  |     | 0.20 | 0.15  | 0.25 | 0.40 | 21, 22, 23, * |
|      | CM   |     |      |       | 0.35 | 0.65 | 24, 25, 26, 27 * |

---

## Author Comment (AC2)

**Referee #2**

First, we would like to thank Referee #2 for his/her careful evaluation of our manuscript. We believe that his/her comments will help to improve the manuscript. Please, find hereafter our responses to the concerns raised by Referee #2.

**General comments**

**1a) The manuscript lacks key sensitivity analyses.**

We made several sensitivity analyses during the model development stage. These analyses allowed us to test the sensitivity of the model to the parameters to choose the best values for our modelled organisms. As instance we made several tests for the following parameters:

- Test of different values of bacterial growth efficiency (bge) from low (0.4) to high (0.8),
- Test of combinations of $Q^N_{C,min}$, $Q^N_{C,max}$, $Q^P_{C,min}$, $Q^P_{C,max}$ (min and max values for N:C and P:C ratio) for all the modelled organisms,
- Test of combinations of half saturation constants of NMPHYTO, CM, PICO and heterotrophic bacteria for nutrients, MOD and MOP uptake,
- Test of different values of bacteria maximum rates of nutrients, MOD and MOP uptake ($\mu^{BAC}_{MAX,X}$)
- Test of different values of nitrification and remineralisation optimal temperature,
- Test of different values of optimal growth temperature of CM,
- Test of different combinations of prey preferences for copepods, NCM and CM,
- Test of different values of bacteria and copepods mortality and predation rates

They also allowed us to compare formulations (Two types of photosynthesis formulations (Platt et al., 1980, Geider et al., 1998) and their associated parameters for CM, NMPHYTO and PICO) and configurations (without mixotrophs by removing them from the present model configuration to compare the ecosystem composition in C biomass and net community production (NCP), results shown in Fig. S1 for this last test). We made the choice to not present them in the manuscript as we wanted our manuscript to answer the questions: "Does mixotrophy represent and advantage for the organisms in the Bay of Marseille (BoM) ? and if it is the case: "When will mixotrophs be advantaged over other organisms (e.g., conditions, specific events) ?" and not only focus on the model development.

**1b) To be able to say that mixotrophy has an important impact on system dynamics, one would need to compare it against versions of the model in which no mixotrophs are considered. Specifically, mixotroph types should be replaced with purely phototrophic (phytoplankton) or heterotrophic (microzooplankton) types.**

We compared the results of a configuration of our model without mixotrophs (Fig. S2). In this configuration, CM are replaced by strict autotrophs (NANOP, phytoplankton between 2 and 20 μm) and NCM are replaced by strict heterotrophs (MICROZ, zooplankton between 20 and 200 μm). As the nanophytoplankton is now represented by NANOP variable, we represented the phytoplankton between 20 and 200 μm by the MICROP variable (Fig. S3). Balance equations for MICROZ and NANOP are presented in Table S1. The balance equations for MICROP remain unchanged.

We ran a first simulation which reproduces SOLEMIO conditions (same properties as typical simulation, Table 4, Table 5 now) and, as we showed that mixotrophy occurred mainly in nutrient limited conditions, we ran a second simulation which reproduces nutrient limited conditions (same properties as nutrient limited simulation, Table 4). We supposed that adding mixotrophs to our model will result in a modification of the ecosystem composition, and especially in an increase of the portion occupied

by the largest organisms (NCM and copepods in our model) as shown by Ward & Follows (2016). We then compared the configurations by studying the ecosystem composition in percentage of C biomass (Fig. S4) and the percentage of each prey in total copepod grazing (Table S2). Next, we provide predation on copepods which is an indicator of the quantity of C transferred to the higher trophic levels, and total photosynthesis and respiration fluxes (Table S3).

Results show that ecosystem composition are significantly affected by the consideration of mixotrophs. Larger organisms (micro and mesozooplankton) represent a larger part of the ecosystem when we consider mixotrophs, in both types of conditions tested (Fig. S4). Moreover, these results illustrate the advantage of mixotrophy for CM as when nutrients limit the organisms' growth, CM and PICO dominate phytoplankton assemblage by occupying the same portion of the ecosystem however, when mixotrophy is not considered, PICO only dominate the assemblage of phytoplankton. Similarly, table S2 showed that microzooplankton (NCM vs MICROZ) represents a larger percentage of total grazing.

Considering mixotrophy also have an impact on carbon fluxes, as shown by Table S3: photosynthesis and respiration are lower when mixotrophy is considered (as also modelled by Leles et al., 2018), but predation flux on copepod which can be used as an indicator of the quantity of C transferred to higher trophic levels, is higher (as also modelled by Ward & Follows, 2016).

We made these analyses, but we decided to not add them to the manuscript as we think that, even if interesting, they are not what we want to focus on. The aim here is to answer the questions: "Does mixotrophy represent and advantage for the organisms in the BoM ? and if it is the case: "When will mixotrophs be advantaged over other organisms (e.g., conditions, specific events in the BoM) ?". By comparing simulations ran in different environmental conditions, we manage to answer these questions.

**1b) This leads to a second complication: the model does not represent micro- phytoplankton nor zooplankton (see next comment); the authors should be clear about this assumption (that all microplankton are considered to be mixotrophs) and the implications related to this needs to be further explored.**

To clarify this, we add Figure 3 which illustrates organisms' repartition in size classes and trophic interactions between them. In our model, the variable NANO aimed to represent the phytoplankton larger than 2 µm and smaller than 2000 µm (nanophytoplankton and microphytoplankton). This variable stands for diatoms, autotrophic dinoflagellates. In the northwestern Mediterranean Sea, diatoms are an important component of phytoplankton assemblage especially during the spring bloom (Margalef, 1978, Leblanc et al., 2018) and cover wide size-range, we decided to consider them as representative of the variable. To avoid confusion, we switch the name NANO to NMPHYTO (for nano+micro-phytoplankton). We modified the lines 136 to 141:

[We considered two types of phytoplankton based on size: nanophytoplankton (NANO) and picophytoplankton (PICO). Nanophytoplankton includes autotrophic flagellates and small diatoms. We used *Minidiscus spp*. as the representative species of nanophytoplankton as the *minidiscus* genus proliferates throughout the NW Miterranean when light and nutrients are less limiting (Leblanc et al., 2018). Picophytoplankton includes autotrophic prokaryotic organisms such as *Prochlorococcus spp*. and S*ynechococcus spp*. The *Synechococcus* genus is ubiquitous in the Mediterranean (Mella-flores et al., 2011) and was therefore considered the representative genus of picophytoplankton in the model.]

to :

[We considered two types of phytoplankton based on size (Fig. 3): picophytoplankton (PICO) and nano+micro-phytoplankton (NMPHYTO). PICO includes autotrophic prokaryotic organisms such as Prochlorococcus spp. and *Synechococcus spp* which are ubiquitous in the Mediterranean (Mella-flores et al., 2011). NMPHYTO aims to represent phytoplankton larger than 2 µm and smaller than 200 µm. It mainly includes diatoms and autotrophic nanoflagellates. As diatoms are an important component of

Mediterranean spring blooms (Margalef, 1978, Leblanc et al., 2018) and cover wide size-range, we decided to consider them as representative of the NMPHYTO]

and the rest of the manuscript accordingly.

**2a) The size classes of the mixotroph types included in the model are not clearly stated. This further complicates the interpretation of the possible trophic interactions allowed in the model.**

We consider CM to be mainly nanoplankton but we also consider CM belonging to a small part of the microplankton (between 20 and 50 μm based on the genus which we used to determine the values of parameters). NCM are microplankton (Esteban et al., 2010). To clarify, we precise the size class of these organisms in the text l.196:

[In Eco3M_MIX-CarbOx the NCM are based on ciliates and belong to microplankton (Esteban et al. 2010, Fig. 3).]

and l.241:

[CM are based on dinoflagellates which belong mainly to nanoplankton but can also be found in microplankton (Stoecker, 1999, Fig. 3).]

We also added Figure 3.

**2b) Specifically, CM only ingest pico- prey while NCM can ingest pico- and nano- prey as well as CMs. This assumption makes me think that authors are assuming that CMs are nano-sized while NCMs are micro-sized. In reality, CMs occur in both size classes. Many CMs are dinoflagellates that are known to feed on prey of similar size or larger.**

We agree that we consider that CM are mainly nano-sized and, we assumed that they feed on preys of strictly smaller size i.e., picophytoplankton and heterotrophic bacteria. This assumption is based on following literature (Christaki et al., 2002 ; Zubkhov & Tarron, 2008, Millette et al., 2017 ; Livanou et al., 2019) that is now, mentioned in the manuscript (l.260):

[CM feed on heterotrophic bacteria (preferred) and picophytoplankton (less preferred, Christaki et al., 2002 ; Zubkhov & Tarron, 2008, Millette et al., 2017 ; Livanou et al., 2019) and the same grazing formulation as for zooplankton and NCM is used except that CM grazing is limited by DIN (DIP) concentration and light (Stoecker, 1997, 1998; Eq. 9).]

**2c) Finally, the model assumes that NCMs least preferred prey are CMs when, in reality, many NCMs acquire their plastids from CMs so CMs should not be the least preferred prey.**

Thank you for pointing this out. In fact, NCM least preferred prey is NMPHYTO (Table E2), prey order in the sentence l.204 is wrong, we corrected it:

[From most to least preferred prey, NCM feed on heterotrophic bacteria, picophytoplankton, CM and nano+micro-phytoplankton (Verity, 1991 ; Price & Turner, 1992 ; Christaki, 1999).]

**2d) Authors must refer to the literature to justify their choices (this is currently lacking in the manuscript).**

Preference values results in the combination of values from the literature and calibrations.

For copepods: We assumed that copepods feed on strictly smaller sized preys with the strongest preference for the largest organism: NCM. NCM are considered to be ciliates, copepods preferential ingestion of ciliates in environments where diatoms and dinoflagellate are present has been demonstrated by Verity (1996). They also represent a preferential food source for the reproduction of some copepod species (Dutz & Peters, 2008). Next, we decided to apply a strongest preference on NMPHYTO as they cover a largest size range with possibly bigger organisms than CM.

For NCM: We made the choice to prioritize the ingestion of smaller organisms by considering bacteria and picophytoplankton as the preys with the highest preference as it is the case for small ciliates (Rassoulzadegan et al. 1988, Price & Turner, 1992, Christaki et al., 1999). We then prioritize nanophytoplankton which has been shown to be a great food source for ciliates in the Gulf of Lion (Christaki et al., 2009). We apply the lowest preference to NMPHYTO as they cover a wide range of size (they can be as large as NCM) and species including diatoms which are associated with smaller ciliates grazing rates (Epstein et al., 1992).

For CM: We assumed that CM consume strictly smaller sized preys. They are known to consume bacteria and picophytoplankton (Christaki et al., 2002 ; Zubkhov & Tarron, 2008, Millette et al., 2017, Livanou et al., 2019).

We add a column [references] to the table E2, to justify the values currently used for preferences, add the references in the text (l.144, l.204, and l.260) and in the caption of the new Figure 3:

[Table E2: Predator preference for their preys (COP: copepods, NMPHYTO: nano+micro-phytoplankton, PICO: picophytoplankton and BACT: heterotrophic bacteria). (20) Verity and Paffenhofer (1996), (21) Price & Turner, 1992, (22) Christaki et al., 2009, (23) Epstein et al., 1992, (24) : Christaki et al., 2002, (25) Zubkhov & Tarron, 2008, (26) Millette et al., 2017, (27) Livanou et al., 2019, (*) Calibrated.

| | | PREYS | | | | | References |
| | | NCM | CM | NMPHYTO | PICO | BACT | |
|---|---|---|---|---|---|---|---|
| | COP | 0.4 | 0.25 | 0.35 | | | 20, * |
| PRED | NCM | | 0.20 | 0.15 | 0.25 | 0.40 | 21, 22, 23, * |
| | CM | | | | 0.35 | 0.65 | 24, 25, 26, 27 * |

]

[Copepods feed with decreasing preference on NCM, nano+micro-phytoplankton (NMPHYTO), and CM (Verity and Paffenhofer, 1996) and release ammonium ($NH_4^+$), phosphate ($PO_4^{3-}$), and dissolved organic carbon (DOC) through excretion, contributing to the POM compartment through egestion and mortality.]

[From most to least preferred prey, NCM feed on heterotrophic bacteria, picophytoplankton, CM and nano+micro-phytoplankton (Verity, 1991 ; Price & Turner, 1992 ; Christaki, 1999).]

[CM feed on heterotrophic bacteria (preferred) and picophytoplankton (less preferred, Christaki et al., 2002 ; Zubkhov & Tarron, 2008, Millette et al., 2017 ; Livanou et al., 2019) and the same grazing formulation as for zooplankton and NCM is used except that CM grazing is limited by DIN (DIP) concentration and light (Stoecker, 1997, 1998; Eq. 9).]

[Figure 3: Repartition of modelled organisms (COP: copepods, PICO: picophytoplankton, NMPHYTO: nano+micro-phytoplankton, and BACT: heterotrophic bacteria) in size classes and trophic interactions between them. Preference values are indicated in grey for copepods (Verity and Paffenhofer, 1996) and NCM (Epstein, 1992; Price & Turner, 1992 ; Christaki, 2009) and CM (Christaki et al., 2002 ; Zubkhov & Tarron, 2008, Millette et al., 2017 ; Livanou et al., 2019).]

**3a) The description of how mixotrophs are being modeled is hard to follow. First, it seems that phagotrophy cannot contribute to carbon growth when light is limiting – many studies found that CMs can also use phagotrophy to supplement carbon (Vargas et al 2012 Aquat Microb Ecol; Edwards 2019 PNAS).**

We are aware that multiple types of CM exist but as mentioned in our manuscript (l.188 to 192), we chose to model one type of CM based on the classification of Stoecker (1998). This type of CM designated as type IIA, is defined as primarily phototrophic organisms which can ingest preys when dissolved inorganic nutrients are limiting. By ingesting preys, they supplement their nutrition in

nitrogen and phosphorus only. When phagotrophy supplements carbon nutrition, CM is no longer considered as a type IIA but as a type IIC (feed when light is limiting, to get carbon) considering this classification.

**3b) Second, it is especially hard to understand Tables 2 and 3 and what analyses were done to generate Figure 4.**

The aim of Table 2 is to sum up the characteristics which must be verified by the types of CM (IIA) and NCM (IIIB) that we modelled. These characteristics were stated by Stoecker (1998) as she aimed, in addition to propose a classification of mixotrophs, to provide conceptual models and then, main characteristics associated with each type of mixotrophs. By answering the question: "Are the characteristics defined in the Stoecker's classification (1998) reproduced for the type of mixotrophs that we are trying to model?" we ensure that the type of mixotrophs that we chose to mirror is correctly modelled as its main characteristics are reproduced. Consequently, Table 3 aims to present the simulations that we designed to verify that the characteristics presented in Table 2 are well reproduced by our model. For each characteristic presented in Table 2, we designed two simulations. We then compare them (i.e., depending on the characteristic tested, we compare the photosynthesis or grazing fluxes of the two simulations) to verify that the characteristic tested is well reproduced by the model. The figure 4 (now figure 5) aims to present these comparisons, i.e., the verification of each characteristic for type of mixotrophs which we chose to model, CM type IIA and NCM type IIIB.

**3c) To robustly evaluate how mixotrophy metabolism varies as a function of environmental conditions, authors should have a baseline simulation where all resources are replete (growth optimal) and then just change a given initial environmental condition (e.g. external inorganic nutrient concentration) to then compare it against the baseline model. However, it seems like the authors varied a bunch of parameters (Table 3) across simulations which make them hard to compare. A suggestion would be to have a baseline model and then run this model against a grid of all possible combinations of environmental conditions for low/high [resource], in which resource could be DIN, prey or light (these would cover the main trade-off expected across different environments).**

We considered your comments and modified our simulations accordingly. We first ran a simulation in replete conditions and then three simulations to test each characteristic: low nutrients where only the DIN and DIP concentrations are modified, low light where only irradiance is modified and low food where only the total prey concentration is modified, for NCM and CM (as both have different preys, we must separate the cases). We modified Table 3 and Section 2.4.1 to this effect (l.295):

[To verify these properties, we designed several numerical experiments (Table 3 and 4) in which we modify one of the following features: prey biomass, DIN and DIP concentrations or irradiance. We first ran a reference simulation (referred as Replete in Table 3) in which we set all the previous features to a maximum value during the entire simulation. Maximum prey biomass was obtained by multiply the initial condition by 2 (sum of the initial carbon prey biomass multiply by 2), maximum DIN and DIP concentrations were chosen based on high values observed at SOLEMIO (Pujo-Pay et al., 2011) and maximum irradiance correspond to the mean value of simulated irradiance for the SOLEMIO station by the meteorological model WRF (Yohia, 2017). Next, we ran low nutrients (low nutrients values observed at SOLEMIO multiply by 0.1, low-nut simulation in Table 3 for NCM and CM), low prey concentration (maximum prey concentration multiplied by 0.5, low-food simulation in Table 3 for NCM and CM) and low light (maximum value multiplied by 0.05, low-light simulation in Table 3 for CM only). For NCM, to verify the light dependant property (NCMP3), it is also necessary to set the NCM concentration to a constant during the entire simulation, we performed another reference simulation and a low-light simulation in which NCM concentration is constant (initial condition, NCM replete with constant and NCM low light with constant in Table 4, respectively). Finally, we compare the simulations to their associated reference simulation.

**Table 3: Summary of the simulations performed to check NCM and CM properties (excluding NCMP3). For NCM, [PREY] stands for the sum of CM, nano+micro-phytoplankton, picophytoplankton and heterotrophic bacterial biomasses. For CM, [PREY] stand for the sum of picophytoplankton and heterotrophic bacterial biomasses.**

| | | NCM properties (Type IIIB, Stoecker, 1998) | | | |
|---|---|---|---|---|---|
| Simulation name | [PREY] (mmol C m$^{-3}$) | [DIN] (mmol N m$^{-3}$) | [DIP] (mmol P m$^{-3}$) | Irradiance (W m$^{-2}$) | Tested property |
| NCM Replete | 1.5 | 1.5 | 0.09 | 120 | Reference simulation |
| NCM Low-Nut | 1.5 | 7.5×10$^{-3}$ | 4.5×10$^{-4}$ | 120 | NCMP1 and NCMP2 |
| NCM Low-Food | 0.75 | 1.5 | 0.09 | 120 | NCMP4 |
| | | CM properties (Type IIA, Stoecker, 1998) | | | |
| Simulation name | [PREY] (mmol C m$^{-3}$) | [DIN] (mmol N m$^{-3}$) | [DIP] (mmol P m$^{-3}$) | Irradiance (W m$^{-2}$) | Tested property |
| CM Replete | 0.92 | 1.5 | 0.09 | 120 | Reference simulation |
| CM Low-Nut | 0.92 | 7.5×10$^{-3}$ | 4.5×10$^{-4}$ | 120 | CMP2 and CMP3 |
| CM Low-Light | 0.92 | 1.5 | 0.09 | 3 | CMP4 |
| CM Low-Food | 0.46 | 1.5 | 0.09 | 120 | CMP1 |

]

For the verification of one of the properties (NCMP3) NCM concentration must be constant, we added Table 4, specify it in the text (l.300-304) and discussed it in a new section of the discussion: 4.1 Mixotrophs representation assessment (l.485-494).

[As biomass measurements were not available for our location, we performed the assessment of mixotrophs based on properties listed in Table 2. We showed that NCM and CM properties were all well reproduced by the model (Fig. 5). The third NCM property : grazing and irradiance are independent (NCMP3, Table 2), required a constant NCM concentration to be verified (Table 4). When irradiance increases, the NCM concentration increases. This feature is only due to the photosynthesis process which become less limited by light. NCM photosynthesis includes a prey dependant (i.e., based on preys' parameters) light limitation function (the closer the function is to 1, the less limited the organisms) which tends to 1 when irradiance increases. Grazing formulation does not include a term of direct dependence on light but includes NCM biomass which explains the increase of grazing when NCM biomass is not set to a constant. It seems difficult to avoid this feature as photosynthesis is known to increase up to a certain value of irradiance which depends on species (Platt et al., 1980 ; Geider, 2013).]

**3d) Second, I think it is much more clear to the reader if the steady-state solutions are presented for these analyses (time-series are more exciting when presenting the results applied to the coastal site).**

For this study with Eco3M_MIX-CarbOx we can consider yearly results as steady-state because for each simulation we reproduce the year 2017 three times (to avoid oscillations due to initial conditions (spin-up)), and we obtain repeating steady-state cycles for the second and third repetitions. These repeating cycles (and not the strait lines usually observed for steady-state) are mainly explained by the fact that we used typical temperature values of the BoM (measurements performed in the BoM, TMED-Net) to perform our simulations. As the BoM is a temperate area, it is important to consider these temperature values as they allow us to reproduce the seasonality observed in the area. We reproduced this seasonality as some of our organisms (NMPHYTO, CM and PICO) growth depends on temperature. By applying a constant or a gaussian type curve for temperature we necessarily advantage one of these organisms: If we chose winter conditions, NMPHYTO will be advantaged due to its low temperature growth optimal, however if we choose a mean temperature (16.5°C, mean of observations for 2017) or summer conditions CM and PICO will be advantaged. To illustrate it, we present the yearly steady states obtained for two types of conditions (Table S4) : low nutrient and low light (Fig. S5).

**3e)  By looking only at steady-state, authors can present a wider range of simulations and clearly show how phototrophy and phagotrophy changes across environments. A suggestion to visualize this, is to make a grid plot (akin to a correlation matrix) where the environments would mirror each other and so phototrophy could be presented in one "side" of the matrix and phagotrophy could be presented on the other "side" of the matrix, with the diagonal grids left empty or dashed. Finally, I think the same runs should be done to evaluate both CM and NCM properties.**

Thank you for this suggestion. We decided to present time-series of grazing and photosynthesis for the properties verification for the reasons mentioned in point 3d) of this review. However, based on your comment, we added Table S5 to the appendices (Appendix X) to present a yearly mean value of photosynthesis and grazing for each simulation designed to verify the NCM and CM properties.

**4a) The analysis lacks a validation for the different plankton groups modelled. The authors state that their model correctly simulates mixotrophy, but the model was only compared against total chlorophyll data. While the perfect dataset to validate mixotrophy is not currently available (we cannot detect mixotrophs in situ with traditional sampling approaches), there are many long-term time-series that provide detailed information on carbon biomass for different taxa and these can be grouped into different functional types to validate the modeled carbon biomass output. There are papers that summarized a list of species known to be mixotrophic and this could be used as a guide to apply these timeseries to validate mixotroph types in ecosystem models.**

As we stated in point 3b), we assume that our model correctly reproduced the processes associated with mixotrophy as we verified that our model reproduces each characteristic stated by Stoecker's classification (1998) well for both types of mixotrophs that we chose to model. As you rightly pointed out, a set of measurements for mixotrophs in the BoM is not available at the moment. We chose to present total chlorophyl measurements to evaluate our representation of the plankton as these measurements are performed at the modelled point. Unfortunately, to the best of our knowledge, there is not any available dataset in the BoM which allow us to compare the carbon biomass of each organism to measurements.

Some studies which investigated phytoplankton succession during the year in the North-Western Mediterranean Sea are available (Marty et al., 2002; Siokou-Frangou et al., 2010 ; Estrada & Vaquë, 2014 ; Mayot et al., 2016). They highlighted three successive periods during the year: (i) at the beginning of the year, winter mixing brings nutrients to the surface layer then initiates a spring bloom first dominated by large cells (mainly diatoms) which belong to microphytoplankton, (ii) microphytoplankton development is followed by nanophytoplankton development which dominates the second stage of the bloom until late spring, (iii) nutrients run out, summer phytoplankton assemblage which contains mainly small cells (pico- and nanophytoplankton), takes place. Due to the high variability of the BoM, seasonal phytoplankton successions during the year may be made more complex. As instance, Rhône River intrusion, Cortiou water intrusion and upwelling events can significantly modify phytoplankton assemblages. Rhône River intrusions can occur all year round and are particularly efficient in low nutrient conditions as they bring nutrient loaded water to the bay and allowed large cells to grow (Gatti et al., 2006; Fraysse et al., 2014). Cortiou water intrusion events can also occur all year-round and brings $NH_4^+$ loaded water which can help reduce nutrient limitation, especially in summer. Upwelling events are mainly visible in summer and can favoured species with lower growth temperature optimum in addition to bring nutrients to the surface (Millot, 1990 ; Pairaud et al., 2011). Due to this high variability, it seems difficult to give a typical succession that the model should reproduce to be considered as correct.

For 2017, Eco3M_MIX-CarbOX reproduced well Rhône River intrusion as we modelled well the initiated development of large cells (NMPHYTO) by nutrients bring by the winter mixing and the Rhône River

intrusion (Fig. S6, 1). This first stage is followed by the replacement of large cells by nano size cells as nutrient concentration decreases (Fig. S6, 2). When nutrients are all consumed, summer phytoplankton assemblage dominated by pico size cells takes place (Fig. S6, 3). This last stage is significantly impacted by upwelling events which result in a high variability of PICO and NANO biomass.

**4b) As of now, it is hard to believe the model predictions are robust, for example, Figure 6 shows copepods and non-constitutive mixotrophs to have a much higher biomass than other groups. The non-constitutive mixotrophs modeled here (i.e. Laboea and Strombidium) are known to occur in low abundance and contribute less to total biomass.**

Thank you for pointing this out. We agree that this result can be surprising. Even if, in the Gulf of Lion and especially in low salinity water from the Rhône River, *Laboea strobila* has been found abundant (Christaki et al., 2009), we do not exclude that, by only considering copepods as predator of NCM, we can underestimate the grazing that occurs on this type of organisms.

Based on your comments, we understand that our manuscript may be missing a part to discuss the assessment of mixotrophs, we then decided to add a part 4.1 Mixotrophs representation assessment, in which we discuss the mixotrophs properties verification (especially NCMP3) and the CM and NCM dynamics representation.

[**4.1 Mixotrophs assessment**
As biomass measurements were not available for our location, we performed the assessment of mixotrophs based on properties listed in Table 2. We showed that NCM and CM properties were all well reproduced by the model (Fig. 5). The third NCM property : grazing and irradiance are independent (NCMP3, Table 2), required a constant NCM concentration to be verified (Table 4). When irradiance increases, the NCM concentration increases. This feature is only due to the photosynthesis process which become less limited by light. NCM photosynthesis includes a prey dependant (i.e., based on preys' parameters) light limitation function (the closer the function is to 1, the less limited the organisms) which tends to 1 when irradiance increases. Grazing formulation does not include a term of direct dependence on light but includes NCM biomass which explains the increase of grazing when NCM biomass is not set to a constant. It seems difficult to avoid this feature as photosynthesis is known to increase up to a certain value of irradiance which depends on species (Platt et al., 1980 ; Geider, 2013).

Regardless of the simulation we modelled close percentage of C biomass for NCM (ciliates) and copepods (difference maximum of 6% Fig. 10a). These percentages are always significantly higher than phytoplankton and heterotrophic bacteria ones. In the Gulf of Lion and especially in low salinity water from the Rhône River, oligotrich ciliates have been found abundant (Christaki et al., 2009). We do not exclude that, by only considering copepods as predator of NCM, we can underestimate the grazing that occurs on this type of organisms. In the present model, we do not consider strict heterotrophs which belong to the nano and micro size classes. These organisms can be important competitors of ciliates, and certain species can even consume ciliates (Stoecker and Capuzzo, 1990 ; Johansson et al., 2004). The adding of these organisms could improve the representation of NCM dynamics and, accordingly, of the ecosystem and then will be considered for an improved version of the model. Moreover, we do not consider a mortality term for NCM. Montagnes (1996) showed that mortality rates for two species of the genus *Strombidium* and two species of the genus *Strombilidium* were rapid. Accordingly, adding this term to the model could allow to represent a more realistic NCM biomass.

Regardless of the simulation, CM percentage in C biomass remains close to the phytoplankton one (Fig. 10a). We performed the assessment of phytoplankton for the typical simulation, by using SOLEMIO chlorophyll measurements (Fig. 5e, statistical analysis presented in Appendix F). According to statistic indicators, Eco3M_MIX-CarbOx reproduced well measured chlorophyl (cost function below 1 and

RMSD close to 0). Especially, the model provided values in the same range than observations with relatively close mean (0.40 for the model and 0.39 for observations). Observed chlorophyl reached a maximum value in mid-March, linked to the Rhône River intrusion which is not reproduced by the model. This maximum can be linked to an input of allochthonous chlorophyl (i.e., phytoplankton development near the nutrients loaded Rhône River plume, which is brought to SOLEMIO by currents, Fraysse et al., 2014). As Eco3M_MIX-CarbOx is dimensionless (only time derivation), we do not represent this input which can explain that we are not able to reproduce this chlorophyll maximum. However, during this event, we reproduced well the development and dominance of large cells (NMPHYTO) commonly observed in these cases (Fraysse et al., 2014).]

**5) Figure 7 requires more explanation and could be modified to provide more interesting results from the simulations. First, are these simulations the steady-state solution of the BoM runs? If not, which simulation runs? (same for Figure 9)**

The simulation used to obtain Figures 7, 8 and 9 (now 8, 9 and 10) are the simulations presented in Table 4 (now Table 5), Section 2.4.2. Figures 7 presents the carbon fluxes for NCM and CM (a, c) in typical conditions (i.e., simulation in SOLEMIO conditions: typical simulation in Table 4) and (b, d) in nutrient limited conditions (nutrient limited simulation in Table 4). In Table 4, typical simulation was referred as realistic simulation, we corrected it to avoid confusion. The same simulations were used to obtain Figure 8 (a and c are the results for typical simulation and b and d for nutrient limited simulation). Figure 9 presents the yearly ecosystem and phytoplankton composition for light limited conditions (low light, high nutrients, low light simulation in Table 4), typical conditions (realistic light, realistic nutrients, typical simulation in Table 4) and nutrient limited conditions (high light, low nutrients, nutrient limited simulation in Table 4). Again, to avoid confusion we modified the term realistic by typical in the Figure 9.

**5b) Second, it is nice to see how much carbon and nitrogen is coming from phototrophy versus phagotrophy, but I don't fully understand the relevance of the fractions into different pools (e.f. respiration, predation, etc).**

We presented the percentages for all the process as we wanted to illustrate the importance of each of them in mixotrophs metabolism. In other words, by presenting the percentage of each process, we can evaluate which of them is the most important source or loss of carbon, nitrogen, and phosphorus. For the NCM, changes in nutrient concentration mainly affect grazing and photosynthesis percentages (so mainly the sources) but for CM, sources and losses are significantly affected by a change in nutrient concentration. Moreover, for CM, we decided to present the total N (P) uptake to discuss the consideration of another form of mixotrophy (osmotrophy) by the model (l.592), and the DOC exudation flux to compare it to Livanou et al. (2021) (l. 602).

**5c) Most relevant would be to show what is the fraction of copepod total predation that corresponds to mixotrophs (relative to other prey), for example. This could then be contrasted against the sensitivity runs that do not consider mixotrophs (related to one of my other comments).**

We agree, so we calculated the fraction of copepod total predation that corresponds to each prey and added it to Table S2 (point 1a) of this review.

**6a) A big chunk of the discussion is describing more results (lines 430-490). I would move this to the Results and would enrich the discussion, as of now the paper does not mention previous ecosystem modeling studies that also tackled this question in other systems, such as using MIRO and ERSEM ecosystem models.**

We used the Figure 9 (now Figure 10) to draw a parallel between the phytoplankton compositions for the nutrient and light conditions (Table 4 now Table 5) and the phytoplankton compositions for specific

events studied in Section 3.2. In other words, we used the specific events to give examples of low light, or low nutrient conditions in the BoM, that is why we place the figure 9 in discussion and not in results.

We gave more examples of studies which modelled mixotrophy (l.612-621) in different types of ecosystems :

[An increasing number of studies has been investigating the impact of mixotrophs on their environment and were able to highlight the crucial role played by these organisms in the food web (Mitra et al., 2016; Ward and Follows, 2016; Ghyoot et al., 2017 ; Stoecker et al., 2017). For instance, once Ward and Follows (2016) started to consider consider mixotrophs in their food web model, the biomass maximum switched to larger organisms which in turn led to an increase in carbon export to depth due to the production of larger carbon-enriched detritus. Still using a modelling approach (MIRO model), Ghyoot and al. (2017) investigated the impact of the introduction of three forms of mixotrophy (osmotrophy, non-constitutive mixotrophy and constitutive mixotrophy) on trophic dynamics in the Southern North Sea. They showed that these three types of mixotrophy have different impact on system dynamics: while results showed that constitutive mixotrophy did not significantly affect the functioning of the ecosystem, osmotrophy increased gross primary production (GPP), sedimentation and bacterial production and non-constitutive mixotrophy also increased remineralisation and transfer to higher trophic level under high irradiance. Mixotrophy was also shown to play an important role in harmful algal blooms (Kempton et al., 2002 ; Burkholder et al., 2008). Accordingly, the need of developing models which include mixotrophy to represent and predict such events has been raised by several authors (Burkholder et al., 2008 ; McGillycuddy, 2010 ; Mitra & Flynn, 2010 ; Flynn & McGillicuddy, 2018).]

and (l.625-635):

[For instance, Leles et al. (2018) investigated the impact of light and nutrient on mixotrophs and on their strict autotrophic and heterotrophic competitors modelling. They showed that changes in light and nutrients resulted in significant changes in ecosystem composition: while strict autotrophs and heterotrophs increased in relative importance in the transition from nutrient to light limitation, nutrient poor conditions favoured the development of mixotrophs. Still using modelling, Schneider et al. (2021) investigate the hypothesis that the biogeochemical gradient of inorganic nutrient and suspended sediment concentrations drives the observed occurrence of constitutive mixoplankton in the Dutch Southern North Sea. They showed that dissolved inorganic phosphate and silica concentration drive the occurrence of constitutive mixoplankton.].

**6b) And finally, it is not clear to me what were the main contributions of this study to this field. Authors should be more clear about it in the discussion and in the abstract.**

This work provides a new biogeochemical model to study mixotrophy in the BoM. With the present study, we provided new insights regarding the condition that lead to the emergence of mixotrophs in the BoM. On a more general note, thanks to its adaptability, this model represents a new tool to perform long-term studies and prediction of mixotroph dynamics in coastal environments, under different environmental forcings particularly under different environmental forcings caused by global change where mixotrophs are expected to play a central role in future ecosystems.

We added at the end of the abstract :

[In addition to provide new insights regarding the condition that lead to the emergence of mixotrophs in the BoM, this work provide a new tool to perform long-term studies and prediction of mixotroph dynamics in coastal environments, under different environmental forcings.]

We reorganised the last point of the discussion (l.648-656):

[revised manuscript text omitted]

**Study area should come at the end of methods description. It is a study case of the model.**

We chose to place the study area subsection (2.1) at the beginning of the material and methods section because the manuscript seemed more consistent to us in this way. In the part 2.4, we mentioned that irradiance constant values were determined based on the irradiance of the model WRF (typical irradiance for the BoM). In the part 2.5, we mentioned again that irradiance was based on the model WRF and we added to the manuscript (l. 326) that we used the environmental forcings described in Table 1 for the three simulations. Consequently, it seems more consistent for us to leave this part at the beginning, in order to define the forcings before using them.

**Model description: please be clear that this is a 0D model (only time derivation, no physics resolved either over depth or horizontally).**

We changed the sentence (l.115-117) :

[The Eco3M_MIX-CarbOx model is a dimensionless model (i.e., we consider a volume of 1 m$^3$ of surface water at SOLEMIO station) which was developed to represent the dynamics of both mixotrophic protists (henceforth referred to as mixotrophs) and the carbonate system in the BoM]

to:

[The Eco3M_MIX-CarbOx model is a dimensionless (0D) model: we consider a volume of 1 m$^3$ of surface water at SOLEMIO station, in this volume the state variables only vary over time as the model is not coupled with a hydrodynamic model. Eco3M_MIX-CarbOx was developed to represent the dynamics of both mixotrophic protists (henceforth referred to as mixotrophs) and the carbonate system in the BoM.].

**Methods: Please explain Eco3M; system of ODEs..**

We modified (l.120-122):

[To obtain the present version of the Eco3M_MIX-CarbOx model, we developed a planktonic ecosystem model which contains mixotrophs, using the Eco3M (Ecological Mechanistic and Molecular Modelling) platform (Baklouti et al., 2006a, b) and added a modified version of the carbonate module from Lajaunie-Salla et al. (2021)]

to:

[To obtain the present version of the Eco3M_MIX-CarbOx model, we developed a planktonic ecosystem model which contains mixotrophs, and added a modified version of the carbonate module from Lajaunie-Salla et al. (2021). The planktonic ecosystem model was developed using the Eco3M (Ecological Mechanistic and Molecular Modelling) platform (Baklouti et al., 2006a, b). The Eco3M platform allows the modelling of the first trophic levels by providing a process library used to build different model configurations. It was developed in Fortran 90/95 and we used an Euler method to solve sink-source equation of each state variable.].

**Methods: If the carbonate cycle is not relevant here, talk less about it.**

We mentioned the carbonate cycle to remind that a companion paper is available to detail its modelling. These sentences allowed us to link both papers then clarify the structuring of the study.

**Figure 2: why zooplankton eat picophytoplankton but not bacteria? It seems weird to me.**

Zooplankton does not eat picophytoplankton neither bacteria (Table B1 and Table E2). We added Figure 3 to avoid confusions.

**Line 132: Uncertain why copepods prefer NCMs and nanophytoplankton over CMs – why is this assumption necessary?**

We made this assumption based on size classes in which belongs each modelled organism (please see point 2 for detailed explanation).

**Why do you mention specific genus when talking about each type represented in the model if you don't have validation data (biomass) for these? Isn't the goal of the model to resolve all taxa that could be included in that box?**

Exactly. We mention a specific genus as we used this genus as a reference for most of the values chosen for parameters (Table E1). As it seems confusing, we removed them.

**Line 144: explain temperature effects in a separate section. Does it affect only nutrient uptake? But temperature affects all metabolic rates.**

We do not find relevant to add a specific section for temperature effect, but we indicated it when temperature influences a process. In addition, temperature affects all the CM, phytoplankton, and heterotrophic bacteria processes. For the zooplankton and NCM, the temperature effect is indirect through their prey.

**Line 147: what about mortality?**

We did not consider a term of mortality for phytoplankton. We performed the necessary tests to add this term when developing the model, but none were conclusive as our phytoplankton compartment was already balanced. We believe that this due to the fact that predators exert a strong top-down control on phytoplankton population.

**Line 189: why quadratic?**

Grazing formulation is based on Fasham et al. (1990). See Appendix B of Fasham et al. (1990) for detailed explanations.

**Line 190: degradation rate or mortality rate? Please provide references that could back up your modeling choices if this was the case.**

We based the modelling of the degradation of sequestered chloroplast on Leles et al. (2018). The equations are available in their supplementary material. They used the term dilution instead of degradation or mortality. We used degradation to speak about the global process and mortality for the rate which multiply chlorophyl quantity.

**Line 199: Hard to follow the calculations to get the photosynthetic rate of NCMs.**

$$P_{MAX,NCM}^C = P_{REF,PREY}^C * f_{PREY}^T * f_{Q,NCM}^G$$

$$Photo_{NCM_C,PREY_C}^{DIC} = P_{MAX,NCM}^C * limI_{PREY} * NCM_C$$

Eq. S1

The calculation of NCM photosynthesis (set of Eq. S1) is based on Geider et al. (1998) formulation. This formulation provides a photosynthesis flux which depends on nutrient concentration, temperature, and light. We first calculate the maximum photosynthetic rate ($P_{MAX,NCM}^C$) by using the C-specific photosynthetic rate at a reference temperature of the prey ($P_{REF,PREY}^C$) which is limited by temperature ($f_{PREY}^T$) and nutrients ($f_{Q,NCM}^G$). Next, we calculate the photosynthesis flux ($Photo_{NCM_C,PREY_C}^{DIC}$) by using the maximum photosynthetic rate calculated before, adding the light limitation ($limI_{PREY}$) and multiplying it by NCM carbon biomass to obtain a final photosynthesis flux in mmolC m$^{-3}$ s$^{-1}$.

As the photosynthesis takes place in the NCM cell, the limitation by nutrient is calculated based on the NCM internal content (N:C ($Q_C^N$) and P:C ($Q_C^P$) ratio) :

$$f_Q^G = \min\left(\frac{Q_c^N - Q_{c,min}^N}{Q_{c,max}^N - Q_{c,min}^N}, \frac{Q_c^P - Q_{c,min}^P}{Q_{c,max}^P - Q_{c,min}^P}\right)$$

Eq. S2

As the photosynthesis is performed by photosynthetic prey chloroplast, the temperature limitation ($f^T$, from Lacroix and Gregoire, 2002) is based on optimal ($T_{OPT}$), lethal growth temperature ($T_{LET}$) and a temperature curve shape factor (β) of the considered photosynthetic prey:

$$f^T = \frac{2 * (1 - \beta) * \frac{(T - T_{LET})}{(T_{OPT} - T_{LET})}}{\left(\frac{(T - T_{LET})}{(T_{OPT} - T_{LET})}\right)^2 + 2 * (-\beta)\frac{(T - T_{LET})}{(T_{OPT} - T_{LET})} - 1}$$

Eq. S3

Same for the light limitation (limI) which is calculated based on Chl:C ratio, maximum photosynthesis rate ($P_{MAX}^C$), chlorophyll-specific light absorption coefficient ($\alpha_{Chl}$) of the considered photosynthetic prey :

$$limI = 1 - \exp\left(\frac{-\alpha_{Chl} * Q_C^{Chl} * E_{PAR}}{P_{MAX}^C}\right)$$

Eq. S4

with $P_{MAX}^C$ calculated with the prey parameters.

To obtain the total photosynthesis flux of NCM, we sum the photosynthesis fluxes calculated for each photosynthetic prey and weighted them by preference.

We add (l.217):

[We based our formulation on Geider et al. (1998) which provide a photosynthesis flux nutrient, temperature, and light dependant. In this formulation, a maximum photosynthetic rate is first calculated ($P_{MAX}^C$) based on the C-specific photosynthetic rate at a reference temperature of the photosynthetic organism ($P_{REF}^C$). This rate is nutrient and temperature dependant and is next multiplied by a light limitation function.]

**Line 211: it seems like the calculations do not take into account chl ingested from each prey. Shouldn't it be more logical this way?**

In the present formulation we do not directly consider the ingested chlorophyll for each prey. It is another possibility to represent this process. We chose the present formulation as it was less complex and more adapted to our model. However, we would like to point out that the present formulation considers the benefits of grazing through the calculation of a quota function ($f_{Q,NCM}^G$, formulation above). Prey dependence is then added through photosynthesis calculation parameters (i.e., we used prey parameters to calculate temperature ($f^T$) and light limitation (limI) functions (formulations above) and based the calculation of the maximum photosynthetic rate on the C-specific photosynthetic rate of the prey ($P_{REF,PREY}^C$)) and the portion of each photosynthetic prey through their associated preference value.

**Lines 228-229: without definitions for intermediate calculation it is very hard to follow equations because the reader gets confused about which are constant values in the model and which are intermediate calculations based on other variables.**
To avoid confusion, we detailed the terms in the text (l.251):

[where $P_{MAX}^C$ is the maximum photosynthetic rate in s$^{-1}$, $Photo_{CM_C}^{DIC}$ is the CM photosynthetic flux in mmol m$^{-3}$ s$^{-1}$, $P_{REF}^C$ is the C-specific photosynthetic rate at a reference temperature (see Appendix E for CM value). $f^T$, $f_Q^G$ and limI are temperature, nutrient and light limitation functions respectively (see Appendix C for detailed formulations of $f^T$ and limI, and Eq. 5 for the formulation of $f_Q^G$).]

**Give references to justify prey preferences. Sensitivity analyses might be needed to test the impact of these assumptions.**

We add a column [References] to Table E2 to indicate the references for prey preferences. Preference values results in the combination of values from the literature and calibrations. To find the better values for our modelled organisms, we made several sensitivity tests during the development stage (test of different combination of values for the copepods, NCM and CM).

**Table 3 is very hard to follow. Instead of giving tested properties, give this in the results and be more clear about these (with references to support these properties).**

Table 3 aims to present the simulations which have been done to test each property presented in Table 2. For us, it seems logical to precise the property associated with their simulations. The number of the tested property is also indicated on the Figure 4 (now Figure 5) for the reader to refer in Table 2. As indicated in Table 2 and in the manuscript (l.190 for example), properties are based on Stoecker (1998). Based on your previous comment, we modified Table 3. We hope it is clearer that way.

**Line 283: confusing sentence. Tota/ sum of what exactly?**

Total carbon biomass and sum of daily averaged carbon biomass. We modified (l.331) :
[We used the total carbon biomass which is calculated by summing daily average biomass of each organism to assess the ecosystem composition and its dynamics during different scenarios over a full year. We used the total phytoplanktonic carbon biomass which is calculated by summing daily average carbon biomass of each phytoplanktonic organism, to assess the phytoplankton composition (given as percentages of nano+micro-phytoplankton, picophytoplankton and CM).]

**Define acronyms throughout.**

Done.

**What about the spin-up period of the model? Give details where the code is deposited at the end of the Methods.**

We agree that this was not clearly stated and indicated model spin-up period on l.312:

[Eco3M_MIX-CarbOx spin-up period is about 3 months. To avoid initial conditions impact on our results, we ran three years of simulation (i.e., repetition of 2017 three times) and we present the results for the second year of simulation.].

For details where the code is deposited, a section [Code availability] at the end of the manuscript is available.

Figures and Tables

[Figure]

**Figure 3**: Repartition of modelled organisms (COP: copepods, PICO: picophytoplankton, NMPHYTO: nano+micro-phytoplankton and BACT: heterotrophic bacteria) in size classes and trophic interactions between them. Preference values are indicated in grey for copepods (Verity, 1996) and NCM (Epstein, 1992; Price & Turner, 1992 ; Christaki, 2009) and CM (Christaki et al., 2002 ; Zubkhov & Tarron, 2008, Millet et al., 2017 ; Livanou et al., 2019). **[added to the manuscript]**

[Figure]

**Figure S1**: Comparison of ecosystem composition in C biomass for the configurations (a) without mixotrophs and, (b) with mixotrophs, and (c) of net community production (difference of total photosynthesis and total respiration) with and without mixotrophs. The ecosystem is dominated by autotrophic processes when values are higher than 0 (green coloured part) and by heterotrophic processes when values are lower than 0 (red coloured part). Results are given for the typical simulation.

[Figure]

**Figure S2**: Schematic representation of the Eco3M_MIX-CarbOx model configuration with strict heterotrophs (MICROZ) and strict autotrophs (NANOP) and without mixotrophs. Each box represents a model compartment (DIM: dissolved inorganic matter, DOM: labile dissolved organic matter, POM: detrital particulate organic matter). State variables are indicated in black (COP: copepods, PICO: picophytoplankton, NANOP: nanophytoplankton, MICROP: microphytoplankton, O2: dissolved oxygen,

CO2: dissolved carbon dioxide, DIC: dissolved inorganic carbon, TA: total alkalinity, pCO2: partial pressure of CO2, CaCO3: calcium carbonate). Elements for which a state variable is expressed with a variable stoichiometry are shown in blue (C: carbon, N: nitrogen, P: phosphorus and, Chl: chlorophyll). Arrows represent processes between two state variables.

[Figure]

**Figure S3**: Repartition of modelled organisms (COP: copepods, MICROZ: microzooplankton, NANOP: nanophytoplankton, MICROP: microphytoplankton, PICO: picophytoplankton and BACT: heterotrophic bacteria) in size classes and trophic interactions between them. Preference values are indicated in grey for copepods (Verity, 1996) and MICROZ (Epstein, 1992; Price & Turner, 1992 ; Christaki, 2009).

[Figure]

**Figure S4**: Yearly ecosystem composition in percentage of C biomass (a, b) in typical conditions (a) with mixotrophs and (b) without mixotrophs, and (c, d) in nutrient limited conditions (c) with mixotrophs and (d) without mixotrophs.

[Figure]

**Figure S5**: Repeating cycles of model simulations for (a) the high light, low nutrient and (b) low light, high nutrients simulations (Table S4). Lines represents daily averaged carbon biomass of copepods (COP), NCM, CM, nano+micro-phytoplankton (NMPHYTO), picophytoplankton (PICO) and heterotrophic bacteria (BACT) for the three years of simulation (repetition of 2017 three times).

[Figure]

**Figure S6:** Time-series of daily averaged carbon biomass of phytoplankton for the typical simulation (Table 5). The different assemblages of phytoplankton community are separated by grey dotted lines: (1) development of large cells initiated by nutrients bring by the Rhône River intrusion and winter mixing, (2) decrease of nutrient concentration, development of nano-size cells, (3) summer phytoplankton assemblage dominated by pico-size cells and influenced by upwelling events.

**Table 4**: Summary of the simulations performed to NCMP3. Prey stands for the sum of CM, nano+micro-phytoplankton, picophytoplankton and heterotrophic bacterial biomasses. **[added to the manuscript]**

| Simulation name | [NCM] (mmol C m$^{-3}$) | [PREY] (mmol C m$^{-3}$) | [DIN] (mmol N m$^{-3}$) | [DIP] (mmol P m$^{-3}$) | Irradiance (W m$^{-2}$) | Tested property |
|---|---|---|---|---|---|---|
| **NCM Replet with constant** | 0.4 | 1.5 | 1.5 | 0.09 | 120 | Reference simulation |
| **NCM Low light with constant** | 0.4 | 1.5 | 1.5 | 0.09 | 3 | NCMP3 |

**Table S1:** Balance equations for MICROZ (microzooplankton) and NANOP (nanophytoplankton).

| Variables | Balance equations |
|---|---|
| MICROZ$_X$
 $X \in [C, N, P]$ | $$\frac{\partial \text{MICROZ}_C}{\partial t} = \sum_{i=1}^{3} \left( \text{Gra}_{\text{MICROZ}_C}^{\text{PHYC}_i} \right) + \text{Gra}_{\text{MICROZ}_C}^{\text{BAC}_C} - \text{Resp}_{\text{MICROZ}_C}^{\text{DIC}} - \text{Exu}_{\text{MICROZ}_C}^{\text{DOC}} - \text{Gra}_{\text{MICROZ}_C}^{\text{COP}_C}$$ $$\frac{\partial \text{MICROZ}_N}{\partial t} = \sum_{i=1}^{3} \left( \text{Gra}_{\text{MICROZ}_N}^{\text{PHYN}_i} \right) + \text{Gra}_{\text{MICROZ}_N}^{\text{BAC}_N} - \text{Exu}_{\text{MICROZ}_N}^{\text{DON}} - \text{Excr}_{\text{MICROZ}_N}^{\text{NH}_4} - \text{Gra}_{\text{MICROZ}_N}^{\text{COP}_N}$$ $$\frac{\partial \text{MICROZ}_P}{\partial t} = \sum_{i=1}^{3} \left( \text{Gra}_{\text{MICROZ}_P}^{\text{PHYP}_i} \right) + \text{Gra}_{\text{MICROZ}_P}^{\text{BAC}_P} - \text{Exu}_{\text{MICROZ}_P}^{\text{DOP}} - \text{Excr}_{\text{MICROZ}_P}^{\text{PO}_4} - \text{Gra}_{\text{MICROZ}_P}^{\text{COP}_P}$$ $$\text{PHY} \in [\text{MICROP, NANOP, PICOP}]$$ |
| NANOP$_X$
 $X \in [C, N, P, Chl]$ | $$\frac{\partial \text{NANOP}_C}{\partial t} = \text{Photo}_{\text{NANOP}_C}^{\text{DIC}} - \text{Resp}_{\text{NANOP}_C}^{\text{DIC}} - \text{Exu}_{\text{NANOP}_C}^{\text{DOC}} - \sum_{i=1}^{2} \left( \text{Gra}_{\text{NANOP}_C}^{\text{ZOOC}_i} \right)$$ $$\frac{\partial \text{NANOP}_N}{\partial t} = \text{Upt}_{\text{NANOP}_N}^{\text{NO}_3} + \text{Upt}_{\text{NANOP}_N}^{\text{NH}_4} + \text{Upt}_{\text{NANOP}_N}^{\text{DON}} - \text{Exu}_{\text{NANOP}_N}^{\text{DON}} - \sum_{i=1}^{2} \left( \text{Gra}_{\text{NANOP}_N}^{\text{ZOON}_i} \right)$$ $$\frac{\partial \text{NANOP}_P}{\partial t} = \text{Upt}_{\text{NANOP}_P}^{\text{PO}_4} + \text{Upt}_{\text{NANOP}_P}^{\text{DOP}} - \text{Exu}_{\text{NANOP}_P}^{\text{DOP}} - \sum_{i=1}^{2} \left( \text{Gra}_{\text{NANOP}_P}^{\text{ZOOP}_i} \right)$$ $$\text{ZOO} \in [\text{COP, MICROZ}]$$ |

**Table S2**: Percentage of copepod total grazing represented by each prey in typical and nutrient limited conditions for the configurations with and without mixotrophs.

| | With mixotrophs | | | Without mixotrophs | | |
|---|---|---|---|---|---|---|
| | NCM | CM | NMPHYTO | MICROZ | MICROP | NANOP |
| **Typical conditions** | 96.4% | 2.4% | 1.2% | 92.2% | 2.6% | 5.2% |
| **Nutrient limited conditions** | 98.4% | 1.5% | 0.1% | 94.3% | 0.7% | 5.0% |

**Table S3**: Yearly mean predation on copepods (PRED$_{COP}$), total photosynthesis (PHOTO$_{TOT}$), and total respiration (RESP$_{TOT}$) fluxes in typical and nutrient limited conditions for the configurations with and without mixotrophs.

| | With mixotrophs | | | Without mixotrophs | | |
|---|---|---|---|---|---|---|
| | RESP$_{TOT}$ (mg m$^{-2}$ d$^{-1}$) | PHOTO$_{TOT}$ (mg m$^{-2}$ d$^{-1}$) | PRED$_{COP}$ (mg m$^{-2}$ d$^{-1}$) | RESP$_{TOT}$ (mg m$^{-2}$ d$^{-1}$) | PHOTO$_{TOT}$ (mg m$^{-2}$ d$^{-1}$) | PRED$_{COP}$ (mg m$^{-2}$ d$^{-1}$) |
| **Typical conditions** | 6.6 | 6.8 | 1.5 | 8.2 | 8.3 | 1.3 |
| **Nutrient limited conditions** | 0.2 | 0.4 | 0.09 | 0.4 | 0.5 | 0.06 |

**Table S4**: Low nutrients and low light simulations properties.

| Simulation name | [DIN] (mmolN m$^{-3}$) | [DIP] (mmolP m$^{-3}$) | Irradiance (W.m$^{-2}$) |
|---|---|---|---|
| **Low nutrients** | $7.5 \times 10^{-3}$ | $4.5 \times 10^{-4}$ | 120 |
| **Low light** | 1.5 | 0.09 | 3 |

**Table S5**: Yearly mean grazing and photosynthesis values for NCM and CM properties verification simulations. **[added to the manuscript]**

| NCM | | |
|---|---|---|
| **Simulation** | Yearly mean grazing (mmolC m$^{-3}$ s$^{-1}$) | Yearly mean photosynthesis (mmolC m$^{-3}$ s$^{-1}$) |
| **NCM-Replet** | $5.16 \times 10^{-6}$ | $2.35 \times 10^{-6}$ |
| **NCM-Low Nut** | $5.16 \times 10^{-6}$ | $2.35 \times 10^{-6}$ |
| **NCM-Low Food** | $1.50 \times 10^{-6}$ | $9.54 \times 10^{-7}$ |
| **NCM-Replet Constant** | $7.60 \times 10^{-7}$ | $1.12 \times 10^{-6}$ |
| **NCM-Low light Constant** | $7.60 \times 10^{-7}$ | $3.70 \times 10^{-7}$ |
| **CM** | | |
| **Simulation** | Yearly mean grazing (mmolC m$^{-3}$ s$^{-1}$) | Yearly mean photosynthesis (mmolC m$^{-3}$ s$^{-1}$) |
| **CM-Replet** | $3.67 \times 10^{-8}$ | $8.81 \times 10^{-6}$ |
| **CM-Low Nut** | $2.02 \times 10^{-7}$ | $1.18 \times 10^{-6}$ |
| **CM-Low Light** | $1.00 \times 10^{-9}$ | $2.70 \times 10^{-7}$ |
| **CM-Low Food** | $1.60 \times 10^{-8}$ | $7.60 \times 10^{-6}$ |

---

## Author Response (AR2)

**Authors response**

Dear editor,

First, we would like to thank you for raising these two concerns. We hope that the modifications provided below will clarify them. Also, we would like to thank again the referee #1 and #2 for their careful evaluation of our manuscript and revised manuscript, and their suggestions which allowed us to improve it.

**First, regarding a potential control simulation without mixotrophs, was something like this performed using the model? The experiment list in Table 3 would suggest not, and I've not been able to see anything like this. If you have performed such an experiment – even if it is confined to supplementary material – please note this in the manuscript. If you have not performed such an experiment, could you please confirm why not and add a note in the manuscript. Alternatively, if such a thing is easy to do, it might be worth performing such an experiment. However, I fully appreciate that while attractive and ostensibly simple to undertake with a model, the reality can be that too many changes are needed to perform such a control.**

We performed such simulations as sensitivity analyses during the development phase of the model and during the discussion phase. In addition to the configuration with mixotrophs, we also implemented two configurations without mixotrophs: a first one in which mixotrophs and their associated biogeochemical processes are simply deleted (D configuration) and a second one in which we replaced mixotrophs by organisms with strict diets (heterotroph and autotroph) (R configuration). Both configurations considered the same forcings as the typical simulation (Table 5).

We add these experiments to the Section 2.4.2 (l.323):

[In these conditions, we also performed two simulations without mixotrophs as control simulations. These simulations correspond to two configurations of Eco3M_MIX-CarbOx without mixotrophs: a first one in which mixotrophs and their associated biogeochemical processes are simply deleted (D configuration in Table 5) and a second one in which we replaced mixotrophs by organisms with strict diets (R configuration in Table 5). These configurations and their results are presented in the supplementary material.]

and Table 5:

**Table 5: Summary of simulations properties. Configurations without mixotrophs are detailed in the supplementary material.**

| Simulation name | Mixotrophs | [DIN] | [DIP] | Irradiance |
|---|---|---|---|---|
| **D configuration** | Absent | SOLEMIO interpolation | SOLEMIO interpolation | WRF |
| **R configuration** | Absent | SOLEMIO interpolation | SOLEMIO interpolation | WRF |
| **Typical** | Present | SOLEMIO interpolation | SOLEMIO interpolation | WRF |
| **Nutrient limited** | Present | $7.5 \times 10^{-3}$ mmolN m$^{-3}$ | $4.5 \times 10^{-4}$ mmolP m$^{-3}$ | WRF $\times$ 2 |
| **Light limited** | Present | 1.5 mmolN m$^{-3}$ | 0.09 mmolN m$^{-3}$ | WRF $\times$ 0.05 |

As suggested, we detailed these configurations (conceptual model, balance equations and modifications from the configuration with mixotrophs) and provided the simulations results (ecosystem composition in carbon biomass, dynamics of modelled organisms in carbon biomass for the three simulated years (2017 repeated three times), total chlorophyll, percentage of each prey in total copepod grazing, predation on copepods by higher trophic levels, and total photosynthesis and respiration fluxes) in the supplementary material which is attached in this file upload.

**Second, I would agree concerning the complexity of Figure 5. Would it be possible to expand the caption to make its content more self-explanatory? For instance, stating that the experiments shown represent hypothesis testing / sensitivity analysis of mixotroph formulation might be helpful. It's also**

**unclear why some experiments are illustrated through grazing while others show photosynthesis. And while Figure 5 has a NCMP4 panel, there's no corresponding entry in Table 2.**

We considered your suggestion and modified the caption of Figure 5:

[Figure 5: Assessing mixotrophs dynamics in the model: (a-d) NCM and (e-f) CM properties (cf., Table 2). Plotted values represent daily averages of grazing and photosynthesis fluxes.]

to:

[Figure 5: Assessment of mixotrophs representation in the model. Each frame represents the test of a property stated by Stoecker (1998) for NCM Type IIIB: grazing is independent of (a) DIM concentration (NCMP1), (c) irradiance (NCMP3), and photosynthesis (b) is independent of DIM concentration (NCMP2), (d) increases with food concentration (NCMP4) ; and CM Type IIA: photosynthesis increases with (e) food concentration (CMP1), (f) DIM concentration (CMP2), and grazing (g) decreases when DIM concentration increases (CMP3), (h) increases with irradiance (CMP4). Properties are detailed in Table 2 and associated simulations are detailed in Tables 3 and 4. Plotted values represent daily averaged grazing and photosynthesis. DIM: dissolved inorganic matter (sum of dissolved inorganic nitrogen and dissolved inorganic phosphorus).]

We hope that this modification will provide a better understanding of figure 5.

Thank you again for your comments, we hope that the above modifications will clarify these points. We remain at your disposal for any further information,

Sincerely,

Lucille Barré, on behalf of all co-authors.